# On the Collapse of Generative Paths:
# A Criterion and Correction for Diffusion Steering

Ziseok Lee [*† 1]   Minyeong Hwang [* 2]   Wooyeol Lee [1]   Sanghyun Jo [3 4]   Jihyung Ko [4]   Young Bin Park [5]
Jae-Mun Choi [5]   Eunho Yang [† 2 6]   Kyungsu Kim [† 1 4 7]

## Abstract

Inference-time steering adapts pretrained diffusion and flow models to new tasks without retraining, often utilizing ratio-of-densities constructions that reweight time-indexed marginals with fixed exponents. We identify *Marginal Path Collapse*, a failure mode in which the intermediate density defined by such compositions becomes nonnormalizable despite valid endpoints. This collapse can arise when composing heterogeneous experts trained with mismatched noise schedules (and/or negative exponents / partial supports). To address this, we provide (i) a sharp sufficient *Path Existence Criterion* that characterizes when the composed intermediate densities are mathematically well-defined, and (ii) *Adaptive Path Correction with Exponents (ACE)*, which generalizes Feynman–Kac steering to support time-varying exponents. Our analysis reveals that ACE controls the quantile radius of the intermediate distributions, providing a theoretical mechanism for path stabilization observed in experiments. On flexible-pose scaffold decoration, a drug design task composed of de-novo, conformer, and protein-conditioned experts, ACE prevents collapse and significantly outperforms constant-exponent baselines. Furthermore, ACE improves attribute success rates in compositional image generation, establishing it as a general framework for compositional sampling.

https://ziseoklee.github.io/projects/ACE/.
*: Equal Contribution, †: Corresponding Author. [1]Department of Biomedical Sciences, Seoul National University, Seoul, South Korea [2]Kim Jaechul Graduate School of AI, Seoul, South Korea [3]OGQ, South Korea [4]Interdisciplinary Program in AI, Seoul National University, Seoul, South Korea [5]Calici, South Korea [6]AITRICS, South Korea [7]School of Transdisciplinary Innovations, Seoul National University, Seoul, South Korea. Correspondence to: Kyungsu Kim <kyskim@snu.ac.kr>, Eunho Yang <eunhoy@kaist.ac.kr>, Ziseok Lee <ziseoklee@snu.ac.kr>.

*Proceedings of the 43rd International Conference on Machine Learning*, Seoul, South Korea. PMLR 306, 2026. Copyright 2026 by the author(s).

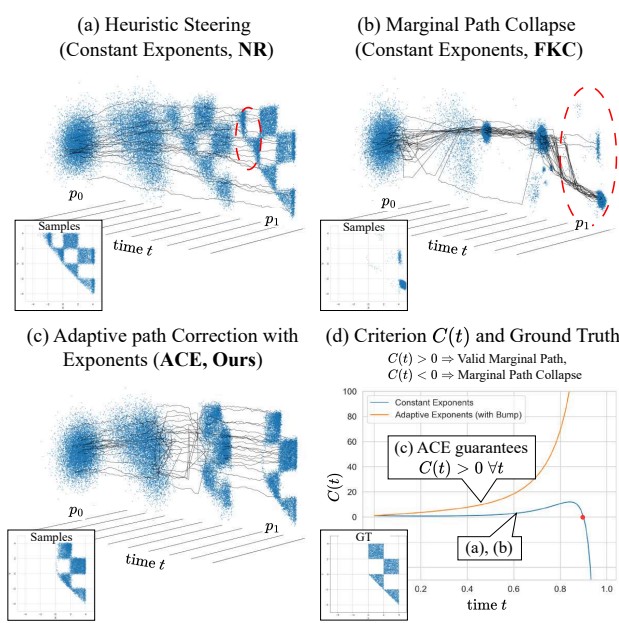

(a) Heuristic Steering (Constant Exponents, **NR**)

(b) Marginal Path Collapse (Constant Exponents, **FKC**)

(c) Adaptive path Correction with Exponents (**ACE, Ours**)

(d) Criterion $C(t)$ and Ground Truth
$C(t) > 0 \Rightarrow$ Valid Marginal Path,
$C(t) < 0 \Rightarrow$ Marginal Path Collapse
(c) ACE guarantees $C(t) > 0 \, \forall t$

*Figure 1.* **Marginal Path Collapse breaks heterogeneous composition.** We synthesize a 2D Checkerboard via three heterogeneous experts: two 1D priors and a 2D constraint. (a) **Standard Steering (NR)** is biased, failing to filter low-density artifacts. (b) **Feynman–Kac Corrector (FKC)** fails catastrophically because the path existence criterion $C(t)$ becomes negative (Collapse). (c) **ACE (Ours)** dynamically adjusts exponents to ensure $C(t) > 0$, recovering the ground truth. (d) **Validity Analysis:** ACE maintains positive criterion values while baselines dive into invalid regions.

## 1. Introduction

Generative models based on stochastic interpolants, such as diffusion (Ho et al., 2020; Song et al., 2021) and flow matching (Lipman et al., 2023), have become the standard for content creation and scientific discovery (Rombach et al., 2022; Schiff et al., 2025; Xie et al., 2024). A key to their success is *inference-time steering*: adapting pretrained models to new goals and tasks via *modular composition*, avoiding the prohibitive cost of monolithic retraining. A widely used steering mechanism is the *ratio-of-densities* $p(x)^{\gamma_1}/q(x)^{\gamma_2}$, which reweights probability landscapes to enforce constraints. More generally, we study compositions of time-indexed marginals $p_t^* \propto \prod_i q_{i,t}^{\gamma_i(t)}, \gamma_i(t) \in \mathbb{R}$. This

template subsumes classifier-free guidance[1] (Ho & Salimans, 2021), Bayesian composition, product-of-experts and more (see Appendix D.1). While these methods are typically applied in homogeneous settings (same noise schedule, same dimensions), we target the open challenge of general heterogeneous composition.

**On the Heterogeneity Gap.** Existing methods (Skreta et al., 2025a; Mark et al., 2025) implicitly rely on the constant-exponent ratio path being normalizable at all sampling times. While typically not an issue in homogeneous setups (e.g., standard CFG (Ho & Salimans, 2021)), this assumption breaks down for *heterogeneous* compositions essential to science. Here, heterogeneity means experts are trained on different noise schedules and data dimensions. For example, *scaffold decoration* (Xie et al., 2024) in drug design naturally combines heterogeneous de-novo (DN) (Hoogeboom et al., 2022; Ketata et al., 2025), conformer (CONF) (Xu et al., 2022; Hassan et al., 2024), and pocket-conditioned (SBDD) experts (Schneuing et al., 2024; Huang et al., 2024b) to generate molecules that satisfy topological, geometric, and pocket constraints simultaneously. Crucially, optimal performance requires task-specific noise schedules (e.g., rapid noise decay for refinement vs. slow decay for exploration) (Vignac et al., 2023; Lee et al., 2024; Choi et al., 2025; Seo et al., 2025), making heterogeneous scheduler combinations not merely incidental, but often necessary. Our DN/CONF/SBDD survey (Appendix E.5, Tables E.7–E.9) confirms that task-specific scheduling is standard. In fact, in our scaffold-decoration setting, forcing schedule alignment can be suboptimal (Appendix E.6).

**Marginal Path Collapse (MPC).** We identify a critical failure mode in these settings: *Marginal Path Collapse*. This occurs when the intermediate density becomes non-normalizable due to mismatched tail contraction induced by schedules and exponents, even if endpoints remain valid (Fig. 1). When collapse happens (the normalizer diverges), the score is mathematically undefined. While numerical SDE solvers may still run, they effectively simulate an unintended path, causing the terminal distribution to diverge from the target. We show that MPC arises naturally and can be common when composing mismatched schedules (Appendix E.4).

**Proposed Framework for Guaranteed Validity.** We provide a comprehensive diagnosis and solution (Table 1):

**1. Diagnosis: Path Existence Criterion (PEC).** We derive a rigorous criterion that certifies path existence when $C(t) > 0$ and detects collapse when $C(t) < 0$, based on schedules and exponents. The PEC explains precisely why constant-exponent baselines fail. Under Gaussian-to-compactly-supported settings, these are sharp universal con-

[1]CFG: $p_t^* = p_t(x \mid y)^\gamma / p_t(x)^{\gamma-1}$

ditions; the boundary case $C(t) = 0$ is handled separately in Appendix B.2.

**2. Solution: Adaptive Path Correction with Exponents (ACE).** We generalize Feynman–Kac steering to support time-varying exponents. ACE dynamically adjusts steering weights to satisfy the PEC, provably ensuring path existence up to the final timestep. We also show that ACE acts as a variance reduction mechanism, minimizing the quantile radius of intermediate distributions to stabilize sampling.

**Experimental Validation.** On a synthetic benchmark, ACE eliminates collapse and reduces error by $4\times$. On flexible-pose scaffold decoration, ACE enables the stable composition of DN/CONF/SBDD experts, surpassing constant-exponent baselines across metrics. In our flexible-pose setting, ACE exceeds specialized monolithic baselines in optimization success rates while maintaining competitive drug-likeness. Finally, on COCO-MIG (Appendix E.7), ACE improves attribute success rates ($+9.6\%p$) over constant baselines, establishing it as a general framework for both heterogeneous and homogeneous diffusion steering.

**Conflict of Interest Disclosure.** The authors declare no financial conflicts of interest related to this work.

*Table 1.* Comparison of steering methodologies. Unlike heuristics or standard correctors (FKC) which assume homogeneity, ACE guarantees path validity under general heterogeneous conditions.

| Feature | Heuristics (e.g., CFG) | FKC (Skreta et al., 2025a) | ACE (Ours) |
|---|---|---|---|
| Heterogeneity Support | Heuristically | ✗ (requires homogeneity) | ✓ |
| Time-Varying $\gamma(t)$ | Heuristically | ✗ (constant $\gamma$) | ✓ |
| Path Existence under Heterogeneity | ✗ | ✗ | **Guaranteed under PEC** |

## 2. Method

**Step 1.** Compute the path existence criterion $C(t)$ from noise schedules and exponents. If it goes negative or near-zero, constant-exponent steering is invalid/unstable.

**Step 2.** Choose bump function(s) to modify either one positive-exponent expert, or a covering subset of positive-exponent experts under partial support, so that $C_k(t) \geq \delta > 0$ for every covered coordinate $k$ on the sampling timesteps $\{0, t_1, \ldots, t_{\text{end}}\}$.

**Step 3.** Sample the corrected path with a weighted particle method (ACE), which reduces to FKC when $\dot{\gamma}_i(t) = 0$.

### 2.1. Preliminaries: Heterogeneous Ratio-of-Densities

**Heterogeneous Experts via Stochastic Interpolants.** We consider $n$ expert probability paths (or *experts*) $\{q_t^{(i)}\}_{t\in[0,1]}$, where each expert $i$ is generated by a distinct *stochastic interpolant* $X_t^{(i)} = \alpha_t^{(i)} X_0^{(i)} + \beta_t^{(i)} X_1^{(i)}$. Here, we assume $X_0^{(i)} \sim \mathcal{N}(0, I_{d_i})$, $X_1^{(i)} \sim q_1^{(i)}$ is the expert's target, and

$\alpha_t^{(i)}, \beta_t^{(i)}$ are differentiable noise schedules satisfying standard conditions. This formulation unifies diffusion and flow matching (Albergo et al., 2025). Crucially, we allow for *heterogeneous noise schedules* where $\alpha_t^{(i)} \neq \alpha_t^{(j)}$.

Associated with each expert is a stochastic differential equation (SDE) on $\mathbb{R}^{d_i}$: $X_0^{(i)} \sim \mathcal{N}(0, I_{d_i})$ and

$$dX_t^{(i)} = \left(v_t^{(i)}(X_t^{(i)}) + \frac{(\sigma_t^{(i)})^2}{2} s_t^{(i)}(X_t^{(i)})\right)dt + \sigma_t^{(i)} dW_t^{(i)} \tag{1}$$

where $s_t^{(i)} = \nabla \log q_t^{(i)}$ is the score and $v_t^{(i)}$ is the velocity.
**The Composed Path.** To compose experts of varying dimensionalities $d_i$, we embed them into a common ambient space $\mathbb{R}^d$ ($d = \max_i d_i$). Given time-dependent exponents $\gamma_i(t)$, the *heterogeneous ratio-of-densities* is defined as:

$$h_t(x) := \prod_{i=1}^{n} \left(\tilde{q}_t^{(i)}(x)\right)^{\gamma_i(t)}, \qquad x \in \mathbb{R}^d. \tag{2}$$

We say the family[2] has the *path existence property* if $Z_t = \int_{\mathbb{R}^d} h_t(x)dx < \infty$ for all $t \in [0, t_{\text{end}}]$. In this case, $p_t^* = h_t/Z_t$ is the valid normalized probability path.

**Path existence requirement.** SDE/ODE samplers and Feynman–Kac correctors assume that $p_t^* = h_t/Z_t$ exists at every timestep, requiring the $h_t$ to be integrable. If $h_t$ becomes non-normalizable (*Marginal Path Collapse*), then $p_t^*$ and its score cease to exist. The sampler still solves a well-posed ODE/SDE, but the resulting flow transports a different density path $\{p_t'\}$ rather than the intended $\{p_t^*\}$. The terminal distribution $p_1'$ produced by the sampler no longer matches the desired target $p_1^*$ (see Appendix A.2).

## 2.2. Marginal Path Collapse

Even if both endpoints $h_0$ and $h_1$ are integrable, *path-existence* is not guaranteed. A simple Gaussian example demonstrates the phenomenon (Figure 2).

**Why collapse occurs.** Let $h_t(x) = q_t^{(1)} q_t^{(2)}/q_t^{(3)} q_t^{(4)}$, where each component is a Gaussian path[3] under the linear interpolant $X_t = (1-t)X_0 + tX_1$:

$$q_t^{(1)} = \mathcal{N}\left(0, \left((1-t)^2 + \tfrac{1}{2}t^2\right)I\right),$$

$$q_t^{(2)} = \mathcal{N}\left(0, \left((1-t)^2 + 7t^2\right)I\right),$$

$$q_t^{(3)} = q_t^{(4)} = \mathcal{N}\left(0, \left(\tfrac{3}{2}(1-t)^2 + t^2\right)I\right) \tag{3}$$

At $t = 0$ and $t = 1$, the ratio $h_t \propto \mathcal{N}(0, \sigma_{\text{eff}}^2(t)I)$ for some finite $\sigma_{\text{eff}}^2(t)$, making it integrable. However,

[2] For expert $i$ acting on coordinates $I_i \subset [d]$, the projection $\pi_i : \mathbb{R}^d \to \mathbb{R}^{d_i}$ and embedding $\iota_i : \mathbb{R}^{d_i} \to \mathbb{R}^d$ are used for $\tilde{q}_t^{(i)}(x) := q_t^{(i)}(\pi_i(x))$ and vector fields $\tilde{v}_t^{(i)} := \iota_i \circ v_t^{(i)} \circ \pi_i$.
[3] A path from $q_0 = \mathcal{N}(0, \sigma_1^2 I)$ to $q_1 = \mathcal{N}(0, \sigma_2^2 I)$ has the closed form expression $q_t = \mathcal{N}\left(0, (\sigma_1^2(1-t)^2 + \sigma_2^2 t^2)I\right)$

by directly computing the ratio at $t = 0.5$, $h_t(x) \geq C \cdot \exp(+0.01\|x\|^2)$, for some constant $C > 0$, which is not integrable on $\mathbb{R}^d$. In fact, we can plot the probability path $p_t^*$ and its effective variance $\sigma_{\text{eff}}^2(t)$ across time as in Figure 2, showing that intermediate paths do not exist despite valid endpoints. We analyze the more general Gaussian mixture case in Appendix B.4.

The Gaussian example provides a crucial intuition: *Marginal Path Collapse* occurs when the variances of the numerator terms shrink "slower" than the variances of the denominator terms. This creates a temporary, fatal imbalance where the combined density becomes explosive rather than decaying at infinity. While this closed-form example is illustrative, most real-world models, especially those operating on complex data like molecules or images, involve non-Gaussian and *compactly supported target distributions* where such a direct variance calculation is impossible.

A natural question arises: can we find a general criterion that diagnoses the risk of collapse for these more complex cases, without needing a closed-form expression for the path?

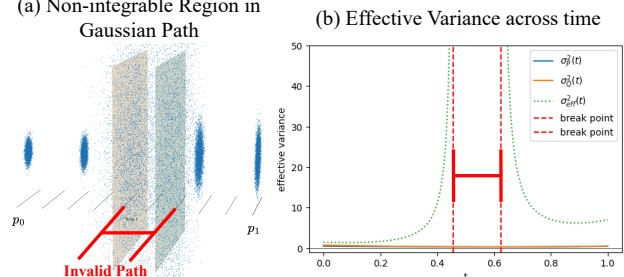

*Figure 2.* **Non-integrable Region in the ratio-of-Gaussians example (Eq. 3).** The path is well-defined at the endpoints, but the intermediate variance explodes, causing *Marginal Path Collapse*.

## 2.3. Path Existence Criterion for Compactly Supported Targets

Compactly supported distributions are common in creative and scientific applications, where data are bounded by physical constraints. In this setting, we can derive a clean criterion $C(t)$ for path existence.

---

**Theorem 2.1** (Path Existence Criterion (PEC)). *For each $i \in [n]$, let $\gamma_i(t), \alpha_t^{(i)}, \{q_t^{(i)}\}_{t \in [0,1]}, h_t(x), I_i$ be as defined in Sec. 2.1. We only assume additionally that $q_1^{(i)}$ has compact support for all $i \in [n]$.*

*If $h_1(x)$ is integrable and for every coordinate $k \in \{1, \ldots, d\}$ and all $t \in [0, t_{end}]$ ($t_{end} < 1$),*

$$C_k(t) := \sum_{i:\, k \in I_i} \frac{\gamma_i(t)}{(\alpha_t^{(i)})^2} > 0, \tag{4}$$

---

*then $\{h_t\}_{t\in[0,t_{end}]}$ has the path existence property. Conversely, if there exists a coordinate $k^* \in \{1,\ldots,d\}$ and $t^* \in [0,t_{end}]$ such that $C_{k^*}(t^*) < 0$, then $h_t$ is not integrable at $t^*$ (Marginal Path Collapse). We write the PEC by $C(t) = \min_k C_k(t)$.*

We provide a proof in Theorem B.1. This theorem provides a tractable checklist: to certify path existence, one only needs to verify endpoint integrability and the positivity of the coefficients $C_k(t)$, which in practice can be checked on the discrete timesteps used by the sampler $\{0, t_1, ..., t_{end}\}$. As for the boundary case $C(t) = 0$, Example B.1 gives a 1D example showing that $C(t) = 0$ can yield either path existence or collapse. Hence, $C(t) > 0$ and $C(t) < 0$ represent the sharpest possible universal conditions.

Figure 3 illustrates the effect of this test on standard noise schedules. Panel (a) plots $C(t)$ for several heterogeneous compositions $h_t = q_t^{(1)} q_t^{(2)} / q_t^{(3)}$ built from linear, cosine, DDPM, and related schedules, revealing that many combinations enter a region with $C(t) < 0$. This shows that widely used heuristic schedules can induce Marginal Path Collapse even when each individual expert is well behaved.

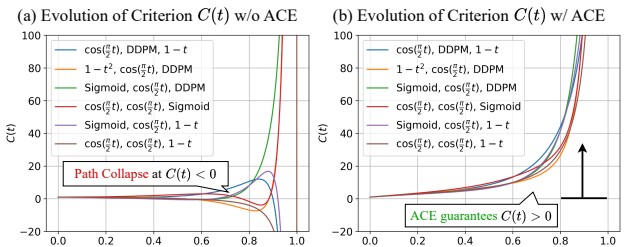

Figure 3. **Common noise schedules and Marginal Path Collapse.** (a) Path-existence criterion $C(t)$ (Eq. 4) for several heterogeneous three-expert compositions $h_t = q_t^{(1)} q_t^{(2)} / q_t^{(3)}$, formed from these schedules. Many combinations enter a region where $C(t) < 0$, implying non-normalizable intermediate densities. (b) The corrected exponents of ACE ensure $C(t) > 0$ for all $t \le t_{\text{end}}$, guaranteeing path existence.

---

**Proposition 2.1** (Concentration Control (Informal)). *Under the Gaussian-prior-to-compactly-supported-target assumptions of Theorem 2.1, the intermediate density $p_t^*$ satisfies sub-Gaussian tail bounds controlled by the PEC $C(t)$. Specifically, the $(1-\varepsilon)$-quantile radius $R_t(\varepsilon)$ scales approximately as:*

$$R_t(\varepsilon) \approx O\left(\frac{1}{\sqrt{C(t)}}\right). \tag{5}$$

---

The formal proof is provided in Proposition E.1. We verify the requisite quadratic envelope condition in Proposition B.2, which establishes that Gaussian-to-compact interpolants admit the bounds necessary for both Theorem 2.1 and the concentration result above.

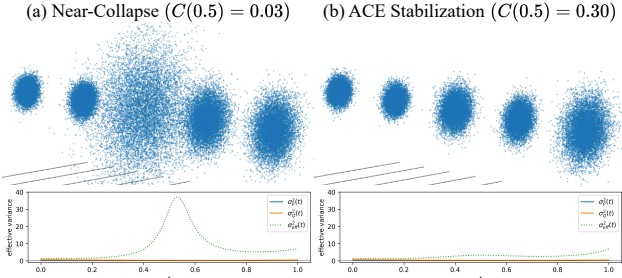

Figure 4. **Concentration control via Precision $C(t)$.** We illustrate Proposition 2.1 on the example path (Eq. 3). (a) A path near the existence boundary ($C(t) \approx 0$) exhibits extreme variance expansion (Green curve) and dispersed trajectories, consistent with the inverse-precision radius bound $R_t \propto C(t)^{-1/2}$. (b) ACE raises $C(t)$, strictly bounding the quantile radius and forcing trajectories to stay concentrated near the mode.

**Interpretation.** This result bridges the gap between binary validity and continuous quality. Even if the path theoretically exists (valid), a vanishing $C(t) \to 0^+$ implies $R_t \to \infty$. This yields large effective radii, producing overly diffuse intermediate distributions that destabilize particle weights and degrade sample quality (Fig. 4, (a)). ACE adapts exponents over time to keep $C(t)$ positive *and* bounded away from zero. This acts as a focusing force, shrinking the effective radius and concentrating probability mass along the optimal trajectory (Fig. 4, (b)).

This discovery motivates the central goal of our work: to develop a method that can correct any given set of schedules to ensure the path existence criterion is always satisfied, thereby transforming unstable heuristics into a robust, guaranteed methodology. We introduce this method next.

### 2.4. Adaptive Path Correction with Exponents: Bump Function Protocol

We develop our correction protocol, ACE, by first constructing a valid exponent schedule $\tilde{\gamma}_i(t)$ and then deriving the sampling dynamics that follow this corrected path.

In practice, we need control over the initial distribution $p_0^* = \Pi_{i=1}^n (p_0^{(i)})^{\gamma_i(0)}/Z_0$ (usually fixed to $\mathcal{N}(0, I)$) and the target distribution $p_1^* = \Pi_{i=1}^n (p_1^{(i)})^{\gamma_i(1)}/Z_1$. However, we do not need to fix the intermediate marginals $p_t^*$, $t \in (0,1)$. The idea is to choose an appropriate $\tilde{\gamma}_i(t)$ that preserves the original exponent values at the beginning and end, $\tilde{\gamma}_i(0) = \gamma_i(0), \tilde{\gamma}_i(1) = \gamma_i(1)$, while ensuring the intermediate densities are all normalizable.

---

**Theorem 2.2** (Adaptive Exponents (Theorem B.2)). *Let a set of noise schedules $\{\alpha_t^{(i)}\}$ and exponent boundary values $\{\gamma_i(0), \gamma_i(1)\}$ be given. Assume that the criterion $C(t)$ (Theorem 2.1) is positive at $t = 0$.*

*Then, there exists a set of differentiable functions $\{\tilde{\gamma}_i(t)\}_{i=1}^n$ such that $\tilde{\gamma}_i(0) = \gamma_i(0), \tilde{\gamma}_i(1) = \gamma_i(1)$*

---

*for all $i$, and $C(t) > 0$ is satisfied for all $t \in [0, t_{end}]$.*

We provide a constructive proof in Theorem B.2 showing that there always exist positive constants $B_1, B_2, \tau > 0$ such that changing the exponent schedules of a covering subset of positive-exponent experts ensures coordinate-wise path existence. In particular, when one expert covers all co-ordinates, this reduces to a single bump $\tilde{\gamma}_j(t) = \gamma_j(t) + b(t)$ for one index $j$. We concretely design the bump function $b(t) = B_1 Q(t) + B_2 L_\tau(t)$ using quadratic $Q(t) = t(1-t)$ and linear $L_\tau(t) = \min(t, \tau(1-t))$ "bump" functions whose endpoint values are fixed to 0 and intermediate values are strictly positive.

**ACE Sampler.** Correcting $\gamma_j(t)$ fixes the target path. Now we have established "what" we are going to sample from. The next question is "how" to sample from the corrected probability path (i.e., a sampler that remains correct when $\dot{\gamma}_j(t) \neq 0$). This introduces an additional weight term $\sum_i \dot{\gamma}_i(t) \log \tilde{q}_t^{(i)}(X_t)$, which ACE tracks via auxiliary dynamics. By extending the Feynman–Kac (Skreta et al., 2025a) weighted SDE to time-dependent exponents $\gamma_i(t)$, the following theorem establishes our sampling algorithm:

---

**Theorem 2.3** (ACE Sampling Dynamics). *Assume the Path Existence Criterion (Theorem 2.1) holds. Let $v_t^*$ be a chosen vector field (e.g., velocity of the geometric mean). The target path $p_t^* \propto \prod(\tilde{q}_t^{(i)})^{\gamma_i(t)}$ is realized by the weighted SDE:*

$$dX_t = \left( v_t^*(X_t) + \frac{\sigma_t^2}{2} s_t^*(X_t) \right) dt + \sigma_t dW_t, \quad (6)$$

*where $s_t^* = \sum \gamma_i(t) \tilde{s}_t^{(i)}$ is the weighted score. Crucially, to account for the time-varying exponents $\dot{\gamma}_i(t)$, the particle importance weights $w_t$ must evolve via:*

$$d \log w_t(X_t) = \left[ \mathcal{F}\left( X_t, s_t^*, v_t^*, \{\gamma_i, \tilde{s}_t^{(i)}, \tilde{v}_t^{(i)}\}_{i=1}^n \right) \right.$$

$$\left. + \sum_{i=1}^n \dot{\gamma}_i(t) \log \tilde{q}_t^{(i)}(X_t) \right] dt \quad (7)$$

*The explicit form of $\mathcal{F}$ and the auxiliary SDE (derived via Itô's formula) for tracking $\log \tilde{q}_t^{(i)}(X_t)$'s are provided in Theorem A.1.*

---

We provide the full derivation and proof in Theorem A.1. Note we can minimize costly divergence computations when simulating the SDE in Theorem 2.3 by choosing $v_t^* = \sum_{i=1}^n \gamma_i(t) \tilde{v}_t^{(i)}$. We used this choice in our scaffold decoration experiment for numeric stability and efficiency.

**Practical Remark.** Theorem 2.3 characterizes the weighted SDE whose marginal $p_t^*$ follows the corrected ratio-of-densities path. In practice, we implement this dynamics using a particle system with importance weights: at each step we propagate particles under the drift and diffusion in Eq. 6, update their log-weights via Eq. 7, and trigger a resampling step whenever the effective sample size (ESS) falls below a threshold. During resampling, high-weight samples are duplicated and low-weight ones are removed in proportion to their weights. As illustrated in Figure 5, this procedure eliminates out-of-distribution trajectories in ACE, whereas the no-resampling heuristic (NR) leaves invalid samples in the batch. The complete algorithm for Theorem 2.3 is provided in Algorithm 1.

(a) Path Correction via Resampling (**ACE**): Successfully removes invalid samples

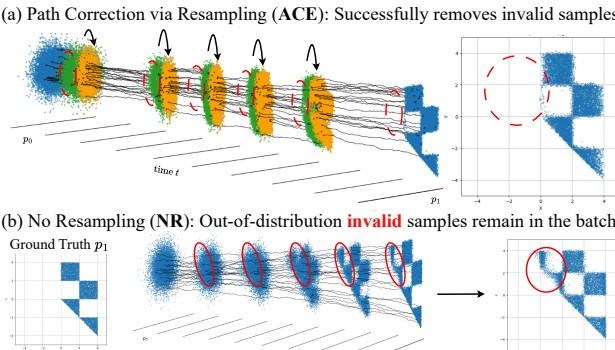

(b) No Resampling (**NR**): Out-of-distribution **invalid** samples remain in the batch

*Figure 5.* **Visualization of the sampling trajectories.** (a) ACE appropriately assigns weights to valid samples such that at each resampling step (green-to-orange), invalid samples are discarded. (b) No resampling (NR), a common heuristic (e.g., CFG), has no corrective mechanism that removes out-of-distribution samples.

**ACE as a repair pipeline.** ACE should be understood as a path-repair framework rather than merely a sampler applied after validity is assumed. The original constant-exponent ratio path is first diagnosed by the Path Existence Criterion. When the criterion is violated, Theorem 2.2 modifies the exponent schedule while preserving the endpoint exponents, thereby restoring a normalizable path. The weighted dynamics of Theorem 2.3 are then applied to this corrected path. Thus, standard CFG/NR/FKC may remain numerically computable on a collapsed path, but they are no longer transporting the intended ratio-of-densities path; ACE restores the missing probability path before sampling it.

**Remark.** Standard Feynman–Kac Correctors (Skreta et al., 2025a) assume constant constraints ($\dot{\gamma} = 0$), in which case the *Adaptive Correction* vanishes. ACE generalizes this by explicitly tracking the component log-densities $\log \tilde{q}_t^{(i)}$ (Eq. A.9) to correct for the distributional shift induced by the changing schedule. Compare Algorithms 1 and 2.

## 2.5. Application: Flexible-Pose Scaffold Decoration

We now demonstrate the practical necessity of our framework on *flexible-pose scaffold decoration*, a task that illustrates all components of the heterogeneous ratio-of-densities setting developed so far. Here the goal is to generate 3D molecules $\mathcal{M} = (\mathcal{M}^{sc}, \mathcal{M}^R)$ that preserve a given scaf-

fold bond topology $\mathcal{T}^{\text{sc}}$ and stably bind to a protein pocket $\mathcal{P}$, while allowing the scaffold's 3D pose to adapt within the pocket (Figure 6). Let $A = \{\mathcal{T}(\mathcal{M}^{\text{sc}}) = \mathcal{T}^{\text{sc}}\}$ and $B = \{\mathcal{M} \leftrightarrow \mathcal{P}\}$ where $\mathcal{M} \leftrightarrow \mathcal{P}$ denotes stable binding[4] and $\mathcal{T}(\cdot)$ extracts molecular topology (Appendix E.1). Since the scaffold coordinates determine the scaffold topology, $A$ gives no additional information about the R-group or pocket binding once $\mathcal{M}^{\text{sc}}$ is known. Thus $A \perp (\mathcal{M}^{\text{R}}, B) \mid \mathcal{M}^{\text{sc}}$, and Bayes' rule gives the heterogeneous ratio-of-densities:

$$p(\mathcal{M}^{\text{sc}}, \mathcal{M}^{\text{R}} \mid A, B) \tag{8}$$

$$\propto p(\mathcal{M}^{\text{sc}}, \mathcal{M}^{\text{R}}, B \mid A) \tag{9}$$

$$\propto p(\mathcal{M}^{\text{sc}} \mid A)p(\mathcal{M}^{\text{R}}, B \mid A, \mathcal{M}^{\text{sc}}) \tag{10}$$

$$\propto p(\mathcal{M}^{\text{sc}} \mid A)\frac{p(\mathcal{M}^{\text{sc}}, \mathcal{M}^{\text{R}} \mid B)}{p(\mathcal{M}^{\text{sc}})} \tag{11}$$

This yields Eq. 12. Importantly, Eq. 12 is a decomposition of the endpoint target distribution, not an assumption that the noisy intermediate marginals satisfy the same Bayesian identity. The intermediate function $h_t$, when normalizable, defines an algorithmic transport path $p_t^* = h_t/Z_t$. This path does not assign physical meaning to noisy molecules, but acts as a probability bridge between Gaussian noise and the decomposed endpoint target.

$$p(\mathcal{M} \mid \mathcal{T}^{\text{sc}}, \mathcal{P}) \underset{\text{Bayes}}{\propto} \frac{p(\mathcal{M}^{\text{sc}} \mid \mathcal{T}(\mathcal{M}^{\text{sc}}) = \mathcal{T}^{\text{sc}}) \, p(\mathcal{M} \mid \mathcal{M} \leftrightarrow \mathcal{P})}{p(\mathcal{M}^{\text{sc}})} \tag{12}$$

We introduce a guidance scale $\omega \geq 1$ so that larger $\omega$ enforces stronger scaffold and pocket conditioning.

$$p_\omega(\mathcal{M}) \propto p(\mathcal{M}) \left( \frac{p(\mathcal{M} \mid \mathcal{T}^{\text{sc}}, \mathcal{P})}{p(\mathcal{M})} \right)^\omega \tag{13}$$

This formulation (Eq. 12–13) decomposes into four factors implemented with three pretrained diffusion experts:

- ($q^{(1)}$ and $q^{(2)}$) Unconditional de-novo model (DN) for $p(\mathcal{M}^{\text{sc}})$ and $p(\mathcal{M})$

- ($q^{(3)}$) Topology-conditioned conformer model (CONF) for $p(\mathcal{M}^{\text{sc}} \mid \mathcal{T}^{\text{sc}})$

- ($q^{(4)}$) Pocket-conditioned SBDD model for $p(\mathcal{M} \mid \mathcal{P})$

Writing the four factors with exponents $\gamma_1(t) = -\omega$, $\gamma_2(t) = -(\omega - 1)$, $\gamma_3(t) = \omega$, $\gamma_4(t) = \omega$ shows that Eq. 13 is exactly a *heterogeneous* ratio-of-densities composition of the form $p_t^* \propto \prod_i (q_t^{(i)})^{\gamma_i(t)}$ (Sec. 2.1).

For the *heterogeneous noise schedules* of the DN, CONF, and SBDD experts used here (full survey in Tables E.7– E.9), the constant-exponent path obtained from Eq. 13 violates the path-existence criterion (Theorem 2.1) for guidance scales $\omega \geq 1.1$, resulting in Marginal Path Collapse.

---

[4]Defined via docking energy $U^{\text{Dock}}(\mathcal{M} \leftrightarrow \mathcal{P}) < \tau^{\text{Dock}}$.

Therefore, to raise $\omega$ while preserving path existence, we construct an *ACE-corrected* exponent schedule using Theorem 2.2. In practice, we apply a bump function of the form $b(t) = B_1 \, t(1-t) + B_2 \, \min(t, \frac{t_{\text{end}}}{1-t_{\text{end}}}(1-t))$, where $B_1, B_2$ are positive constants. This bump is applied to one of the positive-exponent terms (specifically, $\gamma_4$), which is sufficient to ensure $C(t) > 0$ for all $t$, thereby guaranteeing a valid probability path. Finally, by simulating the importance-weighted SDE of Theorem 2.3, ACE provides samples consistent with the corrected path, and hence from the desired target distribution $p_\omega(\mathcal{M})$ in Eq. 13.

# 3. Experiments

**Synthetic Checker Dataset.** We construct a synthetic benchmark that mirrors the heterogeneous conditioning structure encountered in our molecular task (Eq. 12). In both settings, one condition acts *locally* on a subset of variables (e.g., scaffold topology $A$ affecting only $\mathcal{M}^{\text{sc}}$), while another acts *globally* on the full configuration (e.g., pocket binding $B$ affecting $(\mathcal{M}^{\text{sc}}, \mathcal{M}^{\text{R}})$). This form of heterogeneous conditioning is pervasive in scientific problems such as scaffold decoration, linker generation, and protein–protein glue design as discussed in Appendix E.1.

To create the simplest possible analog of Eq. 12, we let $X$ and $Y$ be 1D variables having joint prior $(X, Y) \sim p_{\text{Checker}}$ and impose two constraints: $A = \{X \geq 0\}$ (local constraint on $X$) and $B = \{X + Y \geq 0\}$ (global constraint coupling $(X, Y)$). This induces the heterogeneous factorization

$$p(X, Y \mid A, B) \underset{\text{Bayes}}{\propto} p(X, Y, B \mid A)$$

$$\underset{\text{Bayes}}{\propto} p(X \mid A)p(Y, B \mid A, X)$$

$$\underset{(*)}{\propto} p(X \mid A)\frac{p(X, Y \mid B)}{p(X)} \tag{14}$$

Note that this factorization is required only at the target endpoint. For $(*)$, we use the fact that once $X$ is given, $A$ has no effect on $Y, B$, which can be expressed by $A \perp Y, B \mid X$, implying $p(Y, B \mid A, X) = p(Y, B \mid X)$. Thus, exactly as in Eq. 12, the target distribution naturally decomposes into *three heterogeneous experts*: a 1D expert for $p(X \mid A)$, a 2D expert for $p(X, Y \mid B)$, and a 1D expert for $p(X)$.

To test the theoretical phenomenon identified by our path-existence criterion, for Table 2, we deliberately used experts pretrained under a set of noise schedules $\alpha_t^{(1)} = \cos(\frac{\pi}{2}t)$, $\alpha_t^{(2)} = \text{DDPM}$, $\alpha_t^{(3)} = 1 - t$ known to induce *Marginal Path Collapse* (See Figure 3). Importantly, the collapse behavior we observe here is *not* specific to this choice: Figures E.11–E.16 evaluate other combinations of *widely used* schedules (DDPM, sigmoid, linear, cosine, polynomial) and show that the same failure patterns arise. Mathematical definitions of $p_{\text{Checker}}$ and schedulers as well as

*Table 2.* Distributional similarity metrics ($\downarrow$). For each metric, we report the minimum and mean $\pm$ standard deviation across 5 seeds. Best values are in **bold**. NR denotes no resampling, the common heuristic using only mixed scores. FKC refers to Feynman–Kac correctors, which fail when path existence is not satisfied.

| Method | $B_1$ | $W_1$ ($\downarrow$) | | $W_2$ ($\downarrow$) | | MMD (RBF) ($\downarrow$) | |
|---|---|---|---|---|---|---|---|
| | | Mean $\pm$ Std | Max | Mean $\pm$ Std | Max | Mean $\pm$ Std | Max |
| NR | - | $0.89 \pm 0.02$ | 0.91 | $1.18 \pm 0.02$ | 1.20 | $0.092 \pm 0.004$ | 0.098 |
| FKC | - | $1.37 \pm 1.09$ | 2.92 | $1.59 \pm 1.16$ | 3.24 | $0.419 \pm 0.579$ | 1.380 |
| | 0 | $0.78 \pm 0.15$ | 1.06 | $1.00 \pm 0.17$ | 1.29 | $0.092 \pm 0.027$ | 0.131 |
| | 10 | $\mathbf{0.20 \pm 0.04}$ | 0.25 | $\mathbf{0.29 \pm 0.05}$ | 0.36 | $\mathbf{0.012 \pm 0.003}$ | 0.015 |
| ACE | 20 | $0.39 \pm 0.24$ | 0.82 | $0.54 \pm 0.31$ | 1.08 | $0.035 \pm 0.025$ | 0.080 |
| ($B_2 = 1.5$) | 30 | $0.31 \pm 0.04$ | 0.37 | $0.46 \pm 0.10$ | 0.57 | $0.032 \pm 0.005$ | 0.040 |
| | 40 | $0.47 \pm 0.13$ | 0.82 | $0.65 \pm 0.16$ | 1.09 | $0.072 \pm 0.041$ | 0.184 |
| | 50 | $0.66 \pm 0.25$ | 1.11 | $0.87 \pm 0.29$ | 1.37 | $0.153 \pm 0.140$ | 0.402 |

hyperparameter studies are in Appendix C.

**Flexible-Pose Scaffold Decoration.** We evaluate ACE on a realistic scientific task requiring heterogeneous DN/CONF/SBDD composition. Dataset details, implementation, and metrics appear in Appendix C.2. Briefly, we combine pretrained models EDM (Hoogeboom et al., 2022) as DN, GeoDiff (Xu et al., 2022) as CONF, and DiffS-BDD (Schneuing et al., 2024) as SBDD as defined in Section 2.5. We evaluate on the test set of CrossDocked2020 (Francoeur et al., 2020). As diffusion-steering baselines we adopt NR and FKC (Skreta et al., 2025a), and as task-specific scaffold-decoration models we include Delete (Chen et al., 2025), and AutoFragDiff (Ghorbani et al., 2023). We also implement a practical variant of ACE (ACE-lite), omitting the resampling step to reduce computational overhead.

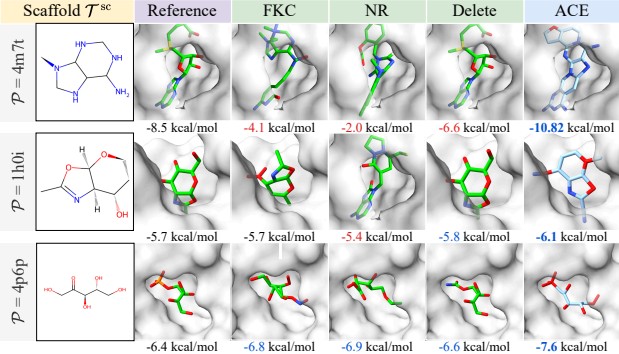

*Figure 6.* **ACE enables high-guidance steering for Flexible-Pose Scaffold Decoration**. ACE generates highly dockable molecules by accurately modeling the ratio-of-density path, whereas FKC and NR produce suboptimal results due to ill-defined probability paths. Our flexible-pose formulation further relaxes the fixed-pose constraint of Delete (Chen et al., 2025), allowing exploration of a larger search space.

### 3.1. Results

We present the main quantitative results in Tables 2–4. Across both synthetic and molecular settings, ACE achieves substantial improvements over NR and FKC, reflecting practical gains of correcting heterogeneous ratio-of-density paths to enforce the path existence. We clarify that ACE

differs from FKC only by its adaptive exponent correction[5] and NR is FKC without resampling; all other components and hyperparameters are shared across methods.

## 4. Discussion

**ACE Prevents Collapse at High Guidance Scales.** High guidance scales ($\omega > 1$) are crucial for generating high-affinity molecules but are also where the risk of path collapse is highest. Our results make this trade-off explicit. As shown in Tables 3 and 4, ACE prevents path collapse across all tested configurations, enabling performance to scale with guidance weight. At $\omega=1.4$, ACE reaches an average Vina score of $-7.10$ kcal/mol. In contrast, the FKC baseline suffers from catastrophic collapse: at $\omega=1.4$, its success rate drops (OSR 0.4), docking scores plateau ($-6.24$), and drug-likeness degrades (QED 0.44 vs. 0.53 for ACE).

**Prevalence of Marginal Path Collapse.** Collapse is not an edge case; it is the default behavior for heterogeneous composition. In Appendix E.4, we evaluated 125 annealed compositions of standard noise schedules (DDPM (Ho et al., 2020), cosine (Nichol & Dhariwal, 2021), sigmoid (Xu et al., 2022), linear (Lipman et al., 2023), and polynomial (Hoogeboom et al., 2022)). Excluding trivial homogeneous cases, the collapse rate for heterogeneous pairs increases sharply with guidance scale: from 41% ($w=1.0$) to 80% ($w=15$) (Table E.10 and Figure E.10). Furthermore, our analysis suggests that the mere *presence* of collapse is more critical than its duration; even short intervals of non-normalizability ($< 10\%$ of trajectory) are sufficient to destabilize importance weights in FKC, whereas ACE reliably recovers the target distribution regardless of collapse duration (Figures E.11–E.16). The failure trends are consistent: NR suffers from inherent approximation errors regardless of path existence, whereas FKC performance degrades specifically because criterion violations destabilize the importance weights. In contrast, ACE reliably restores valid paths and consistently recovers the target distribution across all tested durations.

**Finite scores do not imply valid steering.** Marginal Path Collapse is a global normalizability failure, not necessarily a local numerical failure. The mixed score field may remain finite, so an SDE/ODE solver can still run. However, once $Z_t = \infty$ for some $t$, this field is no longer the score of the intended density $p_t^* = h_t/Z_t$. The solver therefore transports an unintended path $p_t'$, and any later agreement with the target would require accidental off-path recovery. Example A.1 makes this explicit in a Gaussian example where the mixed-score dynamics remain well posed but reach the wrong terminal distribution.

---

[5] $B_1 \in [0, 50]$, $B_2=1.5$ for the 2D experiment and $B_1 = 30$, $B_2 = 0.037, 0.136, 0.236, 0.336$ for $\omega = 1.1, 1.2, 1.3, 1.4$ for the scaffold decoration experiment

*Table 3.* Performance comparison on the CrossDocked dataset, evaluating docking affinity (Vina Score) and drug-likeness (QED, SA, and Lipinski). For OSR and drug-likeness, higher values indicate superior performance; for Vina scores, lower (more negative) values are preferred. Best results are highlighted in bold, and second-best results are underlined. NR and FKC suffer from path existence criterion (PEC) violations. Ref. denotes values for the reference molecule and Pocket-Worst indicates the average of the pocket-wise worst scores.

| Method | $\omega$ | PEC | OSR(↑) | Vina Score (↓) | | | | QED (↑) | | | SA (↑) | | | Lipinski (↑) | | |
|---|---|---|---|---|---|---|---|---|---|---|---|---|---|---|---|---|
| | | | | Pocket-Worst | Top25% | Avg | Std | Top25% | Top50% | Avg | Top25% | Top50% | Avg | Top25% | Top50% | Avg |
| Ref. | - | - | - | | -8.0 | -6.77 | 1.89 | 0.62 | 0.47 | 0.48 | 0.73 | 0.66 | 0.67 | 1.0 | 1.0 | 0.95 |
| ACE | 1.1 | ✓ | 0.71 | -6.74 | -8.30 | -7.02 | 0.19 | **0.65** | 0.50 | 0.51 | **0.69** | 0.56 | **0.57** | 1.00 | 1.00 | **0.99** |
| | 1.2 | ✓ | 0.65 | -6.78 | -8.40 | -7.08 | 0.20 | 0.63 | 0.49 | 0.51 | 0.65 | 0.55 | 0.55 | 1.00 | 1.00 | 0.98 |
| | 1.3 | ✓ | 0.68 | -6.64 | -8.20 | -6.91 | 0.19 | 0.62 | 0.49 | 0.50 | 0.67 | **0.58** | **0.57** | 1.00 | 1.00 | 0.97 |
| | 1.4 | ✓ | **0.75** | **-6.84** | **-8.70** | **-7.10** | 0.19 | 0.64 | **0.54** | **0.53** | 0.65 | 0.57 | 0.56 | 1.00 | 1.00 | 0.98 |
| FKC | 1.1 | ✗ | 0.52 | -5.99 | -7.90 | -6.54 | 0.37 | 0.62 | 0.45 | 0.47 | 0.68 | 0.55 | **0.57** | 1.00 | 1.00 | 0.98 |
| | 1.2 | ✗ | 0.43 | -5.56 | -7.40 | -6.21 | 0.45 | 0.61 | 0.47 | 0.47 | 0.64 | 0.54 | 0.55 | 1.00 | 1.00 | 0.97 |
| | 1.3 | ✗ | 0.49 | -5.90 | -7.90 | -6.45 | 0.40 | 0.54 | 0.42 | 0.41 | 0.67 | 0.56 | **0.57** | 1.00 | 1.00 | 0.97 |
| | 1.4 | ✗ | 0.40 | -5.53 | -7.50 | -6.24 | 0.46 | 0.56 | 0.46 | 0.44 | 0.65 | 0.55 | 0.56 | 1.00 | 1.00 | 0.98 |
| NR | 1.1 | ✗ | 0.46 | -4.98 | -7.70 | -6.32 | 0.82 | 0.54 | 0.42 | 0.42 | 0.62 | 0.52 | 0.53 | 1.00 | 1.00 | 0.97 |
| | 1.2 | ✗ | 0.46 | -5.13 | -7.60 | -6.33 | 0.77 | 0.52 | 0.42 | 0.41 | 0.64 | 0.54 | 0.54 | 1.00 | 1.00 | 0.96 |
| | 1.3 | ✗ | 0.42 | -5.19 | -7.50 | -6.23 | 0.73 | 0.56 | 0.45 | 0.44 | 0.62 | 0.54 | 0.54 | 1.00 | 1.00 | 0.98 |
| | 1.4 | ✗ | 0.47 | -5.10 | -7.70 | -6.35 | 0.82 | 0.54 | 0.39 | 0.40 | 0.64 | 0.55 | 0.54 | 1.00 | 1.00 | 0.97 |

*Table 4.* Performance comparison on the CrossDocked dataset against fixed-pose scaffold decoration baselines, evaluating docking affinity (Vina Score) and drug-likeness (QED, SA, and Lipinski). For OSR and drug-likeness, higher values indicate superior performance; for Vina scores, lower (more negative) values are preferred. Best results are highlighted in bold, and second-best results are underlined. Asterisks (*) denote models that require a reference ligand pose. Pocket-Worst indicates the average of the pocket-wise worst scores.

| Method | $\omega$ | OSR(↑) | Vina Score (↓) | | | | QED (↑) | | | SA (↑) | | | Lipinski (↑) | | |
|---|---|---|---|---|---|---|---|---|---|---|---|---|---|---|---|
| | | | Pocket-Worst | Top25% | Avg | Std | Top25% | Top50% | Avg | Top25% | Top50% | Avg | Top25% | Top50% | Avg |
| Reference | - | - | | -8.0 | -6.77 | 1.89 | 0.62 | 0.47 | 0.48 | 0.73 | 0.66 | 0.67 | 1.0 | 1.0 | 0.95 |
| AutoFragDiff* | - | 0.36 | -5.14 | -7.60 | -6.01 | 0.61 | 0.70 | **0.59** | **0.57** | **0.80** | **0.70** | **0.71** | 1.00 | 1.00 | 0.97 |
| Delete* | - | 0.47 | -3.78 | -8.30 | -5.07 | 1.26 | **0.71** | 0.55 | 0.55 | 0.75 | 0.65 | 0.66 | 1.00 | 1.00 | 0.98 |
| ACE | 1.1 | 0.71 | -6.74 | -8.30 | -7.02 | 0.19 | 0.65 | 0.50 | 0.51 | 0.69 | 0.56 | 0.57 | 1.00 | 1.00 | **0.99** |
| | 1.4 | **0.75** | **-6.84** | **-8.70** | **-7.10** | 0.19 | 0.64 | 0.54 | 0.53 | 0.65 | 0.57 | 0.56 | 1.00 | 1.00 | 0.98 |
| ACE-lite | 1.1 | 0.67 | -6.42 | -8.30 | -6.99 | 0.42 | 0.61 | 0.48 | 0.50 | 0.67 | 0.55 | 0.55 | 1.00 | 1.00 | 0.98 |
| | 1.4 | 0.66 | -6.30 | -8.50 | -7.00 | 0.52 | 0.62 | 0.50 | 0.50 | 0.66 | 0.55 | 0.56 | 1.00 | 1.00 | 0.97 |

*Table 5.* Computational cost reported in terms of VRAM usage (GB) and runtime (min/molecule). ACE-lite significantly reduces the runtime overhead while maintaining performance.

| Method | ACE | ACE-lite | Autofragdiff | Delete | FKC | NR |
|---|---|---|---|---|---|---|
| VRAM | 1.57 | 1.54 | 2.13 | 1.46 | 1.53 | 1.54 |
| Time | 0.34 | 0.19 | 0.29 | 0.31 | 0.19 | 0.19 |

**Sensitivity Analysis of Bump Parameters.** Since the path-existence criterion $C(t)$ depends solely on analytical schedules, the bump parameters $B_1, B_2$ can be pre-screened without expensive model evaluation. Empirically, we identify a stable region ($B_1 \in [10, 30]$, $B_2 \in [1.0, 2.0]$, see Tables C.3–C.5) that consistently yields strong performance. We observe a fundamental trade-off: while increasing $B_1, B_2$ guarantees path existence, these parameters also act as scalar multipliers on the score and velocity fields, linearly amplifying inherent network approximation errors. Intuitively, excessive correction steers the sampling path unnecessarily far from the optimal transport trajectory. We therefore recommend initializing with conservative values (e.g., $B_1 = 20, B_2 = 2$) and increasing them only if the criterion remains unsatisfied.

**Superiority over Specialized Baselines & Efficiency**

**Trade-offs.** ACE demonstrates that composing generalist experts can outperform specialized, monolithic models. As detailed in Table 4, ACE achieves a peak OSR (Optimization Success Rate) of **0.75**, significantly outperforming specialized fixed-pose baselines such as AutoFragDiff and Delete, which struggle with "hard" pose constraints (OSR ranging 0.22–0.47). ACE's flexible-pose framework navigates the chemical space more effectively to identify higher-quality candidates. To mitigate the computational overhead of multi-expert evaluation, we proposed *ACE-lite*, which skips time-intensive resampling. This variant reduces inference time by $\sim 44\%$ (0.34 to 0.19 min/molecule, see Table 5) while maintaining competitive success rates, though it exhibits higher variance in docking scores due to the reintroduction of bias. Although this entails a trade-off in worst-case performance due to non-filtering of invalid paths, ACE-lite maintains competitive OSR over baselines.

**Existing Time-varying Exponent Schedules.** To disentangle our bump function from general time-varying exponents in prior work, we tested empirical dynamic schedules (Wang et al., 2024b) on scaffold decoration (3 seeds) and 2D checker (5 seeds): linearly increasing ($w\alpha t$), decreasing ($w\alpha(1 - t)$), $\Lambda$-shaped ($w\alpha(0.5 - |t - 0.5|)$), and V-shaped ($w\alpha|t - 0.5|$) for $\alpha \in \{0.5, 1, 2, 4, 8\}$ ($w = 1.4$

for scaffold, $w = 1.0$ for 2D). For scaffold decoration, we also tested quadratic bumps $b(t) = B_1 t(1 - t)$ on $\gamma_4$ ($B_1 \in \{10, 20, 30\}$), scaled to deliberately fail the Path Existence Criterion. Across all samplers (ACE/FKC/NR), these heuristic exponent schedules failed to prevent path collapse, degrading performance versus ours. These results confirm that the gains come from validity-preserving exponent design rather than time variation alone; full results are in Appendix C.3.

**Schedule alignment experiment.** Using time-reparameterization (Lai et al., 2025), schedule alignment can restore marginal validity but remains suboptimal in terms of sample quality, as it disregards model-specific allocation of exploration and refinement across time. In the scaffold decoration experiment described in Appendix E.6, FKC with aligned schedules consistently underperforms ACE due to the non-uniform time evolution induced by reparameterization, which skips task-critical regions of the trajectory and destabilizes the composed generative path.

**More Use Cases: Fragment-Based Drug Design.** We also demonstrate ACE on a harder drug design task: fragment linking (Appendix E.2, Example E.3). Generating a molecule $\mathcal{M}$ containing two fragment topologies $\mathcal{T}_1, \mathcal{T}_2$ binding to pocket $\mathcal{P}$ is given by:

$$p_w(\mathcal{M}) \propto p(\mathcal{M}) \left( \frac{p(\mathcal{M}|\mathcal{T}_1, \mathcal{T}_2, \mathcal{P})}{p(\mathcal{M})} \right)^w$$

ACE combines two conformer/de novo experts with an SBDD model (see Eq. E.10 for formal derivations) and applies the bump function to avoid path collapse. We evaluated on non-overlapping fragment pairs sampled from Cross-Docked test set: ACE achieved strong performance (OSR $\geq$ 0.45), remained highly competitive with FFLOM (Jin et al., 2023), a specialized fragment-linking model, and outperformed FKC and NR, which degraded due to path collapse. Full results are in Table E.6.

**Additional Gains Beyond Collapse Repair.** We further evaluate ACE in homogeneous settings on a compositional image generation benchmark in Appendix E.7. Even where path existence holds everywhere, applying time-varying exponents via ACE sharpens intermediate distributions (Proposition 2.1), yielding quantitative gains ($+9.57\%p$ on COCO-MIG) over constant-exponent steering (NR, FKC).

**Scope.** ACE is most useful when a desired target can be expressed as a ratio/product of existing endpoint densities and the experts can be embedded into a common sampling space. It does not repair inaccurate or semantically misaligned experts, nor does it remove the need for compatible output representations. Within this scope, however, ACE addresses a common bottleneck in modular scientific generation: different experts often use different schedules and supports, making naive constant-exponent composition invalid.

**Limitations.** ACE scales linearly with expert count; however, ACE-lite removes gradient overhead, matching standard guidance costs. ACE is a steering correction, not a model correction; if experts are misaligned, ACE strictly enforces a valid path between suboptimal distributions. Our criterion covers Gaussian-based interpolants on compactly supported targets; non-Gaussian priors are future work.

## 5. Related Work

We provide a comprehensive review in Appendix D.

**Inference-time steering.** Diffusion samplers are often steered via ratio-of-densities compositions, most notably classifier-free guidance (CFG) (Dhariwal & Nichol, 2021; Chung et al., 2025). However, standard steering is typically deployed without verifying that the *induced intermediate densities* remain normalizable under the chosen schedules. Feynman–Kac correctors (FKC) (Stoltz et al., 2010; Skreta et al., 2025a; Mark et al., 2025) provide a principled particle interpretation under constant and positive exponent settings, but do not address heterogeneous experts with negative exponents, where we observe *Marginal Path Collapse*.

**Time-dependent guidance.** Time-varying guidance schedules have been explored primarily in *CFG* (homogeneous) for image generation (Wang et al., 2024b; Jin et al., 2026; Zhang & Wan, 2025; Koulischer et al., 2025; Kynkäänniemi et al., 2024), largely as empirical rules to improve sample quality in regimes where path existence is typically not a concern. In contrast, ACE targets *general heterogeneous ratio-of-densities composition* beyond CFG, and derives time-dependent exponents from theory: (i) a *condition for path existence* (Theorem 2.1) and (ii) a *control of quantile radius* (Proposition 2.1).

## 6. Conclusion

We identified *Marginal Path Collapse* as a fundamental failure mode in ratio-of-densities diffusion steering and introduced ACE, a framework that provably resolves it. Our Path Existence Criterion serves as a rigorous diagnostic tool, while ACE generalizes the Feynman–Kac steering framework to support dynamic exponent scheduling, guaranteeing a valid probability path from noise to data. Empirically, ACE eliminates collapse, reduces distributional error by over $4\times$, and enables stable scaffold-based molecular design where existing methods fail. By transforming ratio-of-densities steering from an unstable heuristic into a theoretically grounded methodology, our work establishes the necessary foundations for robust inference-time control. We hope these contributions pave the way for reliable composition of heterogeneous generative models in both creative applications and high-stakes scientific domains.

## Impact Statement

This work introduces a framework for stabilizing heterogeneous model composition, with immediate applications in AI for Science and Structure-Based Drug Design (SBDD). By resolving *Marginal Path Collapse*, ACE enables the reliable use of generative experts for multi-constraint tasks, potentially accelerating the identification of therapeutic candidates. Additionally, our *Path Existence Criterion* improves energy efficiency by preventing computational waste on generative processes that are mathematically destined to fail. Regarding ethics, we acknowledge the dual-use risks inherent in generative chemistry (e.g., designing harmful compounds). However, ACE is a steering algorithm, not a standalone model; it operates strictly within the manifold defined by the pretrained experts. Therefore, its safety profile remains tied to the alignment of the underlying base models. We believe that ensuring mathematical reliability is a prerequisite for building safe and beneficial AI tools.

## Acknowledgments

This work was partly supported by the KHIDI grant funded by the Korean government (MOHW) [No.RS-2025-02307233], the NRF or IITP grants funded by the Korean government (MSIT) [No.RS-2026-25472075, No.RS-2026-25483206, No.RS-2025-02305581, No.RS-2025-25442338 (AI Star Fellowship-SNU), No.RS-2021II211343 (SNU AI), RS-2019-II190075, and No. RS-2023-00209060], the ITIP grant funded by the Korean government (MOTIR) [No.RS-2026-25549946], the Advanced GPU Utilization and AI Computing Infrastructure Enhancement User Support Programs funded by the Korean government (MSIT) [No.05-26-04-0094], the Research grant from SNU, and the Strategic Hub grant for International Research Collaboration of SNU. Kyungsu Kim is affiliated with the School of Transdisciplinary Innovations, Department of Biomedical Science, Interdisciplinary Program in Artificial Intelligence (IPAI), Medical Research Center, and AI Institute at SNU.

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

# Appendix

## Use of Large Language Models

Large language models (LLMs) were used to assist with writing, including grammar, style, and clarity improvements, and for organizational feedback on early drafts. LLMs were not used for generating original scientific content, designing experiments, or analyzing results. All technical contributions, experiments, and conclusions are the work of the authors.

## Reproducibility Statement

To ensure the reproducibility of our results, we have provided the complete source code for ACE, including the implementation of the time-varying exponent scheduler and the Path Existence Criterion, at https://github.com/ziseoklee/ACE. All experiments were conducted using fixed random seeds, which are specified in the configuration files. We have detailed the exact hyperparameters for the baseline methods (CFG, FKC) and our method (bump parameters, guidance scales) in Appendix C. For the molecular benchmarks, we utilized the standard CrossDocked-2020 splits to ensure fair comparison with prior work.

## Compute Resources

All experiments were performed on a computing cluster equipped with NVIDIA RTX 6000 Ada Generation GPUs and A100 GPUs.

## A. Theoretical Foundations of ACE for Heterogeneous Ratio-of-Densities

### A.1. Main Theorems and Proofs

Below we provide the statement and proofs of the main theorem.

---

**Definition A.1** (Ratio-of-Density Probability Path). Fix $t \in [0, 1]$ and let $\mu$ be a $\sigma$-finite reference measure on $\mathbb{R}^d$. For $i = 1, \ldots, n$, let $q_t^{(i)} : \mathbb{R}^{d_i} \to [0, \infty)$ be measurable densities and $\tilde{q}_t^{(i)} = q_t^{(i)} \circ \pi_i$ be their canonical lifts to $\mathbb{R}^d$. We define the **ratio-of-densities path** as the family of functions $\{p_t^*\}_{t \in [0,1]}$ given by:

$$p_t^*(x) := \frac{h_t(x)}{Z_t}, \quad \text{where} \quad h_t(x) = \prod_{i=1}^n \left( \tilde{q}_t^{(i)}(x) \right)^{\gamma_i}, \tag{A.1}$$

provided the following existence conditions are satisfied:

- **Support Inclusion:** Let $I_+ = \{i : \gamma_i > 0\}$ and $I_- = \{i : \gamma_i < 0\}$. Then $\mathrm{supp}(\prod_{i \in I_+} \tilde{q}_t^{(i)}) \subseteq \mathrm{supp}(\prod_{i \in I_-} \tilde{q}_t^{(i)}) =: \Omega_t$, preventing singularities on the boundary $\partial \Omega_t$.

- **Integrability:** The unnormalized product $h_t$ belongs to $L^1(\mu)$ with $0 < Z_t < \infty$.

---

*Remark.* Measurability of $h_t$ is guaranteed as the product of compositions of measurable functions with continuous projections. The integrability condition ensures $p_t^*$ is a valid Radon-Nikodym derivative $d\mathbb{P}_t^*/d\mu$.

---

**Proposition A.1** (Expectation identity under Feynman–Kac dynamics). *Let $(p_t)_{t \in [0,T]}$ be a family of probability densities on $\mathbb{R}^d$ evolving according to the weighted Feynman–Kac PDE*

$$\frac{\partial}{\partial t} p_t(x) = -\nabla \cdot \left( p_t(x) \, v_t(x) \right) + \frac{\sigma_t^2}{2} \Delta p_t(x) + p_t(x) \left( g_t(x) - \int g_t(y) \, p_t(y) \, dy \right), \tag{A.2}$$

*where $v_t : \mathbb{R}^d \to \mathbb{R}^d$ is a drift field, $\sigma_t > 0$ a scalar diffusion coefficient, and $g_t : \mathbb{R}^d \to \mathbb{R}$ a measurable weight function.*

---

*Consider the diffusion process*

$$dx_t = v_t(x_t)\, dt + \sigma_t\, dW_t, \qquad x_0 \sim p_0,$$

*driven by a standard Brownian motion $(W_t)_{t \geq 0}$. Define the weight process*

$$w_T \;=\; \exp\!\left( \int_0^T g_s(x_s)\, ds \right).$$

*Then for any bounded test function $\phi : \mathbb{R}^d \to \mathbb{R}$,*

$$\mathbb{E}_{p_T}[\phi(x_T)] \;=\; \frac{1}{Z_T}\, \mathbb{E}[\, w_T\, \phi(x_T)\,], \qquad\qquad \text{(A.3)}$$

*where the expectation on the right is with respect to the law of the process $(x_t)_{t \in [0,T]}$, and $Z_T > 0$ is a normalizing constant for $w_T$ independent of $x_T$.*

*Proof.* The proof can be found in **Proposition A.1.** of (Skreta et al., 2025a) or Section 4 of (Stoltz et al., 2010). $\qquad\square$

---

**Theorem A.1** (Adaptive Path Correction with time-dependent Exponents (ACE))**.** *For each $i \in \{1, \ldots, n\}$, let*

$$\gamma_i(t) : \mathbb{R} \to \mathbb{R}$$

*be differentiable with respect to $t$ and $\{q_t^{(i)}\}_{t \in [0,1]}$ denote probability densities in $\mathbb{R}^{d_i}$ with respect to the Lebesgue measure. Suppose each probability path $\{q_t^{(i)}\}_{t \in [0,1]}$ is associated with the stochastic differential equation (SDE):*

$$X_0^{(i)} \sim q_0^{(i)}, \qquad dX_t^{(i)} = \left( v_t^{(i)}(X_t^{(i)}) + \tfrac{\sigma_t^{(i)2}}{2} \nabla \log q_t^{(i)}(X_t^{(i)}) \right) dt + \sigma_t^{(i)} dW_t^{(i)} \qquad\qquad \text{(A.4)}$$

*or, equivalently, the ordinary differential equation (ODE):*

$$X_0^{(i)} \sim q_0^{(i)}, \qquad dX_t^{(i)} = v_t^{(i)}(X_t^{(i)}) dt \qquad\qquad \text{(A.5)}$$

*where $W_t^{(i)}$ is a Wiener process in $\mathbb{R}^{d_i}$, $s_t^{(i)}(x^{(i)}) := \nabla_{x^{(i)}} \log q_t^{(i)}(x^{(i)})$, $x^{(i)} \in \mathbb{R}^{d_i}$ is the score function, and the fields $v_t^{(i)}$, $s_t^{(i)} \in \mathcal{C}^1$ are measurable.*

*Let $d = \max_i d_i$, $\pi_i : \mathbb{R}^d \to \mathbb{R}^{d_i}$ be a linear projection map, and $\iota_i : \mathbb{R}^{d_i} \to \mathbb{R}^d$ be a linear embedding map such that $\pi_i \circ \iota_i = \mathrm{Id}_i$. For a vector field $f^{(i)} : \mathbb{R}^{d_i} \to \mathbb{R}^{d_i}$, define its canonical extension $\tilde{f}^{(i)} : \mathbb{R}^d \to \mathbb{R}^d$ by $\tilde{f}^{(i)} = \iota_i \circ f^{(i)} \circ \pi_i$. For $q_t^{(i)} : \mathbb{R}^{d_i} \to [0, \infty)$, define the canonical lift $\tilde{q}_t^{(i)} : \mathbb{R}^d \to [0, \infty)$ by $\tilde{q}_t^{(i)} = q_t^{(i)} \circ \pi_i$.*

*If the assumptions of Definition A.1 hold, then, for any differentiable vector field $v_t^* : \mathbb{R}^d \to \mathbb{R}^d$, the stochastic process given by the following weighted SDE/ODE with $s_t^*(x) := \sum_{i=1}^n \gamma_i(t)\tilde{s}_t^{(i)}(x)$,*

$$X_0 \sim p_0^*, w_0 = \mathbf{1}, \log q_0^{(i)}(X_0) : \text{ standard gaussian density} \qquad\qquad \text{(A.6)}$$

$$dX_t = \left( v_t^*(X_t) + \tfrac{\sigma_t^2}{2} s_t^*(X_t) \right) dt + \sigma_t dW_t \quad or \quad dX_t = v_t^*(X_t) dt \qquad\qquad \text{(A.7)}$$

$$d \log \tilde{q}_t^{(i)}(X_t) = \left( -\nabla \cdot \tilde{v}_t^{(i)} + (v_t^* - \tilde{v}_t^{(i)}) \cdot \tilde{s}_t^{(i)} + \frac{\sigma_t^2}{2} \left( s_t^* \cdot \tilde{s}_t^{(i)} + \nabla \cdot \tilde{s}_t^{(i)} \right) \right) dt + \sigma_t \tilde{s}_t^{(i)} \cdot dW_t \qquad \text{(A.8)}$$

$$or \qquad\qquad \text{(A.9)}$$

$$d \log \tilde{q}_t^{(i)}(X_t) = \left[ -\nabla \cdot \tilde{v}_t^{(i)}(X_t) + (v_t^*(X_t) - \tilde{v}_t^{(i)}(X_t)) \cdot \tilde{s}_t^{(i)}(X_t) \right] dt \qquad\qquad \text{(A.10)}$$

$$\text{(A.11)}$$

$$d \log w_t(X_t) = \left[ \nabla \cdot v_t^*(X_t) + \sum_{i=1}^{n} \dot{\gamma}_i(t) \log \tilde{q}_t^{(i)}(X_t) \right. \tag{A.12}$$

$$\left. + \sum_{i=1}^{n} \gamma_i(t) \left( -\nabla \cdot \tilde{v}_t^{(i)}(X_t) + \left( v_t^*(X_t) - \tilde{v}_t^{(i)}(X_t) \right) \cdot \tilde{s}_t^{(i)}(X_t) \right) \right] dt \tag{A.13}$$

*follows the probability path* $p_t^*(x) = \frac{1}{Z_t} \prod_{i=1}^{n} \left( \tilde{q}_t^{(i)}(x) \right)^{\gamma_i(t)}$, $x \in \Omega_t$ *where* $t \in [0,1]$, $\Omega_t := \text{supp}(p_t^*)$ *and* $Z_t = \int_{\Omega_t} \prod_{i=1}^{n} \left( \tilde{q}_t^{(i)}(x) \right)^{\gamma_i(t)} dx$ *is the normalizing constant only dependent on* $t$.

*Proof.* From Equation (A.5) (or, equivalently, Equation (A.4)), each probability path $\{q_t^{(i)}\}_{t \in [0,1]}$ solves the Fokker-Planck PDE given by $\partial_t q_t^{(i)} = -\nabla \cdot (v_t^{(i)} q_t^{(i)})$. Dividing both sides by $q_t^{(i)}$, we get,

$$\partial_t \log q_t^{(i)} = -\nabla \cdot v_t^{(i)} - v_t^{(i)} \cdot s_t^{(i)} \tag{A.14}$$

By the Leibniz product rule,

$$\frac{1}{Z_t} \partial_t \prod_{i=1}^{n} \left( \tilde{q}_t^{(i)} \right)^{\gamma_i(t)} = \frac{1}{Z_t} \partial_t \prod_{i=1}^{n} \exp \left( \gamma_i(t) \log \tilde{q}_t^{(i)} \right) = p_t^* \sum_{i=1}^{n} \left( \dot{\gamma}_i(t) \log \tilde{q}_t^{(i)} + \gamma_i(t) \partial_t \log \tilde{q}_t^{(i)} \right) \tag{A.15}$$

We derive the unified Feynman–Kac PDE for the heterogeneous product $\{p_t^*\}_{t \in [0,1]}$ by the following:

$$\partial_t \log p_t^* = \sum_{i=1}^{n} \left( \dot{\gamma}_i(t) \log \tilde{q}_t^{(i)} + \gamma_i \partial_t \log \tilde{q}_t^{(i)} \right) - \partial_t \log Z_t \tag{A.16}$$

$$\overset{(A.15)}{=} \sum_{i=1}^{n} \left( \dot{\gamma}_i(t) \log \tilde{q}_t^{(i)} + \gamma_i \partial_t \log \tilde{q}_t^{(i)} \right) - \int_{\Omega} p_t^* \sum_{i=1}^{n} \left( \dot{\gamma}_i(t) \log \tilde{q}_t^{(i)} + \gamma_i(t) \partial_t \log \tilde{q}_t^{(i)} \right) dx \tag{A.17}$$

$$= -\nabla \cdot v_t^* - v_t^* \cdot s_t^* \tag{A.18}$$

$$+ \underbrace{\nabla \cdot v_t^* + v_t^* \cdot s_t^* + \sum_{i=1}^{n} \left( \dot{\gamma}_i(t) \log \tilde{q}_t^{(i)} + \gamma_i \partial_t \log \tilde{q}_t^{(i)} \right)}_{=: g_t} \tag{A.19}$$

$$- \mathbb{E}_{p_t^*} \left[ \sum_{i=1}^{n} \left( \dot{\gamma}_i(t) \log \tilde{q}_t^{(i)} + \gamma_i(t) \partial_t \log \tilde{q}_t^{(i)} \right) \right] \tag{A.20}$$

$$= -\nabla \cdot v_t^* - v_t^* \cdot s_t^* + g_t - \mathbb{E}_{p_t^*}[g_t] \tag{A.21}$$

where we used the boundary assumption from Definition A.1 and the divergence theorem:

$$\mathbb{E}_{p_t^*}[\nabla \cdot v_t^* + v_t^* \cdot s_t^*] = \int_{\Omega_t} (\nabla \cdot v_t^*(x) + v_t^*(x) \cdot s_t^*(x)) p_t^*(x) dx \tag{A.22}$$

$$= \int_{\Omega_t} \nabla \cdot (v_t^*(x) p_t^*(x)) dx = \int_{\partial \Omega_t} (v_t^*(x) \underbrace{p_t^*(x)}_{0 \text{ on } \partial \Omega_t}) \cdot d\mathbf{S} = 0 \tag{A.23}$$

Multiplying $p_t^*$ on both sides of Equation (A.21) yields:

$$\partial_t p_t^* = -\nabla \cdot (v_t^* p_t^*) + p_t^* \left( g_t - \mathbb{E}_{p_t^*}[g_t] \right) \tag{A.24}$$

where

$$g_t = \nabla \cdot v_t^* + v_t^* \cdot s_t^* + \sum_{i=1}^{n} \left( \dot{\gamma}_i(t) \log \tilde{q}_t^{(i)} + \gamma_i \partial_t \log \tilde{q}_t^{(i)} \right) \tag{A.25}$$

$$= \nabla \cdot v_t^* + v_t^* \cdot s_t^* + \sum_{i=1}^{n} \left( \dot{\gamma}_i(t) \log \tilde{q}_t^{(i)} + \gamma_i (-\nabla \cdot v_t^{(i)} - v_t^{(i)} \cdot s_t^{(i)}) \right) \tag{A.26}$$

$$= \nabla \cdot v_t^* + \sum_{i=1}^{n} \dot{\gamma}_i(t) \log \tilde{q}_t^{(i)} + \sum_{i=1}^{n} \gamma_i \left( -\nabla \cdot \tilde{v}_t^{(i)} + (v_t^* - \tilde{v}_t^{(i)}) \cdot \tilde{s}_t^{(i)} \right) \tag{A.27}$$

We can obtain the value of $\log \tilde{q}_t^{(i)}(X_t)$ by simulating an ODE (if $X_t$ follows an ODE) or an SDE (if $X_t$ follows an SDE).

$$\partial_t \log \tilde{q}_t^{(i)} = -\nabla \cdot \tilde{v}_t^{(i)} - \tilde{v}_t^{(i)} \cdot \tilde{s}_t^{(i)} \tag{A.28}$$

**ODE case:** $dX_t = v_t^*(X_t)dt$

$$\frac{d}{dt} \log \tilde{q}_t^{(i)}(X_t) = \partial_t \log \tilde{q}_t^{(i)}(x = X_t) + \nabla \log \tilde{q}_t^{(i)}(x = X_t) \cdot \frac{dX_t}{dt} \tag{A.29}$$

$$= -\nabla \cdot \tilde{v}_t^{(i)}(X_t) + (v_t^*(X_t) - \tilde{v}_t^{(i)}(X_t)) \cdot \tilde{s}_t^{(i)}(X_t) \tag{A.30}$$

**SDE case:** $dX_t = \left( v_t^*(X_t) + \frac{\sigma_t^2}{2} s_t^*(X_t) \right) dt + \sigma_t dW_t$

For the SDE case, we must use Itô's formula to find the differential $d \log \tilde{q}_t^{(i)}(X_t)$. Given the function $f(t, x)$ and the SDE for $X_t$ above, Itó's formula states that

$$df(t, X_t) = \left( \partial_t f + \mu_t \cdot \nabla_X f + \frac{1}{2} \text{Tr}(\Sigma_t^\intercal \nabla_X^2 f \Sigma_t) \right) dt + (\nabla_X f)^\intercal \Sigma_t dW_t,$$

where $f(t, X_t) = \log \tilde{q}_t^{(i)}(X_t)$, $\mu_t = v_t^* + \frac{\sigma_t^2}{2} s_t^*$, and $\Sigma_t = \sigma_t I$.

Applying that $\nabla_X f = \nabla_X \log \tilde{q}_t^{(i)} = \tilde{s}_t^{(i)}$, $\frac{1}{2} \text{Tr}(\Sigma_t^\intercal \nabla_X^2 f \Sigma_t) = \frac{\sigma_t^2}{2} \Delta_X f = \frac{\sigma_t^2}{2} \nabla \cdot \tilde{s}_t^{(i)}$:

$$d \log \tilde{q}_t^{(i)}(X_t) = \left( -\nabla \cdot \tilde{v}_t^{(i)} + (v_t^* - \tilde{v}_t^{(i)}) \cdot \tilde{s}_t^{(i)} + \frac{\sigma_t^2}{2} \left( s_t^* \cdot \tilde{s}_t^{(i)} + \nabla \cdot \tilde{s}_t^{(i)} \right) \right) dt + \sigma_t \tilde{s}_t^{(i)} \cdot dW_t \tag{A.31}$$

By Proposition A.1, we can sample from $p_t^*$ by simulating the following weighted SDE or ODE starting from $X_0 \sim p_0^*, w_0 = 1$:

$$\text{SDE:} \quad dX_t = \left( v_t^*(X_t) + \frac{\sigma_t^2}{2} s_t^*(X_t) \right) dt + \sigma_t dW_t, \qquad d \log w_t = g_t(X_t) dt \tag{A.32}$$

$$\text{ODE:} \quad dX_t = v_t^*(X_t) dt, \qquad\qquad\qquad\qquad\quad d \log w_t = g_t(X_t) dt \tag{A.33}$$

Our result extends, subsumes, and unifies prior formulations ((Skreta et al., 2025a; Mark et al., 2025)) in the literature. □

*Remark.* **Interpretation.** The auxiliary weight process $(w_t)$ plays the role of a *likelihood ratio corrector*: it accounts for the discrepancy between the law induced by the forward dynamics $(X_t)$ and the target density $p_t^*$. In other words, although the marginal of $X_t$ alone may not coincide with $p_t^*$, the pair $(X_t, w_t)$ ensures unbiased recovery of expectations under $p_t^*$ via Proposition A.1. This extends and unifies earlier formulations of weighted Feynman–Kac dynamics in the literature (Skreta et al., 2025a; Mark et al., 2025).

*Remark.* **Practical simulation.** From an algorithmic perspective, the weighted SDE/ODE requires no additional training or architecture-specific modifications. Practitioners only need to simulate sample paths $(X_t)$ according to the chosen dynamics and accumulate weights via the exponential update $d \log w_t = g_t(X_t) \, dt$. In practice, this can be carried out efficiently with Sequential Monte Carlo (SMC) or particle filtering methods, where the weights $w_t$ play the usual role of importance weights.

Although the heterogeneous Feynman–Kac framework (Theorem A.1) may appear abstract, its input–output structure is remarkably simple: given the forward dynamics and score functions $\{v_t^{(i)}, s_t^{(i)}\}_i$, one can simulate particles $(X_t, w_t)$ and obtain unbiased estimators for expectations under $p_t^*$. This makes the method broadly applicable without architectural constraints such as attention-map control or model-specific fine-tuning.

### A.2. Algorithmic Formulations and Comparison: FKC vs. ACE

Here, we present the algorithmic tables for FKC and ACE. As the tables indicate, FKC arises as a special case of ACE when the gamma schedule is constant. Main algorithmic differences are highlighted in blue. This results in an extra update for $\log q$ components, which is needed because $\dot{\gamma}_i(t) \neq 0$.

---

**Algorithm 1** Adaptive Correction with Exponents (ACE, Ours)

---

**Require:**
- Batch size $N$, initial particles $X_0^j \sim p_0^*$, weights $w_0^j = 1/N$
- Networks: scores $s_{\theta_i}^{(i)}$ and velocities $v_{\theta_i}^{(i)}$ for the paths $q_t^{(i)}$
- Projection $\pi_i$, embedding $\iota_i$, **time-varying exponents** $\gamma_i(t)$
- Base drift $v_\phi^*$, noise schedule $\sigma_t$, steps $T$, resampling threshold $\tau$

1: $\Delta t \leftarrow 1/T$
2: **for** $t = 0, \Delta t, \ldots, 1 - \Delta t$ **do**
3:     **Mixture score:** $s_t^*(x) = \sum_{i=1}^n \gamma_i(t)\, \tilde{s}_t^{(i)}(x)$ with $\tilde{s}_t^{(i)}(x) = \iota_i(s_{\theta_i}^{(i)}(\pi_i(x), t))$.
4:     **Drift:** $\mu_t(x) = v_\phi^*(x, t) + \frac{\sigma_t^2}{2} s_t^*(x)$
5:     **Component drift correction** with $\tilde{v}_t^{(i)}(x) = \iota_i(v_{\theta_i}^{(i)}(\pi_i(x), t))$:

$$D_t^{(i)}(x) = -\nabla \cdot \tilde{v}_t^{(i)}(x) + (v_\phi^*(x, t) - \tilde{v}_t^{(i)}(x)) \cdot \tilde{s}_t^{(i)}(x),$$

6:     **for** $j = 1, \ldots, N$ **do**
7:         **Propagate particle:** $X_{t+\Delta t}^j = X_t^j + \mu_t(X_t^j)\Delta t + \sigma_t \sqrt{\Delta t}\, \xi_t^j, \qquad \xi_t^j \sim \mathcal{N}(0, I)$
8:         **Update log-components:** $\log q_{t+\Delta t}^{(i),j} = \log q_t^{(i),j} + \Delta \log q_t^{(i),j}\, \Delta t$ with

$$\Delta \log q_t^{(i),j} = D_t^{(i)}(X_t^j) + \frac{\sigma_t^2}{2}\left(s_t^* \cdot \tilde{s}_t^{(i)} + \nabla \cdot \tilde{s}_t^{(i)}\right) + \sigma_t\, \tilde{s}_t^{(i)} \cdot \xi_t^j (1/\sqrt{\Delta t})$$

9:         **Update weight:** $\log w_{t+\Delta t}^j = \log w_t^j + \Delta \log w_t^j\, \Delta t$ with

$$\Delta \log w_t^j = \nabla \cdot v_\phi^*(X_t^j, t) + \sum_{i=1}^n \dot{\gamma}_i(t)\, \log q_t^{(i),j} + \sum_{i=1}^n \gamma_i(t)\, D_t^{(i)}(X_t^j)$$

10:     **end for**
11:     Compute Effective Sample Size (ESS) $= \dfrac{(\sum_j w_{t+\Delta t}^j)^2}{\sum_j (w_{t+\Delta t}^j)^2}$
12:     **if** ESS $< \tau N$ **then**
13:         Resample particles according to $\{\frac{w_{t+\Delta t}^j}{\sum_j w_{t+\Delta t}^j}\}$
14:         Reset $w_{t+\Delta t}^j = 1/N$
15:     **end if**
16: **end for**

---

---

**Algorithm 2** Feynman–Kac Corrector (FKC, (Skreta et al., 2025a))

---

**Require:**
- Batch size $N$, initial particles $X_0^j \sim p_0^*$, weights $w_0^j = 1/N$
- Pretrained component networks: scores $s_{\theta_i}^{(i)}$ and velocities $v_{\theta_i}^{(i)}$ for $q_t^{(i)}$
- Projections $\pi_i$, embeddings $\iota_i$, constant exponents $\gamma_i$
- Base drift $v_\phi^*(x,t)$, noise schedule $\sigma_t$, total steps $T$, resampling threshold $\tau$

1: $\Delta t \leftarrow 1/T$
2: **for** $t = 0, \Delta t, \ldots, 1 - \Delta t$ **do**
3:      **Mixture score:** $s_t^*(x) = \sum_{i=1}^n \gamma_i \, \tilde{s}_t^{(i)}(x)$ with $\tilde{s}_t^{(i)}(x) = \iota_i(s_{\theta_i}^{(i)}(\pi_i(x), t))$.
4:      **Drift:** $\mu_t(x) = v_\phi^*(x,t) + \frac{\sigma_t^2}{2} s_t^*(x)$
5:      **Component drift correction** with $\tilde{v}_t^{(i)}(x) = \iota_i(v_{\theta_i}^{(i)}(\pi_i(x), t))$:

$$D_t^{(i)}(x) = -\nabla \cdot \tilde{v}_t^{(i)}(x) + (v_\phi^*(x,t) - \tilde{v}_t^{(i)}(x)) \cdot \tilde{s}_t^{(i)}(x),$$

6:      **for** $j = 1, \ldots, N$ **do**
7:          **Propagate particle:** $X_{t+\Delta t}^j = X_t^j + \mu_t(X_t^j)\Delta t + \sigma_t \sqrt{\Delta t}\, \xi_t^j, \qquad \xi_t^j \sim \mathcal{N}(0, I)$
8:          **Update weight:** $\log w_{t+\Delta t}^j = \log w_t^j + \Delta \log w_t^j \, \Delta t$ with

$$\Delta \log w_t^j = \nabla \cdot v_\phi^*(X_t^j, t) + \sum_{i=1}^n \gamma_i \, D_t^{(i)}(X_t^j)$$

9:      **end for**
10:     Compute Effective Sample Size (ESS) $= \dfrac{(\sum_j w_{t+\Delta t}^j)^2}{\sum_j (w_{t+\Delta t}^j)^2}$
11:     **if** ESS $< \tau N$ **then**
12:         Resample particles according to $\{\frac{w_{t+\Delta t}^j}{\sum_j w_{t+\Delta t}^j}\}$
13:         Reset $w_{t+\Delta t}^j = 1/N$
14:     **end if**
15: **end for**

---

*Remark.* **Remark on computability vs. validity.** The FKC algorithm remains numerically computable even when Marginal Path Collapse occurs: the mixed score $s_t^*$ and the update rules in Algorithm 2 produce finite values at every step. However, this computability does *not* imply that the algorithm is sampling from a valid probability model. FKC is theoretically justified only when the target path $p_t^*(x) \propto h_t(x)$ exists as a family of *normalizable* densities for all $t$ on the discretization grid. If for some $t^*$ the ratio-of-densities integrand $h_{t^*}$ fails to lie in $L^1$ (i.e. $Z_{t^*} = \int h_{t^*} = \infty$), then $p_{t^*}^*$ does not exist, and the drift field $s_{t^*}^*$ used by FKC is no longer the score of any probability density. As a result, the reverse SDE/ODE and the weighted Feynman–Kac dynamics no longer transport the intended ratio-of-densities path $\{p_t^*\}$: instead they follow a different density path $\{p_t'\}$ determined by their coefficients. **Even though the sampler remains numerically well-defined, its terminal law $p_t'$ is no longer equal to the desired target $p_1^*$.**

Under standard regularity assumptions on the drift and diffusion, the Fokker–Planck equation associated with a given SDE has a unique weak solution for each initial probability density. Therefore, if there exists $t_c$ with $h_{t_c} \notin L^1$ (Marginal Path Collapse), there is no probability path $\{p_t^*\}$ that both (i) coincides with $h_t/Z_t$ whenever $h_t \in L^1$ and (ii) solves the same Fokker–Planck equation globally. The path produced by FKC, $\{p_t'\}$, is hence a different solution induced by its own initial condition and cannot coincide with the intended ratio-of-densities path.

Empirically, in every heterogeneous setting we tested, violation of the path-existence criterion (Theorem 2.1) leads FKC to complete failure (typically through unstable or degenerate importance weights) despite the algorithm itself producing finite updates. This behavior is exactly the Marginal Path Collapse phenomenon. Our path-existence criterion guarantees $Z_t < \infty$ when $C(t) > 0$ and detects $Z_t = \infty$ when $C(t) < 0$; Theorem 2.2 then shows how ACE constructs corrected exponent schedules ensuring that the entire path $\{p_t^*\}_{t \in [0,1]}$ is well-defined, allowing the weighted SDE established in Theorem 2.3 and Algorithm 1 to provide unbiased samples from the desired target.

**Example A.1** (Finite score fields do not imply valid steering). Consider a 4-expert Gaussian path (from Figure 2). Let $h_t(x) = \frac{q_t^1(x)q_t^2(x)}{q_t^3(x)q_t^4(x)}$ with $q_t^{(1)} \sim N(0, \sigma_1^2(t))$, $q_t^{(2)} \sim N(0, \sigma_2^2(t))$, and $q_t^{(3)} = q_t^{(4)} \sim N(0, \sigma_3^2(t))$.

- **The Collapse:** At $t = 0.5$, $C(0.5) = -1/30$ drops below zero, making $h_{0.5}(x) \propto \exp(x^2/60)$ non-integrable.

- **The Paradox:** The mixed score $s_{\text{mix}}(x, t) = -C(t)x$ remains perfectly finite ($x/30$ at $t = 0.5$). The SDE is well-posed and does not crash.

- **The Detachment:** By standard Fokker-Planck arguments, any Gaussian path $p_t = N(0, \sigma_t^2 I)$ can be simulated by the SDE:

$$dX_t = \left(\frac{g_t^2}{2} - \sigma_t \dot{\sigma}_t\right) s(X_t, t)dt + g_t dW_t$$

We can run the well-posed SDE with the **mixed score** by choosing a reference schedule $\sigma_1(t)$ and constant noise $g_t = \sqrt{2}$ (though the following structural failure applies universally to any choice of reference schedule and $g_t$):

$$dX_t = (1 - \sigma_1(t)\dot{\sigma}_1(t))s_{\text{mix}}(X_t, t)dt + \sqrt{2}dW_t$$

Plugging in the mixed score (linear in $x$) gives an induced SDE of the form:

$$dX_t = -a(t)C(t)X_t dt + g_t dW_t$$

Taking expectations gives an ODE for $m(t) := \mathbb{E}[X_t]$:

$$\dot{m}(t) = -a(t)C(t)m(t), \quad m(0) = 0$$

Hence, $m(t) = 0$ for all $t$. Therefore, $V(t) := \mathbb{E}[X_t^2] = \text{Var}(X_t)$. Applying Itô's formula,

$$d(X_t^2) = [-2a(t)C(t)X_t^2 + g_t^2]dt + 2g_t X_t dW_t$$

and taking expectations yields:

$$\dot{V}(t) = -2a(t)C(t)V(t) + g_t^2$$

For the concrete choice $g_t = \sqrt{2}$ and $2a(t) = 4 - 3t$, this reduces to the ODE:

$$\dot{V}(t) = (3t - 4)C(t)V(t) + 2$$

To succeed, $V(t)$ must track $1/C(t)$ to reach $V(1) = 7$. However, since $V(t) \geq 0$, $V(t)$ cannot become negative. Therefore, once $C(t) < 0$ the sampler cannot remain on the intended path: the target variance $1/C(t)$ is negative and hence not realizable by any probability law. In our concrete well-posed mixed-score SDE above, numerical integration from $V(0) = 1.5$ yields $V(1) \approx 2.415 \neq 7$, so the terminal marginal is also wrong. More generally, any later return to the correct endpoint would require specially tuned off-path dynamics rather than automatic recovery.

## B. Integrability Preservation Along Stochastic Paths

The problem of preserving integrability for ratios of products of densities along stochastic paths is non-trivial. While general ratios can exhibit pathological behavior where integrability is lost, imposing a structural condition of component-wise dominance for GMMs, or positivity of a simple criterion for compactly supported densities, is sufficient to prevent such failures. This section demonstrates that under these conditions, integrability, once established at endpoints, is maintained throughout the evolution.

### B.1. Mathematical Preliminaries

We state mathematical preliminaries.

**Definition B.1** (Generative Stochastic Path for Stochastic Interpolants). Let $X_0 \sim \mathcal{N}(0, \mathbf{I})$ and $X_1 \sim p_1(x)$ be independent random variables. A **generative stochastic path** is a time-indexed random variable $X_t$ for $t \in [0, 1]$ defined by the sample-wise interpolation:

$$X_t = \alpha_t X_0 + \beta_t X_1 \tag{B.1}$$

where $\alpha_t, \beta_t$ are non-negative, differentiable functions of $t$ satisfying the boundary conditions $\alpha_0 = 1, \beta_0 = 0$ and $\alpha_1 = 0, \beta_1 = 1$.

*Remark* (Ornstein-Uhlenbeck Process and Flow Matching as Special Cases of Stochastic Interpolants). The generalized path encompasses the two most common paths in generative modeling.

- **Flow Matching (Linear Path):** Setting $\alpha_t = 1 - t$ and $\beta_t = t$ gives the linear interpolation path $X_t = (1-t)X_0 + tX_1$.

- **Diffusion Models (OU-like Path):** Setting $\alpha_t = e^{-\int_0^t \gamma(s)ds}$ and $\beta_t = \sqrt{1 - e^{-2\int_0^t \gamma(s)ds}}$ corresponds to the path generated by an Ornstein-Uhlenbeck process, commonly used in diffusion models.

Our results hold for the general path, which includes these specific cases.

**Lemma B.1** (The relationship between the velocity and score). *Suppose $v_t(x)$ is a locally Lipschitz vector field which generates the probability path $p_t$ between $p_0 \sim \mathcal{N}(0, I)$ and $p_1$ with differentiable schedules $\alpha_t, \beta_t$ such that $X_t = \alpha_t X_0 + \beta_t X_1 \sim p_t$. Then the score can be expressed as a function of the velocity field by:*

$$\nabla \log p_t(x) = \frac{1}{\alpha_t \left( \frac{\dot{\beta}_t}{\beta_t} \alpha_t - \dot{\alpha}_t \right)} \left( v_t(x) - \frac{\dot{\beta}_t}{\beta_t} x \right) \tag{B.2}$$

*Specifically, for $\alpha_t = 1 - t, \beta_t = t$ (Flow Matching), the score can be expressed as*

$$\nabla \log p_t(x) = \frac{t v_t(x) - x}{1 - t} \tag{B.3}$$

*Proof.* The proof can be found in **B.4** of (Domingo-Enrich et al., 2024). □

**Lemma B.2** (Sum of Independent Random Variables). *Let $A$ and $B$ be two independent random variables in $\mathbb{R}^d$ with probability density functions $p_A(a)$ and $p_B(b)$, respectively. The probability density function of their sum, $C = A + B$, is given by the convolution of their individual PDFs:*

$$p_C(c) = (p_A * p_B)(c) = \int_{\mathbb{R}^d} p_A(y) p_B(c - y) dy \tag{B.4}$$

*Proof.* The proof can be found in Chapter 6 of (Ross, 2020). □

**Proposition B.1** (Distribution Along the Path of the Stochastic Interpolant). *The probability density function $p_t(x)$ of the random variable $X_t = \alpha_t X_0 + \beta_t X_1$ is given by the convolution of the scaled final density with a Gaussian kernel:*

$$p_t(x) = \left( \frac{1}{\beta_t^d} p_1 \left( \frac{\cdot}{\beta_t} \right) \right) * \mathcal{N}(x|\mathbf{0}, \alpha_t^2 \mathbf{I}) \tag{B.5}$$

*Proof.* $X_t$ is the sum of two independent random variables: $A = \alpha_t X_0$ and $B = \beta_t X_1$. The density of $A$ is $\mathcal{N}(x|\mathbf{0}, \alpha_t^2 \mathbf{I})$. The density of $B$ is $\frac{1}{\beta_t^d} p_1(\frac{x}{\beta_t})$. By Lemma B.2, the density of their sum $X_t$ is the convolution of their respective densities. □

**Lemma B.3** (Integrability of Exponential Functions with Quadratic Exponents). *Let*

$$f(x) = \exp\left(-\tfrac{1}{2}x^\top A x + b^\top x + c\right)$$

*be an exponential function with a quadratic exponent, where $A \in \mathbb{R}^{d \times d}$ is symmetric. Then $f \in L^1(\mathbb{R}^d)$ if and only if $A \succ 0$.*

*Proof.* ($\Rightarrow$) If $A \succ 0$, then $-\tfrac{1}{2}x^\top A x$ dominates the linear term $b^\top x$, so $f(x)$ is bounded by a Gaussian density and is therefore integrable over $\mathbb{R}^d$. ($\Leftarrow$) Suppose $A \not\succ 0$. Then there exists a unit eigenvector $v$ of $A$ with eigenvalue $\lambda \le 0$.

Decompose $x = tv + y$ with $y \perp v$; in this orthogonal basis the exponent becomes

$$-\tfrac{1}{2}\lambda t^2 + b_1 t \; - \; \tfrac{1}{2}y^\top A_\perp y + b_\perp^\top y.$$

Consider integrating $f$ over the unbounded tube

$$\mathcal{C}_r = \{(t, y) : \|y\| \le r\}, \qquad r > 0.$$

If $\lambda < 0$, the term $-\tfrac{1}{2}\lambda t^2$ grows *positively* and the integral diverges. If $\lambda = 0$, then the exponent along the $t$-direction is at most linear: If $b_1 \ne 0$, the integrand grows (or decays too slowly) linearly; if $b_1 = 0$, the integrand has no decay along $t$. In either case the integral over $\mathcal{C}_r$ diverges. Hence $f \notin L^1(\mathbb{R}^d)$ whenever $A$ is not positive definite. $\square$

## B.2. Compactly Supported Distributions

We consider compactly supported densities, a condition satisfied by the vast majority of real-world data. Examples include:

- **Images/Videos**: Pixel values are bounded, typically in a hypercube like $[0,1]^{H \times W \times C}$. Videos are a sequence of images spread across the time dimension $T$, bounded in a hypercube of even higher dimension, like $[0,1]^{H \times W \times C \times T}$.

- **3D Molecules/Shapes**: Atomic coordinates are constrained within a finite volume centered at the center of mass.

**Proposition B.2** (Isotropic Gaussian to Compactly Supported Target Density). *Let $p_0(x) = \mathcal{N}(x; 0, \sigma_0^2 I)$ and let $p_1$ be a probability density with compact support in $\mathbb{R}^d$. Let $\{p_t\}_{t \in [0,1]}$ be the probability path generated by stochastic interpolant $X_t = \alpha_t X_0 + \beta_t X_1$. Then, for any $t \in [0,1)$, there exist finite positive constants $0 < C_\pm, \mu_t, V_t, R_t < \infty$ such that $p_t(x)$ is bounded by Gaussian envelopes:*

$$C_- \exp\left(\frac{-\|x - \mu_t\|^2 + \|\mu_t\|^2 - V_t}{2\alpha_t^2 \sigma_0^2}\right) \le p_t(x) \le C_+ \exp\left(\frac{-(\|x\| - R_t)^2 + R_t^2}{2\alpha_t^2 \sigma_0^2}\right)$$

*Proof.* Let $\mathcal{N}_t(x) = \mathcal{N}(x; 0, \sigma_t^2 I)$ be the density of $\alpha_t X_0$, where $\sigma_t^2 = \alpha_t^2 \sigma_0^2$. Let $\rho_t(y)$ be the density of $Y_t = \beta_t X_1$. Since $p_1$ is compactly supported, $\rho_t$ is supported on a compact set $\Omega_t$. The path density is the convolution $p_t = \mathcal{N}_t * \rho_t$. We analyze the ratio $p_t(x)/\mathcal{N}_t(x)$:

$$\frac{p_t(x)}{\mathcal{N}_t(x)} = \int_{\Omega_t} \frac{\mathcal{N}_t(x-y)}{\mathcal{N}_t(x)} \rho_t(y) dy = \int_{\Omega_t} \exp\left(\frac{2x \cdot y - \|y\|^2}{2\sigma_t^2}\right) \rho_t(y) dy = \mathbb{E}_{Y \sim \rho_t}\left[\exp\left(\frac{2x \cdot Y - \|Y\|^2}{2\sigma_t^2}\right)\right].$$

**Upper Bound:** Let $R_t = \sup_{y \in \Omega_t} \|y\| < \infty$. Using Cauchy-Schwarz, $2x \cdot y - \|y\|^2 \le 2\|x\| R_t$. Thus,

$$p_t(x) \le \mathcal{N}_t(x) \exp\left(\frac{2\|x\| R_t}{2\sigma_t^2}\right) = \frac{1}{(2\pi\sigma_t^2)^{d/2}} \exp\left(\frac{-\|x\|^2 + 2\|x\| R_t}{2\sigma_t^2}\right).$$

Completing the square in the exponent yields the form in the proposition statement.

**Lower Bound:** We apply Jensen's inequality ($\mathbb{E}[e^Z] \ge e^{\mathbb{E}[Z]}$). Let $\mu_t = \mathbb{E}[Y]$ and $V_t = \mathbb{E}[\|Y\|^2]$.

$$\frac{p_t(x)}{\mathcal{N}_t(x)} \ge \exp\left(\mathbb{E}_{Y \sim \rho_t}\left[\frac{2x \cdot Y - \|Y\|^2}{2\sigma_t^2}\right]\right) = \exp\left(\frac{2x \cdot \mu_t - V_t}{2\sigma_t^2}\right).$$

Multiplying by $\mathcal{N}_t(x)$ and rearranging terms yields the lower bound. $\square$

*Remark.* The compact support assumption is crucial. It ensures $R_t < \infty$ (valid upper bound) and the existence of all moments $\mu_t, V_t$ (valid lower bound). For heavy-tailed target distributions (e.g., Cauchy), these moments may not exist, invalidating the lower bound.

---

**Theorem B.1** (Integrability Preservation Condition for Compactly Supported Targets). *For each $i \in \{1, \ldots, n\}$, let $\{q_t^{(i)}\}_{t \in [0,1]}$ be probability paths in $\mathbb{R}^{d_i}$ generated by $X_t^{(i)} = \alpha_t^{(i)} X_0^{(i)} + \beta_t^{(i)} X_1^{(i)}$ where $X_0^{(i)} \sim q_0^{(i)} = \mathcal{N}(0, I)$ and $X_1^{(i)} \sim q_1^{(i)}$ has compact support. Let $d := \max_i d_i$ and $\gamma_i(t) \in \mathbb{R}$ for $t \in [0, 1]$.*

*Let $\pi_i : \mathbb{R}^d \to \mathbb{R}^{d_i}$ be projections onto coordinate sets $I_i$[a]. Define the lifted product*

$$h_t(x) := \prod_{i=1}^{n} (\tilde{q}_t^{(i)}(\pi_i(x)))^{\gamma_i(t)}$$

*If $h_1(x)$ is integrable and for every coordinate $k \in \{1, \ldots, d\}$ and all $t \in [0, 1)$,*

$$C_k(t) := \sum_{i:\, k \in I_i} \frac{\gamma_i(t)}{(\alpha_t^{(i)})^2} > 0, \tag{B.6}$$

*then $\{h_t\}_{t \in [0,1]}$ has the path existence property (i.e., $h_t \in L^1(\mathbb{R}^d)$ for all $t \in [0, 1]$).*

*Conversely, if there exists a coordinate $k^* \in \{1, \ldots, d\}$ and $t^* \in [0, 1)$ such that $C_{k^*}(t^*) < 0$, then $\{h_t\}_{t \in [0,1]}$ is not integrable at $t^*$ (Marginal Path Collapse).*

---
[a]We can write $\pi_i(x_1, \ldots, x_d) = (x_{k_1}, \ldots, x_{k_{d_i}})$. Let $I_i := \{k_1, \ldots, k_{d_i}\}$ be the set of coordinate indices that $\pi_i$ projects onto. Lifted densities are $\tilde{q}_t^{(i)} = q_t^{(i)} \circ \pi_i$.

---

*Proof.* **1. Sufficiency for path existence.**

**Endpoint ($t = 1$):** Integrability of $h_1$ holds by assumption.

**Interval $t \in [0, 1)$:** We construct an integrable upper bound. Applying Proposition B.2 to the lifted densities $\tilde{q}_t^{(i)}(x) = q_t^{(i)}(\pi_i(x))$, we use the upper bound formula for $\gamma_i(t) > 0$ and the lower bound formula for $\gamma_i(t) < 0$ (since negative exponents reverse the inequality).

$$h_t(x) \leq \prod_{\gamma_i > 0} \left( C_{+,i} e^{\frac{-\|\pi_i(x)\|^2 + 2\|\pi_i(x)\| R_t^{(i)}}{2(\alpha_t^{(i)})^2}} \right)^{\gamma_i} \prod_{\gamma_i < 0} \left( C_{-,i} e^{\frac{-\|\pi_i(x) - \mu_t^{(i)}\|^2 + \|\mu_t^{(i)}\|^2 - V_t^{(i)}}{2(\alpha_t^{(i)})^2}} \right)^{\gamma_i}$$

The integrability is determined by the coefficient of the quadratic term $\|x\|^2$ in the exponent. Expanding $\|\pi_i(x)\|^2 = \sum_{k \in I_i} x_k^2$, the aggregate quadratic term in the exponent is:

$$\sum_{i=1}^{n} \gamma_i(t) \left( -\frac{\|\pi_i(x)\|^2}{2(\alpha_t^{(i)})^2} \right) = -\frac{1}{2} \sum_{i=1}^{n} \sum_{k \in I_i} \frac{\gamma_i(t)}{(\alpha_t^{(i)})^2} x_k^2 = -\frac{1}{2} \sum_{k=1}^{d} \left( \sum_{i:\, k \in I_i} \frac{\gamma_i(t)}{(\alpha_t^{(i)})^2} \right) x_k^2. \tag{B.7}$$

By Condition B.6, the coefficient for every $x_k^2$ is strictly negative. Thus, the upper bound behaves as $\exp(-\frac{1}{2} x^\top \Lambda_t x + O(\|x\|))$ with positive definite $\Lambda_t$, ensuring integrability (Lemma B.3).

**2. Sufficiency for Marginal Path Collapse at $t^*$.**

We construct a diverging lower bound. Applying Proposition B.2 using the lower bound formula for $\gamma_i(t) > 0$ and the upper bound formula for $\gamma_i(t) < 0$, we derive:

$$h_t(x) \geq \prod_{\gamma_i > 0} \left( C_{-,i} e^{\frac{-\|\pi_i(x) - \mu_t^{(i)}\|^2 + \|\mu_t^{(i)}\|^2 - V_t^{(i)}}{2(\alpha_t^{(i)})^2}} \right)^{\gamma_i} \prod_{\gamma_i < 0} \left( C_{+,i} e^{\frac{-\|\pi_i(x)\|^2 + 2\|\pi_i(x)\| R_t^{(i)}}{2(\alpha_t^{(i)})^2}} \right)^{\gamma_i}$$

The quadratic term in exponent is precisely equation B.7 since the quadratic terms are the same for both upper and lower bounds in Proposition B.2. Thus, the exponent can be written as $-\frac{1}{2} x^\top A x + D(x)$, where $A$ is a diagonal matrix with

entries $A_{jj} = \sum_{i:k \in I_i} \frac{\gamma_i(t)}{(\alpha_t^{(i)})^2}$, and $D(x)$ collects the linear and constant terms. At time $t^*$, since $A_{k^*k^*} < 0$, the matrix $A$ is not positive definite ($A \not\succ 0$), and by Lemma B.3, the lower bound diverges, leading to Marginal Path Collapse. $\qquad\square$

*Remark.* As a consequence of Theorem B.1 (sufficiency), the path $p_t^* = h_t/Z_t$, $t \in [0,1]$ with $Z_t = \int_{\mathbb{R}^d} h_t(x)dx$ is well defined on $\mathbb{R}^d$, establishing the conditions for ACE (Theorem A.1).

*Remark.* For the boundary case where $C(t) = \min_k C_k(t) = 0$, the dominant quadratic term (Eq. B.7) vanishes, so sub-quadratic terms determine existence or collapse. As we show below in Example B.1, $C(t) = 0$ can yield either outcome. Hence, $C(t) > 0$ and $C(t) < 0$ represent the **sharpest possible universal conditions** under our assumptions.

---

**Example B.1** (The Boundary Case $C(t) = 0$). Consider a three-expert 1D unnormalized path

$$h_t(x) = \frac{q_t^{(1)}(x) q_t^{(2)}(x)}{q_t^{(3)}(x)}$$

where each expert $q_t^{(i)}$ defines a probability path between noise $\mathcal{N}(0,1)$ and data. Experts 1, 2 target the compactly supported uniform distribution $\mathcal{U}[-a,a]$ with noise schedules $\alpha_t, \beta_t$ and expert 3 targets $\mathcal{U}[-b,b]$ with noise schedules $\tilde{\alpha}_t, \tilde{\beta}_t$. One concrete example is $\alpha_t = 1 - t^2, \tilde{\alpha}_t = 1 - t, \beta_t = \tilde{\beta}_t = t$. We assume $a \le b$.

- **Valid Boundaries:** Both $h_0, h_1$ are normalizable since $h_0 = \mathcal{N}(0,1)$ and at $t = 1$, $h_1(x) = \frac{b}{2a^2} 1_{[-a,a]}(x)$.

- **Intermediate State:** At $t^* = \sqrt{2} - 1, \alpha_{t^*}^2 = 2\tilde{\alpha}_{t^*}^2$, causing $C(t^*) = 0$.

- **Tail Expansion:** By exact convolution (Lemma B.2) and the Mills ratio $\left( \bar{\Phi}(z)/(\frac{e^{-z^2/2}}{\sqrt{2\pi}}) \sim 1/z \right)$, the experts' tails decay as

$$p_t(x) \sim \frac{C_t}{|x| - a\beta_t} \exp(-\frac{(|x| - a\beta_t)^2}{2\alpha_t^2})$$

where $\sim$ means that the quotient of the two functions converges to 1 as $|x| \to \infty$ (Small, 2010). For $q^{(3)}$ we may simply replace $a$ with $b$. Canceling quadratic terms at $t^*$, the ratio's tail behaves as (up to a constant factor):

$$h_{t^*}(x) \sim \frac{1}{|x|} \exp\left( \frac{2(a\beta_{t^*} - b\tilde{\beta}_{t^*})}{\alpha_{t^*}^2} |x| \right) \text{ as } |x| \to \infty$$

Near $x = 0$, $h_{t^*}$ is finite and positive. Integrability depends on the tails:

1. **Collapse** ($a\beta_{t^*} = b\tilde{\beta}_{t^*}$): Exponent is zero. Tails decay as $1/|x|$, which diverges logarithmically.

2. **Existence** ($a\beta_{t^*} < b\tilde{\beta}_{t^*}$): Exponent is strictly negative ($-c|x|$). Tails decay exponentially ($\sim \frac{1}{|x|} e^{-c|x|}$), making it integrable.

---

## B.3. Adaptive Exponents and Bump Function Corrections

If the criterion in Theorem B.1 is violated, we can adapt the exponents $\gamma_i(t)$ to restore integrability. Since most schedulers $\alpha_t \to 0$ as $t \to 1$, analyzing the criterion as $t \to 1$ involves singularities. We take a practical approach by considering a discretization of the time interval $0 = t_0 < \cdots < t_M = t_{\text{end}} < 1$.

The standard ratio-of-densities formulation often encounters a singularity at $t = 1$ where $\alpha_1 = 0$. By truncating the steering phase at $t_{\text{end}}$ (e.g., 0.999), Theorem B.2 ensures the density $p_{t_{\text{end}}}$ is well-defined. The final transition from $p_{t_{\text{end}}}$ to $p_1$ is achieved via a single discretization step (Euler-Maruyama):

$$X_1 = X_{t_{\text{end}}} + v_{t_{\text{end}}}^*(X_{t_{\text{end}}}) \cdot (1 - t_{\text{end}})$$

Since $1 - t_{\text{end}}$ is negligible ($\approx 10^{-3}$), the error introduced by assuming a constant drift over this interval is bounded by $\mathcal{O}((1 - t_{\text{end}})^2)$, which is numerically insignificant compared to the stability gained by avoiding the singularity. This effectively "jumps" over the collapse region.

**Theorem B.2** (Adaptive Exponents). *Let $\alpha_t^{(i)}$ be differentiable on $[0, 1]$ with $\alpha_0^{(i)} = 1, \alpha_1^{(i)} = 0$, and $\alpha_t^{(i)} > 0$ for $t \in (0, 1)$. Let $\gamma_i(t)$ be the linear interpolation of fixed boundary values $\gamma_i(0), \gamma_i(1)$ such that*

$$S(t, \{\gamma_i\}_i) := \sum_{i=1}^{n} \frac{\gamma_i(t)}{(\alpha_t^{(i)})^2}$$

*satisfies $S(0, \{\gamma_i\}_i) > 0$. Then, there exist differentiable functions $\tilde{\gamma}_i(t)$ satisfying the boundary conditions such that $S(t, \{\tilde{\gamma}_i\}_i) > 0$ for all $t \in [0, t_{end}]$.*

*Proof.* We prove by constructing a working solution. Note that the solution is not unique.

**Step 1.** First check if the default solution (linear interpolation of boundary values), $\gamma_i(t) = (1-t)\gamma_i(0) + t\gamma_i(1)$ for all $i$ satisfies $S(t, \{\gamma_i\}_i) > 0 \ \forall t \in [0, t_{end}]$. If so, we are done.

**Step 2.** If the default $\gamma_i$ fails, it is because there exists some $t$ where the criterion drops below zero.

1. Since $S(0, \{\gamma_i\}_i) > 0$ by assumption and $S$ is continuous in $t$, there exists a small time $\delta > 0$ such that $S(t, \{\gamma_i\}_i) > 0$ for all $t \in [0, \delta]$. Thus, the failure must occur in the compact interval $[\delta, t_{end}]$. Let $S_{\min}$ be the global minimum on this interval:

$$S_{\min} := \min_{t \in [\delta, t_{end}]} S(t, \{\gamma_i\}_i).$$

If $S_{\min} > 0$, we are done. Assume $S_{\min} \leq 0$.

2. Define the **bump function** $b(t)$ such that $b(0) = b(1) = 0$ and $b(t) > 0$ on $(0, 1)$. We combine a quadratic bump $Q(t) = t(1-t)$ with a linear bump $L_\tau(t)$ using coefficients $B_1, B_2 \geq 0$:

$$b(t) = B_1 Q(t) + B_2 L_\tau(t)$$

where $L_\tau(t) = \min(t, \tau(1-t))$ is a linear bump that increases until $t = \frac{\tau}{1+\tau}$ and drops thereafter. By choosing a sufficiently large $\tau$ (specifically $\tau > \frac{t_{end}}{1-t_{end}}$), we ensure the peak occurs after $t_{end}$, so $L_\tau(t) = t$ for all $t \in [0, t_{end}]$.

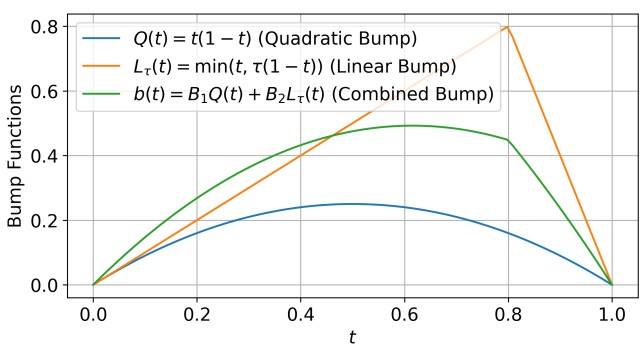

*Figure B.1.* Illustration of the bump function $b(t) = B_1 Q(t) + B_2 L_\tau(t)$.

3. Choose an index $j$ where $\gamma_j > 0$ to apply the correction.

4. Since $\alpha_t^{(j)}$ is bounded and $Q(t), L_\tau(t)$ is strictly positive on $[\delta, t_{end}]$, the following minimum exists and is strictly positive:

$$c_1 := \min_{t \in [\delta, t_{end}]} \frac{Q(t)}{(\alpha_t^{(j)})^2} > 0, \quad c_2 := \min_{t \in [\delta, t_{end}]} \frac{L_\tau(t)}{(\alpha_t^{(j)})^2} > 0.$$

5. Finally, define the **adaptive exponent** via

$$\tilde{\gamma}_j(t) := \gamma_j(t) + \underbrace{B_1 Q(t) + B_2 L_\tau(t)}_{b(t)}$$

where we choose the coefficients $B_1, B_2$ large enough to counteract the worst-case collapse:

$$B_1 c_1 + B_2 c_2 = |S_{\min}| + \epsilon$$

for any $\epsilon > 0$. All other functions are preserved: $\tilde{\gamma}_i = \gamma_i$ for $i \neq j$. Hyperparameter choices are studied in Tables C.3–C.5.

**Step 3. Verification.** We verify that $S(t, \{\tilde{\gamma}_i\}_i)$ is strictly positive for all $t \in [0, t_{\text{end}}]$.

- For $t \in [0, \delta)$: $S(t) > 0$ by the initial continuity argument, and since $b(t) \geq 0$, the sum increases, preserving positivity.

- For $t \in [\delta, t_{\text{end}}]$:

$$S(t, \{\tilde{\gamma}_i\}_i) = S(t, \{\gamma_i\}_i) + \frac{B_1 Q(t) + B_2 L_\tau(t)}{(\alpha_t^{(j)})^2}$$

$$\geq S_{\min} + |S_{\min}| + \epsilon$$

$$= \epsilon > 0.$$

Thus, the modified path satisfies the existence criterion on the entire simulation interval. $\qquad\square$

---

**Corollary B.1** (Extension to Partial Support). *When experts act on heterogeneous supports, we apply the logic of Theorem B.2 repeatedly across coordinate subsets to guarantee the coordinate-wise criterion $C_k(t) > 0$ for all coordinates $k$. Apply the following steps:*

1. ***Subset Application:*** *Let $I_{(k)} = \{i : k \in I_i\}$ be the index set of the experts acting on coordinate $k$. $C_k(t)$ is exactly the scalar sum $S(t, \{\gamma_i\}_{i \in I_{(k)}})$.*

2. ***Base Case:*** *Since Theorem B.2 assumes $C(0) = \min_k C_k(0) > 0$ and $\alpha_0^{(i)} = 1$, we have $S(0, \{\gamma_i\}_{i \in I_{(k)}}) = C_k(0) > 0 \forall k$.*

3. ***Coordinate-wise Bumps:*** *Applying Theorem B.2 to $I_{(k)}$ yields nonnegative bumps $b_{i,k}(t)$ ensuring $C_k(t) > 0 \ \forall t \in [0, t_{end}]$.*

4. ***Global Fix:*** *Adding a positive bump strictly increases $S$. Thus, for each expert $i$, summing the required bumps $\tilde{\gamma}_i(t) = \gamma_i(t) + \sum_k b_{i,k}(t)$ guarantees $C_k(t) > 0$ simultaneously for all $k$.*

---

*Remark.* In our experiments, modifying a single index $j$ sufficed (one expert covered all coordinates).

### B.4. Gaussian Mixture Models

We investigate integrability preservation for Gaussian Mixture Models (GMMs). Consider probability paths $\{p_t\}_{t \in [0,1]}$ generated by stochastic interpolants (Definition B.1), where the initial distribution $p_0$ is standard normal $\mathcal{N}(0, I)$ and the final distribution $p_1$ is a GMM.

---

**Definition B.2** (Gaussian Mixture Model). A **Gaussian Mixture Model (GMM)** $p(x)$ in $\mathbb{R}^d$ is a probability density of the form:

$$p(x) = \sum_{j=1}^{J} w_j \mathcal{N}(x; \mu_j, \Sigma_j) \tag{B.8}$$

where weights $w_j > 0$ sum to 1, and each covariance matrix $\Sigma_j$ is positive definite.

---

**Product of GMMs.** Similar to the compactly supported target case, integrability at the boundaries does not automatically imply integrability at intermediate times. A counterexample is provided in the main text (Eq. 3 and Fig. 2), demonstrating that intermediate paths can diverge even when endpoints are valid.

To guarantee integrability for ratios of product-of-GMMs, we must enforce a stronger structural condition. We first derive precise exponential bounds.

---

**Lemma B.4** (Exponential Bounds for GMMs). *Let $q(x) = \sum_{j=1}^{J} c_j \mathcal{N}(x; \mu_j, \Sigma_j)$. Define the extremal eigenvalues $\lambda_{\max}(q) := \max_j \lambda_{\max}(\Sigma_j)$ and $\lambda_{\min}(q) := \min_j \lambda_{\min}(\Sigma_j)$. Then there exist finite constants $K_\pm, L_\pm > 0$ such that for all $x \in \mathbb{R}^d$:*

$$q(x) \leq K_+ \exp\left(-\frac{\|x\|^2}{2\lambda_{\max}(q)} + L_+\|x\|\right), \tag{B.9}$$

$$q(x) \geq K_- \exp\left(-\frac{\|x\|^2}{2\lambda_{\min}(q)} - L_-\|x\|\right). \tag{B.10}$$

*Proof.* **Upper Bound:** $q(x) \leq \sum_j \mathcal{N}(x; \mu_j, \Sigma_j)$. For each component $j$, we bound the quadratic form in the exponent using Rayleigh quotients:

$$(x - \mu_j)^\top \Sigma_j^{-1}(x - \mu_j) \geq \lambda_{\max}(\Sigma_j)^{-1}\|x - \mu_j\|^2 \geq \frac{1}{\lambda_{\max}(q)}(\|x\|^2 - 2\|x\|\|\mu_j\|).$$

Substituting this into the Gaussian PDF, we define $L_+ = \max_j \frac{\|\mu_j\|}{\lambda_{\max}(q)}$ and collect constants into $K_+$.

**Lower Bound:** $q(x) \geq c_{j^*}\mathcal{N}(x; \mu_{j^*}, \Sigma_{j^*})$ for any index $j^*$. We upper bound the quadratic form:

$$(x - \mu_{j^*})^\top \Sigma_{j^*}^{-1}(x - \mu_{j^*}) \leq \lambda_{\min}(\Sigma_{j^*})^{-1}\|x - \mu_{j^*}\|^2 \leq \frac{1}{\lambda_{\min}(q)}(\|x\|^2 + 2\|x\|\|\mu_{j^*}\| + \|\mu_{j^*}\|^2).$$

Substituting this yields the form in Equation (B.10) with $L_- = \frac{\|\mu_{j^*}\|}{\lambda_{\min}(q)}$. $\qquad\square$

We now state the sufficient and necessary conditions for integrability of heterogeneous products.

---

**Theorem B.3** (Integrability Preservation for Heterogeneous Products). *Let $\{q_t^{(i)}\}$ be GMM probability paths generated by $X_t^{(i)} = \alpha_t^{(i)} X_0^{(i)} + \beta_t^{(i)} X_1^{(i)}$ with $X_0^{(i)} \sim \mathcal{N}(0, I)$. Let $\pi_i : \mathbb{R}^d \to \mathbb{R}^{d_i}$ be projections onto coordinate index sets $I_i$, and define the composite function $g_t(x) := \prod_{i=1}^{n}(\tilde{q}_t^{(i)}(x))^{\gamma_i}$, where $\tilde{q}_t^{(i)} = q_t^{(i)} \circ \pi_i$.*

*For each coordinate $k \in \{1, \ldots, d\}$ and time $t \in [0, 1)$, we define two stability coefficients. The **sufficiency criterion** $C_k^{suff}(t)$ is defined as:*

$$C_k^{suff}(t) := \sum_{i:k\in I_i, \gamma_i > 0} \frac{\gamma_i}{\lambda_{\max}(q_t^{(i)})} + \sum_{i:k\in I_i, \gamma_i < 0} \frac{\gamma_i}{\lambda_{\min}(q_t^{(i)})}. \tag{B.11}$$

*The **necessity criterion** $C_k^{nec}(t)$ is defined as:*

$$C_k^{nec}(t) := \sum_{i:k\in I_i, \gamma_i > 0} \frac{\gamma_i}{\lambda_{\min}(q_t^{(i)})} + \sum_{i:k\in I_i, \gamma_i < 0} \frac{\gamma_i}{\lambda_{\max}(q_t^{(i)})}. \tag{B.12}$$

*If $g_1(x)$ is integrable, then:*

1. *(**Sufficiency**) If $C_k^{suff}(t) > 0$ for all $k$ and all $t \in [0, 1)$, then $g_t(x)$ is integrable for all $t \in [0, 1]$.*

2. *(**Necessity**) Conversely, if there exists a coordinate $k$ and time $t$ such that $C_k^{nec}(t) < 0$, then $g_t(x)$ is not integrable.*

---

*Proof.* **Sufficiency ($C_k^{\text{suff}}(t) > 0$):** We construct an integrable upper bound for $g_t(x)$. To upper bound a product with positive and negative exponents, we require an *upper* bound for terms with $\gamma_i > 0$ and a *lower* bound for terms with $\gamma_i < 0$ (since negative exponents invert the inequality). Applying Lemma B.4:

$$g_t(x) \leq \prod_{\gamma_i > 0}\left(K_{+,i} e^{-\frac{\|\pi_i(x)\|^2}{2\lambda_{\max}(q_t^{(i)})} + L_{+,i}\|\pi_i(x)\|}\right)^{\gamma_i} \prod_{\gamma_i < 0}\left(K_{-,i} e^{-\frac{\|\pi_i(x)\|^2}{2\lambda_{\min}(q_t^{(i)})} - L_{-,i}\|\pi_i(x)\|}\right)^{\gamma_i}$$

$$= K \exp\left(-\frac{1}{2}\sum_{k=1}^{d} C_k^{\text{suff}}(t)x_k^2 + \sum_{i=1}^{n} \delta_i\|\pi_i(x)\|\right).$$

The coefficient of the quadratic term $x_k^2$ in the exponent corresponds exactly to $C_k^{\text{suff}}(t)$ defined in Eq. B.11. The linear terms are bounded by $D\|x\|$ for some finite $D > 0$ (the constants $\delta_i$ are also finite). Thus, the exponent behaves as $-\frac{1}{2}x^\top \text{diag}(C^{\text{suff}}(t))x + O(\|x\|)$. Since $C_k^{\text{suff}}(t) > 0$ for all $k$, the quadratic decay dominates the linear growth, ensuring integrability on $\mathbb{R}^d$.

**Necessity ($C_k^{\text{nec}}(t) < 0$):** We prove this by constructing a lower bound that diverges. To lower bound the product, we require a *lower* bound for terms with $\gamma_i > 0$ and an *upper* bound for terms with $\gamma_i < 0$. Applying the reverse inequalities from Lemma B.4:

$$g_t(x) \geq \prod_{\gamma_i > 0} \left( K_{-,i} e^{-\frac{\|\pi_i(x)\|^2}{2\lambda_{\min}(q_t^{(i)})} - L_{-,i}\|\pi_i(x)\|} \right)^{\gamma_i} \prod_{\gamma_i < 0} \left( K_{+,i} e^{-\frac{\|\pi_i(x)\|^2}{2\lambda_{\max}(q_t^{(i)})} + L_{+,i}\|\pi_i(x)\|} \right)^{\gamma_i}$$

$$= \tilde{K} \exp\left( -\frac{1}{2} \sum_{k=1}^{d} C_k^{\text{nec}}(t) x_k^2 - \tilde{D}(x) \right)$$

where the quadratic coefficient is exactly $C_k^{\text{nec}}(t)$ defined in Eq. B.12, and $\tilde{D}(x)$ captures linear terms. The exponent is quadratic with coefficient matrix $A$, a diagonal matrix whose entries are $A_{jj} = C_k^{\text{nec}}(t)$. Let $k^*$ be a coordinate such that $C_{k^*}^{\text{nec}}(t) < 0$. Then $A_{k^*k^*} < 0$, so $A$ is not positive definite ($A \not\succ 0$), and by Lemma B.3 the integral over $\mathbb{R}^d$ diverges. $\square$

**Revisiting the counterexample in the main text.** We apply Theorem B.3 to analyze the pathological Gaussian path example presented in Eq. 3 and Figure 2 of the main text. Since the components are single Gaussians (trivial GMMs with $J = 1$), the spectral bounds collapse to the scalar variance: $\lambda_{\max}(q_t^{(i)}) = \lambda_{\min}(q_t^{(i)}) = \sigma_i^2(t)$. Consequently, the sufficiency and necessity criteria coincide: $C_k^{\text{suff}}(t) = C_k^{\text{nec}}(t) =: C(t)$. The dimension is $d = 1$. The exponents are $\gamma_1 = 1, \gamma_2 = 1$ (numerator) and $\gamma_3 = -1, \gamma_4 = -1$ (denominator). The variance paths are given by $\sigma_i^2(t) = (1-t)^2\sigma_{i,0}^2 + t^2\sigma_{i,1}^2$. We evaluate the criterion at the critical time $t = 0.5$:

$$\sigma_1^2(0.5) = (0.5)^2(1) + (0.5)^2(0.5) = 0.375 \quad \sigma_2^2(0.5) = (0.5)^2(1) + (0.5)^2(7) = 2.0$$
$$\sigma_3^2(0.5) = (0.5)^2(1.5) + (0.5)^2(1) = 0.625 \quad \sigma_4^2(0.5) = 0.625 \quad \text{(identical to } q^{(3)})$$

Substituting these into Eq. B.12, $C(0.5) \approx -0.03 < 0$. Since $C(0.5) < 0$, the necessity condition of Theorem B.3 implies that $g_{0.5}(x)$ is **not integrable**. This theoretical prediction aligns perfectly with the empirical divergence and "Marginal Path Collapse" illustrated in Figure 2.

## C. Experimental Setup

For full reproducibility, we provide the exact analytical expressions for the noise schedules $\alpha_t$ and $\beta_t$ used in our experiments in Table C.1. The stochastic interpolant is defined as $X_t = \alpha_t X_0 + \beta_t X_1$.

*Table C.1.* Exact formulations of noise schedules used in experiments.

| Schedule Name | Signal Scale $\alpha_t$ | Noise Scale $\beta_t$ |
|---|---|---|
| $1 - t$ | $1 - t$ | $t$ |
| $1 - t^2$ | $1 - t^2$ | $t$ |
| $\cos(\frac{\pi}{2}t)$ | $\cos\left(\frac{\pi}{2}t\right)$ | $\sin\left(\frac{\pi}{2}t\right)$ |
| DDPM | $\exp\left(-\frac{1}{4}(19.9t^2 + 0.1t)\right)$ | $\sqrt{1 - \alpha_t^2}$ |
| Sigmoid$^\dagger$ | $\sqrt{1 - \exp(-\eta(1-t))}$ | $t$ |
| Custom | $1 - 4t + 7t^2 - 4t^3$ | $t$ |

**Custom**: This polynomial schedule was used specifically for the synthetic quantitative benchmark (Table C.2) to test robustness against non-monotonic effective variances. **DDPM** corresponds to the variance preserving (VP) schedule with linear $\beta$-scheduling from $\beta_{\min} = 0.1$ to $\beta_{\max} = 20$ (scaled to $t \in [0,1]$). The expression is derived from $\alpha_t = \sqrt{\exp(-\int_0^t \beta(s)ds)}$. For **Sigmoid** we utilize a numerically stable formulation involving the softplus function. The term $\eta(x)$ is defined as:

$$\eta(x) = \frac{20}{12}\text{softplus}(12(x - 0.5)) + 0.001x, \quad \text{where softplus}(z) = \log(1 + e^z).$$

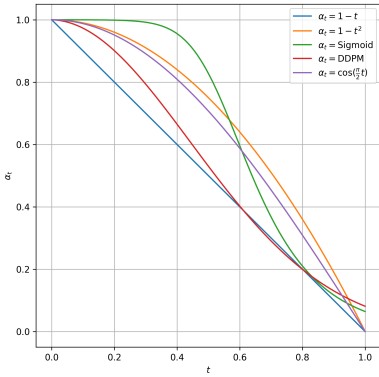

*Figure C.2.* Visualization of common noise schedules used in experiments.

## C.1. Synthetic Dataset: 2D Checker

**Dataset.** We evaluate the fidelity of samples generated by ACE, NR, and FKC by computing distributional discrepancies against the ground truth, reporting Wasserstein distances $(W_1, W_2)$ and Maximum Mean Discrepancy with an RBF kernel (MMD). The ground truth 2D Checkerboard distribution on the domain $\mathcal{D} = [-4, 4]^2$ is formally defined by the density:

$$p_{\texttt{Checker}}(x, y) = \frac{1}{32} \cdot \mathbf{1}\left(\left\lfloor \frac{x}{2}\right\rfloor + \left\lfloor\frac{y}{2}\right\rfloor \text{ is odd}\right) \cdot \mathbf{1}\left((x, y) \in \mathcal{D}\right), \tag{C.1}$$

where $\lfloor \cdot \rfloor$ denotes the floor function and $\mathbf{1}(\cdot)$ is the indicator function. The support consists of alternating $2 \times 2$ squares.

**Implementation.** For all experiments, we sampled 10,000 samples over 1,000 SDE steps with noise level $\sigma_t = 0.5$. For the 2D benchmark experiments, we chose to resample based on Effective Sample Size (ESS), a commonly used technique in sequential Monte Carlo (Naesseth et al., 2019). We used a threshold of 0.7 (Hyperparameter study in Figure. C.3). For both 1D and 2D experts, we trained a multilayer perceptron (MLP) with a time embedding module concatenated to the input. The main network consisted of four hidden layers of 256 units with SiLU activations. Depending on the task dimensionality, the input layer size was $d + 256$ (where $d = 1, 2$ for 1D/2D inputs, respectively) and the output dimension matched the data dimension. Models were trained with the interpolant loss (Albergo et al., 2025) using the Adam optimizer at learning rate $2 \times 10^{-3}$, for 2000 epochs on 1D tasks and 10,000 epochs on 2D tasks. We use Hutchinson's estimator (Hutchinson, 1989) to compute divergences.

**Simple Experiment for Hyperparameter Search.** To efficiently search for the effective sample size (ESS) threshold used in SMC simulation, we conduct a simplified ablation study using a reduced bump function without the linear term ($B_2 = 0$), retaining only the initial bump amplitude $B_1 = 30$ and the custom scheduler in Table C.1. We first evaluate a range of ESS thresholds $\tau \in \{0.1, 0.3, 0.5, 0.7, 0.9\}$ and find that $\tau = 0.7$ achieves the best performance (Table C.3). We then verify that this choice is robust across different values of $B_1$ through additional experiments, as shown in Fig. C.4. Based on this selected threshold and the schedule combination of Table 2, we further conduct sensitivity analyses using the complete bump function, which includes both quadratic and linear components, as reported in Tables C.3, C.4, and C.5.

**Path collapse frequencies.** For annealed three-expert composition $q_t^{(1)}(q_t^{(2)}/q_t^{(3)})^w$, there are $5^3 = 125$ total triplets, 120 heterogeneous triplets (excluding $\alpha_t^{(1)} = \alpha_t^{(2)} = \alpha_t^{(3)}$), and 100 likelihood-nonhomogeneous ($\alpha_t^{(2)} \neq \alpha_t^{(3)}$) triplets, the heterogeneous steering case. Among the 100 heterogeneous steering cases, path collapse frequencies are given in Table. E.10 with the full list of schedule compositions given in Figure. E.10.

**Visualization Details.** All sample/trajectory visualizations use seed 0. In criterion plots Figures 3 and E.10, we visualize the criterion $C(t)$ to demonstrate the prevalence of path collapse. To maintain a consistent and readable scale across all 100 combinations, we fix the y-axis range to $[-20, 100]$. We observed that in some cases (e.g., high-guidance regimes $w = 15$), the criterion may exhibit rapid asymptotic growth followed by a sharp singularity as $t \to 1$. In these cases we cease plotting the line segment after $t_{\text{end}}$ to prevent visual clutter. This visualization focuses on the practically relevant operational range for discretized sampling ($t \leq t_{\text{end}}$, Theorem B.2).

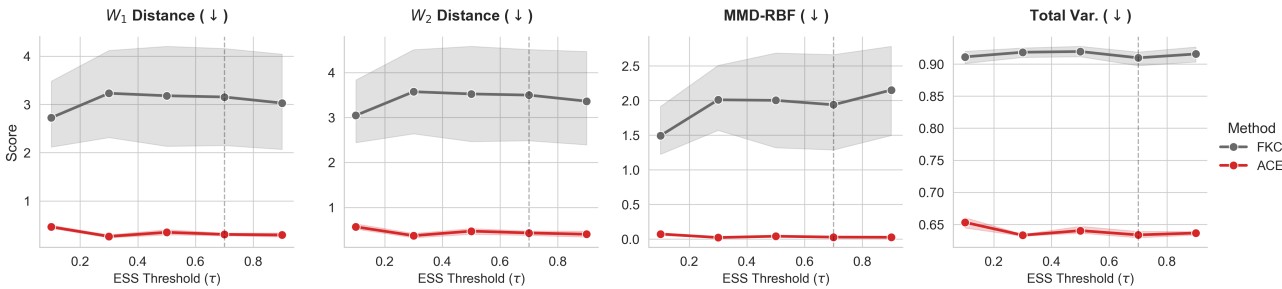

*Figure C.3.* **Performance comparison across varying ESS Thresholds.** Across multiple ESS thresholds, $\tau \in \{0.1, 0.3, 0.5, 0.7, 0.9\}$, ACE displays stable performance (with best values at $\tau = 0.7$) across different hyperparameter settings, outperforming FKC by a large margin. FKC consistently performs worse, implying its performance degradation is due to path collapse and not due to suboptimal parameter tuning. We used the same evaluation setup as in our main experiments. Shaded regions denote variance across 5 seeds.

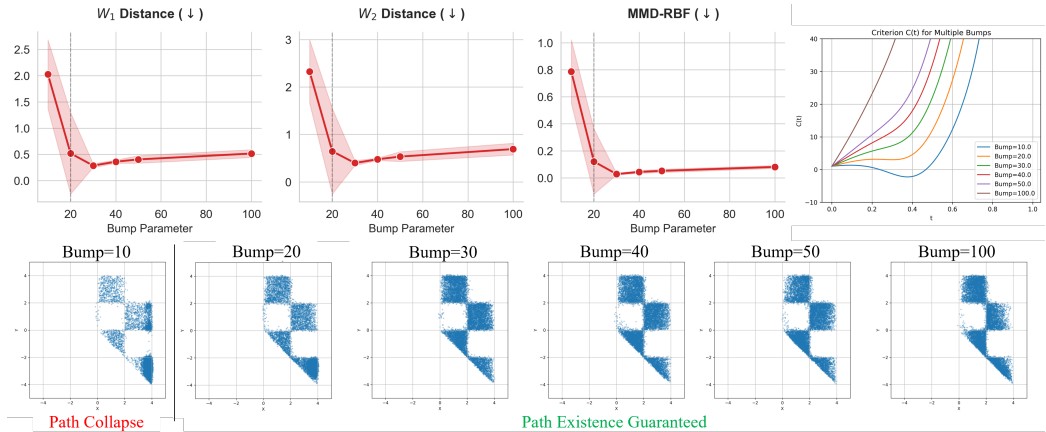

*Figure C.4.* **Sensitivity to the bump parameter $B_1$ ($B_2 = 0$).** Performance peaks near $B_1 = 30$, but remains robust even at high values ($B_1 = 100$), whereas criterion violations at $B_1 = 10$ cause significant degradation. Shaded regions denote variance across 5 seeds.

### C.2. Flexible-Pose Scaffold Decoration

**Dataset.** We validate our flexible-pose scaffold decoration framework on the CrossDocked2020 test set (Francoeur et al., 2020). Specifically, we select 76 pocket-ligand pairs from the original 100 test pairs, excluding cases where the ligand contains more than 29 atoms, which exceeds the predefined maximum supported by EDM during training. We adhere to this constraint because our SDE-based simulation requires EDM score predictions over the entire molecule. Furthermore, scaffolds are extracted as ring structures, defined as a randomly selected ring along with atoms within a bond distance of two from the ring. Further, due to limited atom type support of EDM, we removed atoms other than C, O, N, and F.

**Our Implementation.** As described in Sec. 2.5, we combine pretrained density models GeoDiff (Xu et al., 2022) as CONF, DiffSBDD (Schneuing et al., 2024) as SBDD, and EDM (Hoogeboom et al., 2022) as DN. DiffSBDD is trained on the CrossDocked dataset, while GeoDiff and EDM are trained on QM9 (Ramakrishnan et al., 2014). This combination enables molecule generation that preserves scaffold topology while allowing scaffold poses to adapt flexibly within the binding pocket. We perform 500 denoising steps with $\log q$ updates and apply resampling every 10 steps. For weighting the ratio-of-density term, we use weight values of 1.1, 1.2, 1.3, and 1.4, with corresponding bump function coefficients $B_1 = 30$ and $B_2 = 0.037, 0.136, 0.236$, and $0.336$, respectively. The $B_2$ values are selected to satisfy the prescribed criterion under the fixed $B_1 = 30$. Since our diffusion models treat molecules as point clouds, we apply a post-processing step for bond prediction. Starting from the predefined scaffold topology, bonds are added between nearby atoms based on geometric proximity and valence constraints. For evaluation, we sample five candidate molecules for each pocket–scaffold pair and compute the metrics reported below. All experiments are conducted using Python 3.8.9 on NVIDIA A6000 GPUs. We additionally note that ACE is evaluated with a batch size of 5 and requires approximately 1.5 GB of GPU VRAM.

**Baselines.** To demonstrate the effectiveness of ACE over existing diffusion steering methods, as well as the practical importance of the ACE-based framework for flexible-pose scaffold decoration, we consider two categories of baselines.

*Table C.2.* Distributional similarity metrics ($\downarrow$) under the custom schedule. Results exhibit the same trend as Table 2. For each metric, we report the minimum and mean $\pm$ standard deviation across 5 seeds. Best values are in **bold**. NR denotes no resampling, the common heuristic using only mixed scores. FKC refers to Feynman–Kac correctors, which fail when path-existence conditions are not satisfied.

| Method | $W_1$ ($\downarrow$) | | $W_2$ ($\downarrow$) | | MMD (RBF) ($\downarrow$) | |
|---|---|---|---|---|---|---|
| | Min | Mean $\pm$ Std | Min | Mean $\pm$ Std | Min | Mean $\pm$ Std |
| NR | 0.77 | $0.78 \pm 0.02$ | 1.06 | $1.07 \pm 0.02$ | 0.066 | $0.068 \pm 0.001$ |
| FKC | 2.09 | $2.13 \pm 0.04$ | 2.39 | $2.44 \pm 0.05$ | 1.07 | $1.43 \pm 0.31$ |
| ACE ($B_1 = 10, B_2 = 0$) | 1.47 | $2.02 \pm 0.65$ | 1.77 | $2.32 \pm 0.67$ | 0.44 | $0.78 \pm 0.23$ |
| ACE ($B_1 = 20, B_2 = 0$) | 0.13 | $0.52 \pm 0.77$ | 0.18 | $0.64 \pm 0.90$ | 0.009 | $0.12 \pm 0.24$ |
| ACE ($B_1 = 30, B_2 = 0$) | 0.24 | $\mathbf{0.28} \pm 0.036$ | 0.35 | $\mathbf{0.40} \pm 0.51$ | 0.019 | $\mathbf{0.027} \pm 0.0064$ |
| ACE ($B_1 = 40, B_2 = 0$) | 0.32 | $0.36 \pm 0.031$ | 0.44 | $0.475 \pm 0.025$ | 0.034 | $0.043 \pm 0.0099$ |
| ACE ($B_1 = 50, B_2 = 0$) | 0.30 | $0.40 \pm 0.07$ | 0.38 | $0.53 \pm 0.10$ | 0.034 | $0.052 \pm 0.01$ |
| ACE ($B_1 = 100, B_2 = 0$) | 0.43 | $0.52 \pm 0.07$ | 0.56 | $0.69 \pm 0.12$ | 0.070 | $0.080 \pm 0.011$ |

*Table C.3.* ACE Sensitivity Analysis: $W_1$ ($\downarrow$)

| $B_1 \setminus B_2$ | 0.0 | 0.5 | 1.0 | 1.5 | 2.0 | 2.5 | 3.0 | 4.0 | 8.0 | 16.0 |
|---|---|---|---|---|---|---|---|---|---|---|
| 0.0 | $1.23_{\pm 0.87}$ | $0.59_{\pm 0.39}$ | $1.10_{\pm 0.93}$ | $0.78_{\pm 0.15}$ | $0.37_{\pm 0.07}$ | $0.56_{\pm 0.30}$ | $0.75_{\pm 0.06}$ | $0.66_{\pm 0.17}$ | $0.66_{\pm 0.09}$ | $2.60_{\pm 0.58}$ |
| 10.0 | $0.71_{\pm 0.02}$ | $0.52_{\pm 0.07}$ | $0.33_{\pm 0.05}$ | $\mathbf{0.20}_{\pm 0.04}$ | $0.21_{\pm 0.04}$ | $0.53_{\pm 0.06}$ | $0.49_{\pm 0.04}$ | $0.53_{\pm 0.16}$ | $0.70_{\pm 0.25}$ | $1.68_{\pm 0.53}$ |
| 20.0 | $0.33_{\pm 0.19}$ | $0.26_{\pm 0.07}$ | $0.26_{\pm 0.03}$ | $0.39_{\pm 0.24}$ | $0.33_{\pm 0.07}$ | $0.47_{\pm 0.11}$ | $0.46_{\pm 0.07}$ | $0.52_{\pm 0.13}$ | $0.63_{\pm 0.13}$ | $1.30_{\pm 0.35}$ |
| 30.0 | $0.33_{\pm 0.11}$ | $0.35_{\pm 0.17}$ | $0.33_{\pm 0.10}$ | $0.31_{\pm 0.04}$ | $0.41_{\pm 0.10}$ | $0.36_{\pm 0.11}$ | $0.35_{\pm 0.12}$ | $0.45_{\pm 0.12}$ | $0.83_{\pm 0.24}$ | $1.41_{\pm 0.60}$ |
| 40.0 | $0.44_{\pm 0.12}$ | $0.41_{\pm 0.08}$ | $0.42_{\pm 0.11}$ | $0.47_{\pm 0.13}$ | $0.49_{\pm 0.10}$ | $0.51_{\pm 0.10}$ | $0.58_{\pm 0.16}$ | $0.73_{\pm 0.53}$ | $0.78_{\pm 0.27}$ | $1.45_{\pm 0.45}$ |
| 50.0 | $0.56_{\pm 0.24}$ | $0.59_{\pm 0.12}$ | $0.59_{\pm 0.15}$ | $0.66_{\pm 0.25}$ | $0.63_{\pm 0.08}$ | $0.71_{\pm 0.21}$ | $0.59_{\pm 0.10}$ | $0.66_{\pm 0.15}$ | $0.84_{\pm 0.15}$ | $1.29_{\pm 0.40}$ |
| 100.0 | $0.91_{\pm 0.07}$ | $0.93_{\pm 0.20}$ | $1.12_{\pm 0.40}$ | $0.84_{\pm 0.07}$ | $0.98_{\pm 0.13}$ | $0.98_{\pm 0.15}$ | $0.98_{\pm 0.22}$ | $1.03_{\pm 0.18}$ | $1.66_{\pm 0.66}$ | $2.12_{\pm 0.37}$ |

First, we construct two baselines based on FKC and CFG (classifier-free guidance) to approximate the same target density (Sec. 2.5). The CFG-based approach is referred to as NR (Non-Resampling), as its simulation is equivalent to FKC without the resampling step. We note that these baselines share the same underlying models as our method for each component probability path. In addition, to evaluate the practical advantages of flexible-pose scaffold decoration enabled by ACE, we include fixed-pose scaffold decoration baselines: Delete (Chen et al., 2025) (with a maximum of 50 steps), and AutoFragDiff (Ghorbani et al., 2023).

**Metrics.** To compare ACE with the baseline diffusion steering methods NR and FKC, we evaluate the sampling trajectories using three categories of metrics. (i) *Vina score*, computed with QVina2 (Alhossary et al., 2015), measures pocket compatibility. When QVina2 fails (e.g., due to fragmented molecules), a score of zero is assigned. We report *P.Worst*, defined as the average of the pocket-wise worst Vina scores; *Top25%*, corresponding to the top 25th percentile among all generated molecules; and the mean and standard deviation computed over all generated molecules. (ii) *Optimization Success Rate* (OSR) measures the fraction of generated molecules whose docking scores are better than that of the reference ligand. (iii) *Drug-likeness* metrics assess whether the generated distribution preserves the prior distribution of drug-like molecules. Specifically, we report QED (Bickerton et al., 2012), SA (synthetic accessibility), and the Lipinski score, defined as the number of satisfied Lipinski rules divided by five (Lipinski et al., 1997).

### C.3. Time-Varying Exponent Baseline

To disentangle our bump function from general time-varying exponents in prior work, we tested empirical dynamic schedules (Wang et al., 2024b) on scaffold decoration (3 seeds, Figure C.5) and 2D checker (5 seeds, Figure C.6): linearly increasing ($w\alpha t$), decreasing ($w\alpha(1 - t)$), $\Lambda$-shaped ($w\alpha(0.5 - |t - 0.5|)$), and V-shaped ($w\alpha|t - 0.5|$) for $\alpha \in \{0.5, 1, 2, 4, 8\}$ ($w = 1.4$ for scaffold, $w = 1.0$ for 2D). For scaffold decoration, we also tested quadratic bumps $b(t) = B_1 t(1 - t)$ on $\gamma_4$ ($B_1 \in \{10, 20, 30\}$), scaled to deliberately fail the Path Existence Criterion. Across all samplers (ACE/FKC/NR), these heuristic exponent schedules failed to prevent path collapse, degrading performance versus ours. These results confirm that the gains come from validity-preserving exponent design rather than time variation alone.

## D. Background and Related Work

Our work sits at the intersection of three key areas: inference-time control for generative models, the theory of path sampling correctors, and the practical challenge of composing pretrained models for scientific discovery. We situate our contributions

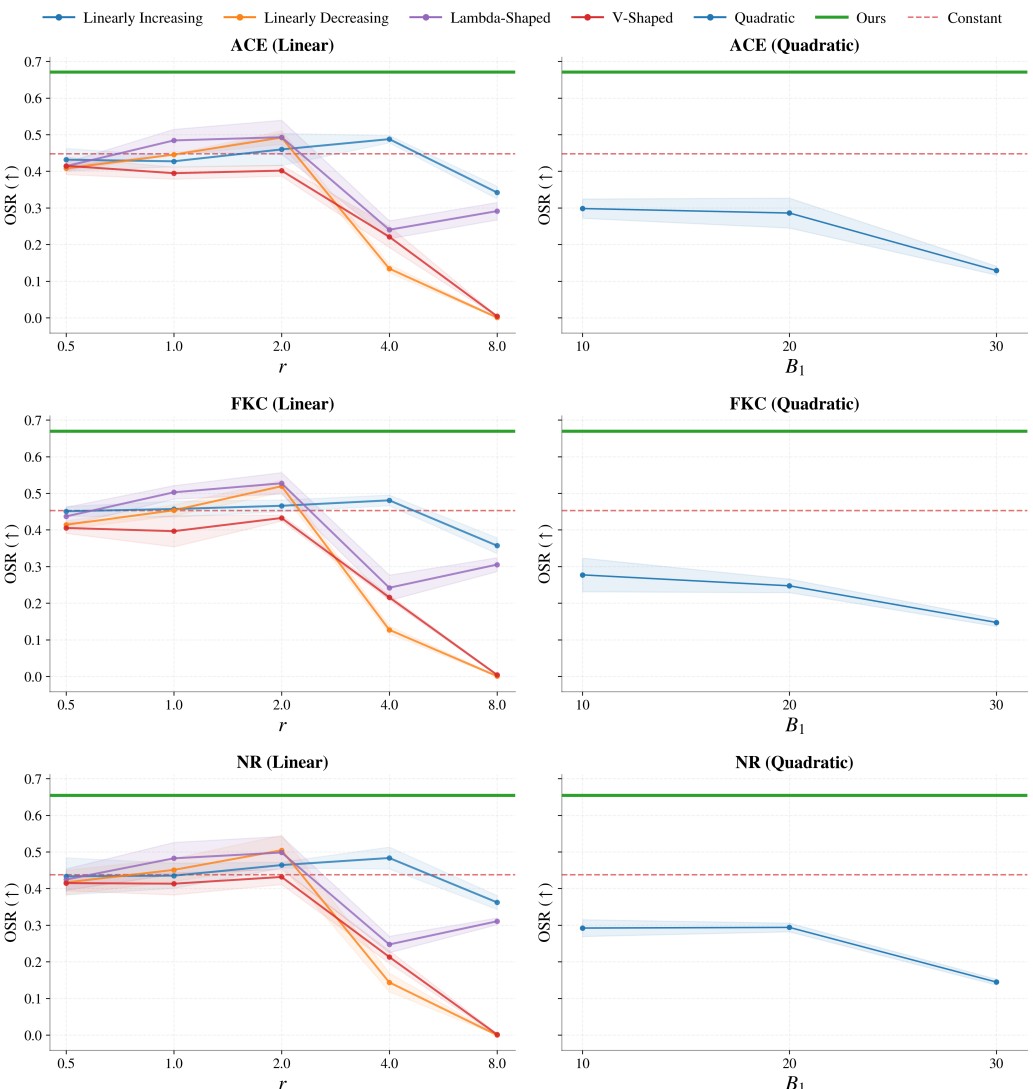

*Figure C.5.* Scaffold decoration OSR (Optimization Success Rate) of ours (green) versus time-varying exponent baselines.

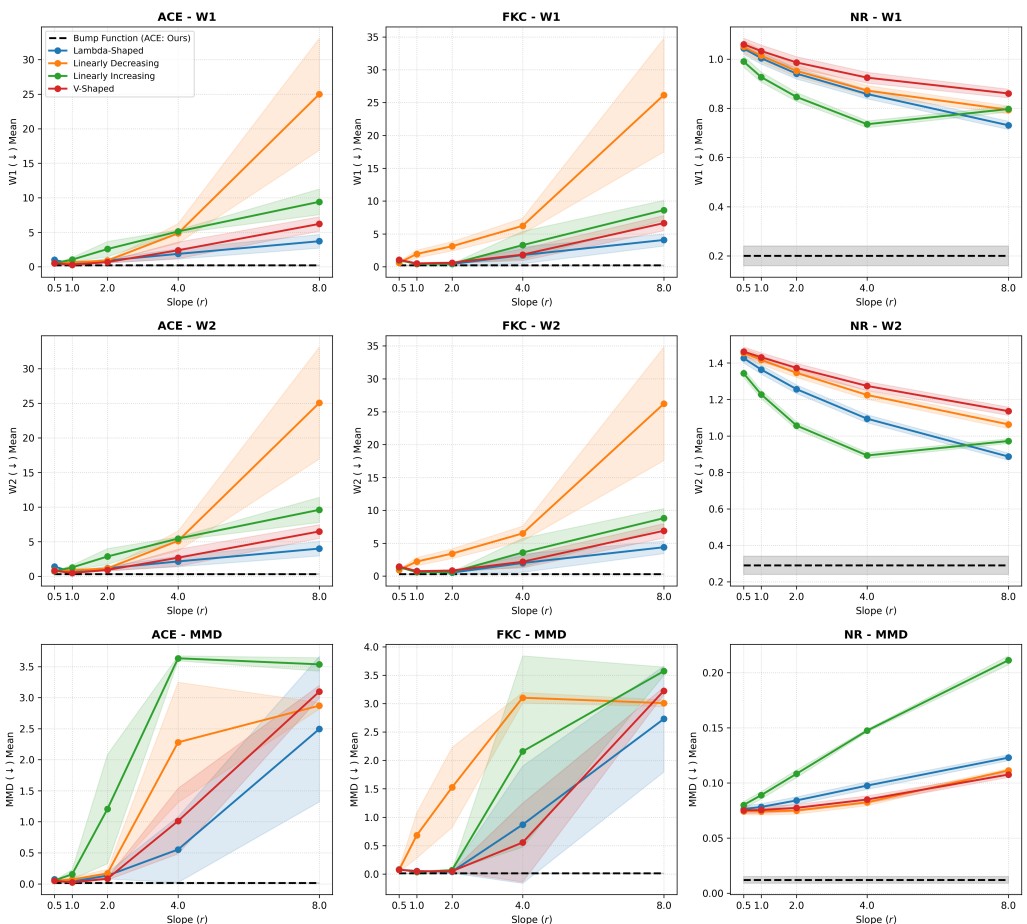

*Figure C.6.* Distributional similarity metrics for the checker distribution of ours (dotted line) versus time-varying baselines.

*Table C.4.* ACE Sensitivity Analysis: $W_2$ ($\downarrow$)

| $B_1 \setminus B_2$ | 0.0 | 0.5 | 1.0 | 1.5 | 2.0 | 2.5 | 3.0 | 4.0 | 8.0 | 16.0 |
|---|---|---|---|---|---|---|---|---|---|---|
| 0.0 | 1.46 $\pm0.92$ | 0.74 $\pm0.45$ | 1.31 $\pm0.99$ | 1.00 $\pm0.17$ | 0.50 $\pm0.10$ | 0.73 $\pm0.35$ | 0.96 $\pm0.07$ | 0.86 $\pm0.20$ | 0.85 $\pm0.13$ | 2.95 $\pm0.67$ |
| 10.0 | 0.90 $\pm0.02$ | 0.67 $\pm0.09$ | 0.45 $\pm0.07$ | **0.29** $\pm0.05$ | 0.31 $\pm0.06$ | 0.70 $\pm0.07$ | 0.66 $\pm0.06$ | 0.68 $\pm0.20$ | 0.87 $\pm0.28$ | 1.95 $\pm0.55$ |
| 20.0 | 0.47 $\pm0.23$ | 0.37 $\pm0.09$ | 0.39 $\pm0.06$ | 0.54 $\pm0.31$ | 0.47 $\pm0.10$ | 0.64 $\pm0.12$ | 0.62 $\pm0.07$ | 0.70 $\pm0.16$ | 0.80 $\pm0.17$ | 1.52 $\pm0.38$ |
| 30.0 | 0.45 $\pm0.13$ | 0.50 $\pm0.21$ | 0.47 $\pm0.15$ | 0.46 $\pm0.10$ | 0.57 $\pm0.13$ | 0.47 $\pm0.16$ | 0.47 $\pm0.18$ | 0.63 $\pm0.15$ | 1.02 $\pm0.28$ | 1.65 $\pm0.63$ |
| 40.0 | 0.63 $\pm0.18$ | 0.60 $\pm0.12$ | 0.57 $\pm0.13$ | 0.65 $\pm0.16$ | 0.67 $\pm0.14$ | 0.72 $\pm0.14$ | 0.77 $\pm0.17$ | 0.94 $\pm0.55$ | 1.00 $\pm0.32$ | 1.69 $\pm0.48$ |
| 50.0 | 0.74 $\pm0.26$ | 0.76 $\pm0.17$ | 0.79 $\pm0.18$ | 0.87 $\pm0.29$ | 0.83 $\pm0.11$ | 0.94 $\pm0.21$ | 0.78 $\pm0.12$ | 0.87 $\pm0.18$ | 1.08 $\pm0.19$ | 1.51 $\pm0.42$ |
| 100.0 | 1.14 $\pm0.09$ | 1.16 $\pm0.23$ | 1.38 $\pm0.47$ | 1.06 $\pm0.09$ | 1.24 $\pm0.14$ | 1.23 $\pm0.18$ | 1.22 $\pm0.21$ | 1.29 $\pm0.19$ | 1.97 $\pm0.68$ | 2.40 $\pm0.37$ |

*Table C.5.* ACE Sensitivity Analysis: MMD-RBF ($\downarrow$)

| $B_1 \setminus B_2$ | 0.0 | 0.5 | 1.0 | 1.5 | 2.0 | 2.5 | 3.0 | 4.0 | 8.0 | 16.0 |
|---|---|---|---|---|---|---|---|---|---|---|
| 0.0 | 0.305 $\pm0.486$ | 0.069 $\pm0.065$ | 0.297 $\pm0.632$ | 0.092 $\pm0.027$ | 0.023 $\pm0.006$ | 0.064 $\pm0.047$ | 0.095 $\pm0.008$ | 0.084 $\pm0.034$ | 0.130 $\pm0.043$ | 2.030 $\pm1.049$ |
| 10.0 | 0.056 $\pm0.008$ | 0.034 $\pm0.009$ | 0.017 $\pm0.003$ | **0.012** $\pm0.003$ | 0.014 $\pm0.002$ | 0.063 $\pm0.011$ | 0.054 $\pm0.009$ | 0.078 $\pm0.037$ | 0.164 $\pm0.119$ | 0.799 $\pm0.406$ |
| 20.0 | 0.025 $\pm0.023$ | 0.018 $\pm0.006$ | 0.019 $\pm0.003$ | 0.035 $\pm0.025$ | 0.029 $\pm0.010$ | 0.057 $\pm0.022$ | 0.054 $\pm0.015$ | 0.076 $\pm0.038$ | 0.117 $\pm0.031$ | 0.598 $\pm0.269$ |
| 30.0 | 0.035 $\pm0.018$ | 0.037 $\pm0.031$ | 0.033 $\pm0.016$ | 0.032 $\pm0.005$ | 0.047 $\pm0.018$ | 0.042 $\pm0.019$ | 0.040 $\pm0.018$ | 0.064 $\pm0.034$ | 0.259 $\pm0.140$ | 0.764 $\pm0.575$ |
| 40.0 | 0.051 $\pm0.013$ | 0.048 $\pm0.011$ | 0.053 $\pm0.019$ | 0.072 $\pm0.041$ | 0.070 $\pm0.022$ | 0.072 $\pm0.018$ | 0.098 $\pm0.055$ | 0.199 $\pm0.272$ | 0.178 $\pm0.102$ | 0.845 $\pm0.451$ |
| 50.0 | 0.094 $\pm0.069$ | 0.097 $\pm0.040$ | 0.102 $\pm0.046$ | 0.153 $\pm0.140$ | 0.117 $\pm0.026$ | 0.164 $\pm0.089$ | 0.113 $\pm0.036$ | 0.152 $\pm0.065$ | 0.252 $\pm0.083$ | 0.626 $\pm0.319$ |
| 100.0 | 0.281 $\pm0.056$ | 0.282 $\pm0.117$ | 0.376 $\pm0.206$ | 0.243 $\pm0.036$ | 0.356 $\pm0.084$ | 0.355 $\pm0.115$ | 0.344 $\pm0.127$ | 0.374 $\pm0.110$ | 0.863 $\pm0.712$ | 1.234 $\pm0.452$ |

in relation to each (Table 1).

**Inference-Time Control and Guidance.** A major advantage of diffusion models is their steerability at inference time. Classifier-free guidance (CFG) is a widely used heuristic that mixes conditional and unconditional scores to enhance sample fidelity (Dhariwal & Nichol, 2021; Chung et al., 2025). More recent work has proposed training-free or post hoc steering frameworks, including universal training-free guidance (Ye et al., 2024; Bansal et al., 2023), scalable steering pipelines (Singhal et al., 2025), discrete model steering (Rector-Brooks et al., 2025), refined steering via manifold-constrained guidance (Li et al., 2026) and kernel density (Hu et al., 2025), RL-based steering (Wagenmaker et al., 2025), and energy- or self-guided sampling approaches (Epstein et al., 2023; Yu et al., 2023; Song et al., 2023). These methods often adopt ratio-of-densities formulations (e.g., $p(x \mid c)^w / p(x)^{w-1}$), a powerful primitive that also underlies contrastive decoding in language models (Li et al., 2023a) and controlled generation in discrete diffusion models (Schiff et al., 2025; Hasan et al., 2025). While practically effective, these approaches focus on how to steer (e.g., by mixing scores) but provide no guarantees that the resulting sampling path remains valid when models are combined multiplicatively. Our work is the first to formalize and solve this underlying path-existence problem.

**Theoretical Foundations and Path Sampling Correctors.** The steering of stochastic processes has been studied using the Feynman–Kac (FK) formula (Stoltz et al., 2010), which connects pathwise sampling weights to principled unbiased estimators. This provides the basis for modern corrector methods in diffusion models (Skreta et al., 2025a;b; Mark et al., 2025; Hasan et al., 2025). Feynman–Kac Correctors (FKC) can in principle yield unbiased samples, but their guarantees require restrictive assumptions: (*i*) homogeneity of models with compatible noise schedules and dimensions, and (*ii*) constant exponents in the ratio-of-densities. These assumptions exclude important real-world cases, such as heterogeneous pretrained models or time-dependent negative exponents for denominator annealing. When violated, the FK framework provides no guidance, and *Marginal Path Collapse* occurs. Our work extends the FK formulation by introducing a path-existence criterion for heterogeneous products and incorporating time-varying exponents into an FK-style partial differential equation, enabling robust correction beyond prior limits.

**Time-Dependent Guidance vs. ACE.** Recent works have proposed time-varying guidance schedules to improve sample quality, primarily in the context of image generation (Wang et al., 2024b; Jin et al., 2026; Zhang & Wan, 2025; Koulischer et al., 2025; Kynkäänniemi et al., 2024). Crucially, these approaches are empirical heuristics designed to optimize the fidelity-diversity trade-off within a regime where the path is already valid (homogeneous CFG). In contrast, ACE targets the more general setting of heterogeneous model composition, where constant schedules often yield mathematically undefined (non-integrable) paths. Unlike heuristic schedules, the ACE schedule is derived from the Path Existence Criterion (Theorem 2.1) to strictly enforce integrability. Furthermore, while heuristics tune for visual appeal, ACE utilizes the schedule to control the quantile radius of the distribution (Proposition E.1), theoretically linking schedule intensity to

variance reduction. Thus, ACE subsumes CFG heuristics as a special case while solving the fundamental existence problem in complex steering tasks.

**Composing Pretrained Models for Scientific Discovery.** The ability to combine specialized pretrained models is central in domains like structure-based drug design (SBDD). A researcher may wish to compose a protein-pocket conditioned ligand generator (Guan et al., 2023; Schneuing et al., 2024), a molecule conformer generator (Xu et al., 2022), and a denovo generation model (Hoogeboom et al., 2022) into a single product distribution. Similar needs arise in scaffold decoration and fragment-based pipelines (Tan et al., 2022; Hu et al., 2023; Liu et al., 2025). However, naive multiplicative composition can trigger path collapse, making the generative process invalid. Our ACE framework directly enables robust multi-expert composition by guaranteeing path existence and providing adaptive exponent scheduling. This transforms composition from an unreliable heuristic into a principled, verifiable procedure. Applications to SBDD and scaffold decoration are especially compelling, as contemporary diffusion methods (Peng et al., 2022; Gao et al., 2024; Huang et al., 2024c;b) often assume fixed poses or homogeneous conditions—assumptions ACE can relax.

### D.1. Common Steering Objectives as Ratio-of-Densities Paths.

Many standard inference-time control techniques can be unified under the ratio-of-densities framework, where the target density takes the form $p^\star(x) \propto \prod_i q^{(i)}(x)^{\gamma_i}$. Along the generative trajectory, this induces a time-indexed family $p_t^\star(x) \propto \prod_i q_t^{(i)}(x)^{\gamma_i}$, whose intermediate densities must remain normalizable for valid sampling. **Marginal Path Collapse** is the violation of this assumption.

**Classifier-free guidance (CFG):** To enhance samples consistent with condition $y$, CFG reweights the conditional model relative to the unconditional one, $p^\star(x) \propto p_\theta(x \mid y)^\gamma p_\theta(x)^{1-\gamma}$, implying the path $p_t^\star(x) \propto q_t(x \mid y)^\gamma q_t(x)^{1-\gamma}$ with score $\gamma \nabla \log q_t(x \mid y) + (1 - \gamma)\nabla \log q_t(x)$.

**Product-of-experts:** To enforce multiple constraints $c_1, \ldots, c_K$, one multiplies separate experts $q^{(k)}(x) = p_\theta(x \mid c_k)$, yielding $p^\star(x) \propto \prod_{k=1}^K q^{(k)}(x)$ and the path $p_t^\star(x) \propto \prod_{k=1}^K q_t^{(k)}(x)$.

**Reward-tilted / RL-style guidance:** Combining a base model $p_\theta(x)$ with a reward $r(x)$ creates a target $p^\star(x) \propto p_\theta(x) \exp(\beta r(x))$. This can be viewed as a product of base density $q_{\text{base}}(x)$ with a reward-density $q_{\text{rew}}(x) \propto \exp(\beta r(x))$, leading to the path $p_t^\star(x) \propto q_t^{\text{base}}(x) q_t^{\text{rew}}(x)$.

**Contrastive / reference model decoding:** To suppress generic patterns from a reference model $p_{\text{ref}}$, the target is defined as $p^\star(x) \propto p_{\text{LM}}(x)^\gamma p_{\text{ref}}(x)^{-\gamma}$, inducing the path $p_t^\star(x) \propto q_t^{\text{LM}}(x)^\gamma q_t^{\text{ref}}(x)^{-\gamma}$.

**Bayesian composition:** When only conditionals $p_\theta(x \mid c_k)$ are available, Bayes yields

$$p(x \mid c_{1:K}) \propto p(x) \prod_k p(c_k \mid x) \propto \underbrace{p(x)}_{\text{prior}} \prod_k \underbrace{p_\theta(x \mid c_k)/p_\theta(x)}_{\text{marginal}}$$

The assumptions and derivation under the heterogeneous conditioning structure is given in Eq. 14. This induces the path $p_t^\star(x) \propto q_t^{\text{prior}}(x) \prod_k q_t^{(k)}(x) q_t^{\text{marg}}(x)^{-1}$ which fits our general ratio-of-densities template.

**Scientific applications.** Our molecular DN/CONF/SBDD compositions for scaffold decoration, protein glue generation, fragment linking are the examples of Bayesian composition; the full derivations are given in Section E.1.

## E. Additional Analyses: Heterogeneous Composition and Path Collapse

### E.1. Heterogeneous Conditioning Structures in Scientific Tasks

**Definition E.1** (Heterogeneous Conditioning Structure). We define a heterogeneous conditioning structure as a property of the **target distribution** $p(X, Y \mid A, B)$ where conditioning variables exert partial or differing constraints on the system components. For instance, condition $A$ may constrain only the subset $X$, while condition $B$ constrains the joint system $(X, Y)$. This structure is distinct from global conditioning (where a condition affects all variables uniformly) and is pervasive in scientific modeling (see Figure E.7).

**Decomposition and Diffusion Steering.** Crucially, these heterogeneous target structures can be decomposed via Bayes' rule (Eq. 14) into products of simpler marginal or conditional distributions. This decomposition is powerful because it allows complex targets to be modeled by composing separate pretrained experts. We visualize this abstractly in Figure E.7a,

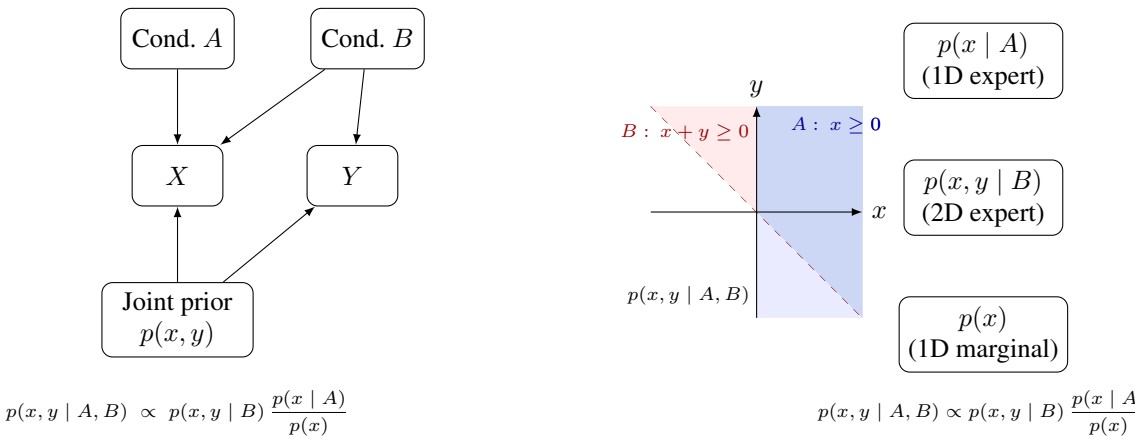

*(a)* Heterogeneous conditioning structure.

*(b)* Synthetic checker: 1D+2D experts.

*Figure E.7.* **Illustration of heterogeneous conditioning structure and its decomposition.** (a) Heterogeneous conditioning structure: condition $A$ acts only on $X$, condition $B$ acts on $(X, Y)$, leading to the Bayes factorization $p(x, y \mid A, B) \propto p(x \mid A)\, p(x, y \mid B)/p(x)$. (b) Synthetic checker: $A = \{x \geq 0\}$ constrains only the $x$-axis, while $B = \{x + y \geq 0\}$ constrains the joint plane, motivating a heterogeneous composition of two 1D experts and one 2D expert.

which highlights how the joint prior governs the fusion of heterogeneous signals. For a concrete example, consider the constrained distribution $p(x, y \mid x \geq 0,\ x + y \geq 0)$ in Figure E.7b. It decomposes into

$$p(x, y \mid A, B) \propto p(x, y \mid B)\frac{p(x \mid A)}{p(x)}$$

allowing a heterogeneous task to be solved by combining a joint expert $p(x, y|B)$ with a marginal expert $p(x|A)$. While this compositional approach aligns with the principles of diffusion steering (e.g., FKC (Skreta et al., 2025a)), previous frameworks overlook the implications of mixing experts with differing domains and schedules, a "blind spot" that leads to the instabilities discussed in Appendix E.4.

**Prevalence in Molecular Generation.** Heterogeneous conditioning is the norm in molecular discovery. Figures E.8a and E.8b illustrate three critical tasks—*scaffold decoration* (Chen et al., 2025; Ghorbani et al., 2023; Xie et al., 2024), *protein-glue generation* (Mogaki et al., 2019), and *fragment linking* (Igashov et al., 2024)—that fundamentally rely on this structure. In Examples E.1–E.3, we demonstrate that each of these tasks decomposes into a combination of three generative primitives: de novo generation (DN) (Hoogeboom et al., 2022), conformer generation (CONF) (Xu et al., 2022), and structure-based drug design (SBDD) (Schneuing et al., 2024). Consequently, solving these high-value problems requires the ability to faithfully compose these specific model families.

**Common Notation and Primitives.** In the following examples, we represent a molecule with $N$ atoms as a 3D point cloud $\mathcal{M}$ with bond topology $\mathcal{T}$. We denote the target protein pocket by $\mathcal{P}$ (.pdb). The condition of *stable binding* is denoted as $\mathcal{M} \leftrightarrow \mathcal{P}$ (satisfied when docking energy $U^{\text{Dock}} < \tau^{\text{Dock}}$). We utilize three fundamental pretrained model families:

- **DN**: Unconditional *De Novo* generation ($p(\mathcal{M})$).

- **CONF**: Topology-conditioned *Conformer* generation ($p(\mathcal{M} \mid \mathcal{T})$).

- **SBDD**: Pocket-conditioned *Structure-Based Drug Design* ($p(\mathcal{M} \mid \mathcal{P})$).

**Example E.1** (Scaffold Decoration). Scaffold decoration (Xie et al., 2024) is a pivotal strategy in lead optimization, enabling the refinement of physicochemical properties and potency while preserving the biological activity of a validated core structure. The goal is to generate an R-group side chain $\mathcal{M}^{\text{R}}$ optimized for binding to a pocket $\mathcal{P}$, while preserving a conserved scaffold backbone $\mathcal{M}^{\text{sc}}$ with fixed topology $\mathcal{T}^{\text{sc}}$ (SMILES).

**Formulation.** The task is to sample a molecule $\mathcal{M} = (\mathcal{M}^{\text{sc}}, \mathcal{M}^{\text{R}})$ from:

$$(\mathcal{M}^{\text{sc}}, \mathcal{M}^{\text{R}}) \sim p(\mathcal{M}^{\text{sc}}, \mathcal{M}^{\text{R}} \mid \mathcal{T}(\mathcal{M}^{\text{sc}}) = \mathcal{T}^{\text{sc}}, (\mathcal{M}^{\text{sc}}, \mathcal{M}^{\text{R}}) \leftrightarrow \mathcal{P}). \tag{E.1}$$

This exhibits the heterogeneous conditioning structure $p(X, Y \mid A, B)$ where $X = \mathcal{M}^{\text{sc}}$ (constrained by topology $A$) and $Y = \mathcal{M}^{\text{R}}$ (where the joint system is constrained by binding $B$).

**Decomposition.** Applying Bayes' rule, we decompose the target into a ratio of experts:

$$p^*(\mathcal{M}^{\text{sc}}, \mathcal{M}^{\text{R}}) \propto \frac{p(\mathcal{M}^{\text{sc}} \mid \mathcal{T}(\mathcal{M}^{\text{sc}}) = \mathcal{T}^{\text{sc}}) \, p(\mathcal{M} \mid \mathcal{M} \leftrightarrow \mathcal{P})}{p(\mathcal{M}^{\text{sc}})}. \tag{E.2}$$

ACE allows us to sample from this using off-the-shelf experts:

$$p(\mathcal{M}^{\text{sc}} \mid \mathcal{T}^{\text{sc}}) \rightarrow \textbf{CONF} \quad \text{(Fixes scaffold geometry)} \tag{E.3}$$
$$p(\mathcal{M}^{\text{sc}}) \rightarrow \textbf{DN} \quad \text{(Prior correction)} \tag{E.4}$$
$$p(\mathcal{M} \mid \mathcal{P}) \rightarrow \textbf{SBDD} \quad \text{(Optimizes binding)} \tag{E.5}$$

---

**Example E.2** (Protein Glue Generation). Protein glues (Mogaki et al., 2019) represent a transformative therapeutic class capable of targeting "undruggable" proteins by inducing novel interactions, such as ubiquitin-mediated degradation. This task requires designing a linker $\mathcal{M}^{\text{linker}}$ that connects two molecular fragments $\mathcal{M}_1^{\text{frag}}, \mathcal{M}_2^{\text{frag}}$ such that they simultaneously bind to two distinct protein surfaces $\mathcal{P}_1, \mathcal{P}_2$. We assume each fragment-specific pocket constraint acts locally through its fragment coordinates, while the global molecular prior couples the full molecule.

**Formulation.** The full molecule $\mathcal{M} = (\mathcal{M}_1^{\text{frag}}, \mathcal{M}_2^{\text{frag}}, \mathcal{M}^{\text{linker}})$ is sampled from:

$$\mathcal{M} \sim p(\mathcal{M} \mid \mathcal{M}_1^{\text{frag}} \leftrightarrow \mathcal{P}_1, \mathcal{M}_2^{\text{frag}} \leftrightarrow \mathcal{P}_2). \tag{E.6}$$

This corresponds to $p(X, Y \mid A, B)$ where $A$ constrains fragment 1 and $B$ constrains fragment 2.

**Decomposition.** We decompose the posterior to isolate the binding constraints:

$$p^*(\mathcal{M}) \propto \frac{p(\mathcal{M}_1^{\text{frag}} \mid \mathcal{P}_1) \, p(\mathcal{M}_2^{\text{frag}} \mid \mathcal{P}_2) \, p(\mathcal{M})}{p(\mathcal{M}_1^{\text{frag}}) \, p(\mathcal{M}_2^{\text{frag}})}. \tag{E.7}$$

This task effectively composes two local binding models with a global prior:

$$p(\mathcal{M}_i^{\text{frag}} \mid \mathcal{P}_i) \rightarrow \textbf{SBDD} \quad \text{(Fragment-specific binding)} \tag{E.8}$$
$$p(\mathcal{M}_i^{\text{frag}}), \, p(\mathcal{M}) \rightarrow \textbf{DN} \quad \text{(Global \& Marginal priors)} \tag{E.9}$$

---

**Example E.3** (Fragment Linking). Fragment linking (Bedwell et al., 2022) is the cornerstone of fragment-based drug discovery (FBDD), enabling the construction of high-affinity ligands by chemically bridging low-affinity fragments bound to adjacent sub-pockets. The goal is to assemble a ligand by connecting two pre-determined fragments $\mathcal{M}_1^{\text{frag}}, \mathcal{M}_2^{\text{frag}}$ (with fixed chemical topologies $\mathcal{T}_1, \mathcal{T}_2$) via a generated linker $\mathcal{M}^{\text{linker}}$, ensuring the final structure binds to a target pocket $\mathcal{P}$. We assume that the local topology constraints on $\mathcal{M}_1^{\text{frag}}, \mathcal{M}_2^{\text{frag}}$ affect the remaining ligand only through the fragment coordinates, while pocket binding acts globally on $\mathcal{M}$.

**Formulation.** The molecule $\mathcal{M}$ is sampled from:

$$\mathcal{M} \sim p(\mathcal{M} \mid \mathcal{T}(\mathcal{M}_1^{\text{frag}}) = \mathcal{T}_1, \, \mathcal{T}(\mathcal{M}_2^{\text{frag}}) = \mathcal{T}_2, \, \mathcal{M} \leftrightarrow \mathcal{P}). \tag{E.10}$$

This represents a complex conditioning structure $p(X, Y \mid A, B, C)$ where $A, B$ enforce local topology and $C$ enforces global binding.

**Decomposition.** The target density factorizes as:

$$p^*(\mathcal{M}) \propto \frac{p(\mathcal{M}_1^{\text{frag}} \mid \mathcal{T}_1)\, p(\mathcal{M}_2^{\text{frag}} \mid \mathcal{T}_2)\, p(\mathcal{M} \mid \mathcal{P})}{p(\mathcal{M}_1^{\text{frag}})\, p(\mathcal{M}_2^{\text{frag}})}. \tag{E.11}$$

ACE constructs this path by coordinating three distinct model families:

$$p(\mathcal{M}_i^{\text{frag}} \mid \mathcal{T}_i) \to \textbf{CONF} \quad \text{(Fragment topology)} \tag{E.12}$$

$$p(\mathcal{M}_i^{\text{frag}}) \to \textbf{DN} \quad \text{(Marginal correction)} \tag{E.13}$$

$$p(\mathcal{M} \mid \mathcal{P}) \to \textbf{SBDD} \quad \text{(Global binding)} \tag{E.14}$$

## E.2. More Use Cases: Fragment Linking inside a Protein Pocket

**Dataset.** We validate our fragment linking framework on the CrossDocked2020 test set (Francoeur et al., 2020). Following the preprocessing protocol used in flexible-pose scaffold decoration, we exclude complexes whose reference ligand contains more than 29 atoms, since this exceeds the predefined maximum supported by the pretrained EDM model. We also remove atoms other than C, O, N, and F due to the limited atom-type support of EDM. For each eligible pocket–ligand complex, we construct a fragment-linking task by extracting two non-overlapping fragment topologies from the reference ligand. Specifically, we apply the same scaffold extraction procedure used in the scaffold decoration experiment twice, where each fragment is defined as a randomly selected ring together with atoms within a bond distance of two from the ring. We retain one valid pocket–fragment-pair per complex when two non-overlapping fragments can be successfully extracted. This procedure yields 66 valid pocket–fragment pairs, which are used for evaluation.

**Implementation Details.** As described in Example E.3, fragment linking inside a pocket aims to generate a molecule $\mathcal{M}$ that contains two given fragment topologies, $\mathcal{T}_1$ and $\mathcal{T}_2$, while binding to the target pocket $\mathcal{P}$. We instantiate this target distribution by composing two fragment-preserving conformer/de novo experts with a pocket-conditioned SBDD model, following the ratio-of-densities formulation in Eq. E.10. Concretely, we use the same pretrained density models as in flexible-pose scaffold decoration: GeoDiff (Xu et al., 2022) as CONF, DiffSBDD (Schneuing et al., 2024) as SBDD, and EDM (Hoogeboom et al., 2022) as DN. DiffSBDD is trained on CrossDocked, while GeoDiff and EDM are trained on QM9 (Ramakrishnan et al., 2014). The two fragment topology constraints are incorporated by applying the fragment-preserving density-ratio term independently to each extracted fragment, while the SBDD model provides the pocket-binding condition.

We use the same ACE sampling configuration as in the scaffold decoration experiment. We perform 500 denoising steps with $\log q$ updates and apply resampling every 10 steps. For the ratio-of-density weight, we evaluate $\omega \in \{1.1, 1.2, 1.3, 1.4\}$ and use the bump function with $B_1 = 30$ and $B_2 = 0.037, 0.136, 0.236$, and $0.336$, respectively. These $B_2$ values are selected to satisfy the prescribed path-existence criterion under the fixed choice of $B_1 = 30$. Since the pretrained diffusion models generate molecules as point clouds, we apply the same post-processing step for bond prediction. The two input fragment topologies are preserved, and additional bonds are inferred based on geometric proximity and valence constraints. We run the fragment-linking experiment with a single random seed. All experiments are conducted using Python 3.8.9 on NVIDIA A6000 GPUs.

**Baselines.** We compare ACE with diffusion-steering baselines that approximate the same composed target density. In particular, we include FKC and NR, where NR corresponds to the non-resampling variant of FKC and is equivalent to a classifier-free-guidance-style simulation without the resampling step. These baselines use the same pretrained component models as ACE, differing only in the sampling and weighting procedure. In addition, we compare against FFLOM (Jin et al., 2023), a task-specific fragment-linking baseline from the literature. FFLOM is included to evaluate the practical advantage of our zero-shot composition framework against a dedicated fragment-linking method.

**Metrics.** We use the same evaluation metrics as in the flexible-pose scaffold decoration experiment. Pocket compatibility is measured by the Vina score computed using QVina2 (Alhossary et al., 2015). When QVina2 fails, for example due to invalid or fragmented molecules, we assign a score of zero. We report *Pocket-Worst*, the average pocket-wise worst Vina score; *Top25%*, the top 25th percentile among all generated molecules; and the mean and standard deviation over all generated molecules. We also report Optimization Success Rate (OSR), defined as the fraction of generated molecules whose docking scores are better than that of the reference ligand. Finally, we evaluate drug-likeness using QED (Bickerton et al., 2012), SA, and the Lipinski score, defined as the number of satisfied Lipinski rules divided by five (Lipinski et al., 1997). Full results

are in Table E.6.

*Table E.6.* Comparison of ACE, FKC, NR and Task-Specific Baseline (FFLOM) in fragment linking. Boldface and underlining indicate the best and second-best values, respectively. Formatting is omitted when all values are identical.

| Method | $\omega$ | PEC | OSR(↑) | Vina Score (↓) | | | | QED (↑) | | | SA (↑) | | | Lipinski (↑) | | |
|---|---|---|---|---|---|---|---|---|---|---|---|---|---|---|---|---|
| | | | | Pocket-Worst | Top25% | Avg | Std | Top25% | Top50% | Avg | Top25% | Top50% | Avg | Top25% | Top50% | Avg |
| ACE | 1.1 | ✓ | **0.46** | -4.07 | **-7.70** | -5.32 | 0.87 | 0.60 | 0.51 | 0.49 | 0.62 | 0.54 | 0.52 | 1.00 | 1.00 | 0.97 |
| | 1.2 | ✓ | 0.45 | -3.95 | **-7.70** | -5.28 | 0.86 | 0.61 | **0.52** | **0.50** | 0.62 | 0.53 | 0.53 | 1.00 | 1.00 | 0.98 |
| | 1.3 | ✓ | **0.46** | -3.96 | -7.40 | -5.24 | 0.87 | 0.59 | 0.48 | 0.49 | 0.60 | 0.52 | 0.52 | 1.00 | 1.00 | 0.97 |
| | 1.4 | ✓ | **0.46** | -3.64 | **-7.70** | -5.18 | 0.96 | **0.64** | **0.52** | **0.50** | 0.62 | 0.53 | 0.53 | 1.00 | 1.00 | 0.97 |
| FKC | 1.1 | ✗ | 0.31 | -3.11 | -7.00 | -4.71 | 1.07 | 0.51 | 0.40 | 0.40 | 0.65 | 0.54 | 0.54 | 1.00 | 1.00 | 0.98 |
| | 1.2 | ✗ | 0.34 | -3.35 | -7.00 | -4.86 | 1.04 | 0.55 | 0.42 | 0.43 | 0.63 | 0.53 | 0.53 | 1.00 | 1.00 | **0.99** |
| | 1.3 | ✗ | 0.30 | -3.45 | -7.10 | -4.87 | 0.97 | 0.51 | 0.40 | 0.40 | 0.62 | 0.54 | 0.53 | 1.00 | 1.00 | 0.98 |
| | 1.4 | ✗ | 0.36 | -3.27 | -7.20 | -4.81 | 1.04 | 0.54 | 0.43 | 0.42 | 0.65 | 0.55 | 0.55 | 1.00 | 1.00 | **0.99** |
| NR | 1.1 | ✗ | 0.37 | -3.28 | -7.40 | -4.89 | 0.99 | 0.56 | 0.43 | 0.41 | 0.58 | 0.51 | 0.52 | 1.00 | 1.00 | 0.98 |
| | 1.2 | ✗ | 0.40 | -3.54 | -7.30 | -5.05 | 0.99 | 0.55 | 0.41 | 0.41 | 0.59 | 0.53 | 0.52 | 1.00 | 1.00 | 0.98 |
| | 1.3 | ✗ | 0.36 | -3.45 | -7.20 | -4.99 | 1.02 | 0.54 | 0.36 | 0.40 | 0.60 | 0.53 | 0.52 | 1.00 | 1.00 | 0.97 |
| | 1.4 | ✗ | 0.33 | -3.62 | -7.30 | -4.92 | 0.96 | 0.52 | 0.40 | 0.40 | 0.59 | 0.51 | 0.51 | 1.00 | 1.00 | 0.98 |
| FFLOM | - | - | 0.24 | **-5.28** | -6.80 | **-5.93** | 0.47 | 0.59 | 0.46 | 0.45 | **0.70** | **0.61** | **0.60** | 1.00 | 1.00 | 0.98 |

### E.3. A Survey on Schedules: The Infeasibility of Homogeneous Composition

**From Heterogeneous Targets to Heterogeneous Paths.** While the heterogeneous *conditioning structure* dictates which expert models must be combined to define the target at $t = 1$, the resulting *probability path* for $t \in (0, 1)$ depends on the noise schedulers of those experts. This creates a systemic challenge: different model families (DN, CONF, SBDD) are historically trained with different noise schedules. We conducted a comprehensive survey of existing diffusion and flow-based models across these domains (summarized in Tables E.7, E.8, and E.9). The results show a lack of standardization, with the literature employing a fragmented mix of sigmoid, polynomial, and cosine schedules. Therefore, constructing generative paths for heterogeneous scientific tasks *inevitably* leads to heterogeneous schedules, where the mismatch in noise contraction rates induces path collapse. This confirms that the collapse phenomenon is not an edge case, but an inherent obstacle arising from the modular nature of scientific generative modeling. We provide a detailed quantitative analysis of this phenomenon, demonstrating its systematic prevalence across standard model combinations and its detrimental impact on sampling, in Appendix E.4.

**The Infeasibility of Homogeneous Composition.** Since Tables E.7–E.9 show that several schedulers appear across tasks, one might wonder whether composing models under a *shared* (homogeneous) scheduler could avoid path collapse. We clarify that using a common schedule across experts is generally infeasible in molecular generation due to two fundamental constraints.

**1. Heterogeneity in Representations.** Unlike images, molecules admit multiple incompatible representations. Some models encode atom types as continuous one-hot vectors $H^{\text{one-hot}} \in \mathbb{R}^N$ evolving under Gaussian convolution; others use categorical variables with discrete states (Dunn & Koes, 2025; Lin et al., 2023) or operate in latent spaces (Ketata et al., 2025; Oestreich et al., 2025). Because these spaces differ in dimensionality, continuity, and semantic meaning, they cannot be embedded into a unified ambient space where a single diffusion path is consistently defined. This fundamentally constrains which experts can be mathematically composed.

**2. The "Empty Intersection" of Schedules.** Even if we restrict our scope to models with compatible continuous representations, a homogeneous triplet often does not exist. In our scaffold decoration experiment, all applicable DN models (EDM, GCDM) employ **quadratic** schedulers, while SBDD models (DiffSBDD, DualDiff) use either quadratic or sigmoid schedules. However, state-of-the-art CONF models do not use the quadratic schedulers but mostly adopt the **sigmoid** schedules. Consequently, it is impossible to construct a homogeneous DN/CONF/SBDD triplet. This structural mismatch necessitates the use of heterogeneous schedules (quadratic for DN/SBDD, sigmoid for CONF), making the path-existence guarantees of ACE a prerequisite for reliable generation.

*Table E.7.* (**CONF**) Diffusion and flow-based models for molecular conformer generation. $N$ denotes the number of atoms. Here, we put the data distribution at $t = 1$, $\beta_t$ controls the data signal while $\alpha_t$ controls the noise signal.

| Model / Paper | Scheduler | Main Datasets | Domain |
|---|---|---|---|
| **GeoDiff** (Xu et al., 2022) | $\alpha_t = \exp\left(-\frac{5}{6}[\log(1 + e^{12(t-\frac{1}{2})}) - \log(1 + e^{-6})]\right)$ 
 $\beta_t = \sqrt{1 - \exp\left(-\frac{5}{3}[\log(1 + e^{12(t-\frac{1}{2})}) - \log(1 + e^{-6})]\right)}$ | GEOM-QM9, GEOM-Drugs | $\mathbb{R}^{3N}$ |
| **TorsionDiff** (Jing et al., 2022) | $\alpha_t = (\frac{\pi}{100})^t \pi^{1-t}$ 
 $\beta_t = 1$ | GEOM-Drugs | $SO(2)^M$ |
| **EC-Conf** (Fan et al., 2024) | $\alpha_t = T(1-t)$ 
 $\beta_t = 1$ | GEOM-QM9, GEOM-Drugs | $\mathbb{R}^{3N}$ |
| **GADIFF** (Wang et al., 2025) | $\alpha_t = \exp\left(-\frac{5}{6}[\log(1 + e^{12(t-\frac{1}{2})}) - \log(1 + e^{-6})]\right)$ 
 $\beta_t = \sqrt{1 - \exp\left(-\frac{5}{3}[\log(1 + e^{12(t-\frac{1}{2})}) - \log(1 + e^{-6})]\right)}$ | GEOM-QM9, GEOM-Drugs | $\mathbb{R}^{3N}$ |
| **ET-Flow** (Hassan et al., 2024) | $\alpha_t = 1 - t$ 
 $\beta_t = t$ | GEOM-QM9, GEOM-Drugs | $\mathbb{R}^{3N}$ |
| **AGDIFF** (Vieira Wyzykowski et al., 2025) | $\alpha_t = \exp\left(-\frac{5}{6}[\log(1 + e^{12(t-\frac{1}{2})}) - \log(1 + e^{-6})]\right)$ 
 $\beta_t = \sqrt{1 - \exp\left(-\frac{5}{3}[\log(1 + e^{12(t-\frac{1}{2})}) - \log(1 + e^{-6})]\right)}$ | GEOM-QM9, GEOM-Drugs | $\mathbb{R}^{3N}$ |
| **LoQI** (Nikitin et al., 2025) | $\alpha_t = \cos\left(\frac{\pi}{2}t\right)$ 
 $\beta_t = \sin\left(\frac{\pi}{2}t\right)$ | ChEMBL3D | $\mathbb{R}^{3N}$ |
| **CoFM** (Xu et al., 2025) | $\alpha_t = 1 - t$ 
 $\beta_t = t$ | GEOM-QM9 | $\mathbb{R}^{3N}$ |

*Table E.8.* (**DN**) Diffusion and flow-based models for de novo molecular generation across coordinate-space, latent-space, and mixed graph–geometry domains. $N$ denotes the number of atoms and $A$, $B$, $C$ the number of atom types, bond types, and formal charge types. $\mathcal{A} = [A], \mathcal{B} = [B]$, and $\mathcal{C}$ are the categorical spaces of $A$ atom types, $B$ bond types, and $C$ formal charge types. Here, we put the data distribution at $t = 1$, $\beta_t$ controls the data signal while $\alpha_t$ controls the noise signal.

| Model / Paper | Scheduler | Main Datasets | Domain |
|---|---|---|---|
| **EDM** (Hoogeboom et al., 2022) | $\alpha_t = \sqrt{1 - (1 - (t-1)^2)^2}$ 
 $\beta_t = 1 - (t-1)^2$ | GEOM-QM9, GEOM-Drugs | $\mathbb{R}^{3N+AN}$ |
| **MiDi** (Vignac et al., 2023) | $\alpha_t = \cos\left(\frac{\pi}{2}t\right)$ 
 $\beta_t = \sin\left(\frac{\pi}{2}t\right)$ | QM9, GEOM-Drugs | $\mathbb{R}^{3N} \times \mathcal{A}^N \times \mathcal{B}^{\frac{N(N-1)}{2}}$ |
| **GCDM** (Morehead & Cheng, 2024) | $\alpha_t = \sqrt{1 - (1 - (t-1)^2)^2}$ 
 $\beta_t = 1 - (t-1)^2$ | GEOM-QM9, GEOM-Drugs | $\mathbb{R}^{3N+AN}$ |
| **EDM-SyCo** (Ketata et al., 2025) | $\alpha_t = \sqrt{1 - (1 - (t-1)^2)^2}$ 
 $\beta_t = 1 - (t-1)^2$ | ZINC250K, GuacaMol | Latent Euclidean Space |
| **DrugDiff** (Oestreich et al., 2025) | $\alpha_t = \sqrt{1 - (1 - (t-1)^2)^2}$ 
 $\beta_t = 1 - (t-1)^2$ | ChEMBL, GuacaMol | Latent Euclidean Space |
| **VEDA** (Zhang et al., 2025) | $\alpha_t = T_{\min}\left(\frac{T_{\max}}{T_{\min}}\right)^{(1-\rho)(1-t)+\rho\frac{2}{\pi}\arcsin\sqrt{1-t}}$ 
 $\beta_t = 1$ | GEOM-QM9, GEOM-Drugs | $\mathbb{R}^{3N} \times \mathcal{A}^N$ |
| **FlowMol3** (Dunn & Koes, 2025) | $\alpha_t = 1 - t$ 
 $\beta_t = t$ | GEOM-Drugs | $\mathbb{R}^{3N} \times \mathcal{A}^N \times \mathcal{B}^{\frac{N(N-1)}{2}} \times \mathcal{C}^N$ |

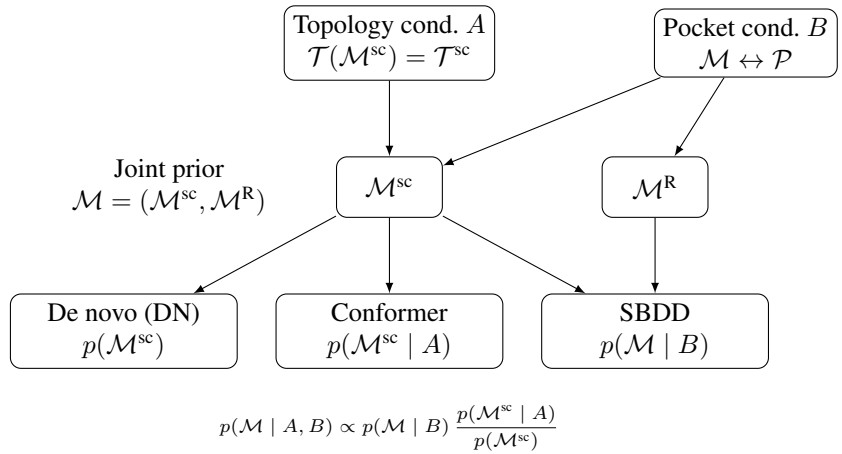

$$p(\mathcal{M} \mid A, B) \propto p(\mathcal{M} \mid B) \frac{p(\mathcal{M}^{\text{sc}} \mid A)}{p(\mathcal{M}^{\text{sc}})}$$

*(a)* Scaffold decoration: DN/CONF/SBDD composition.

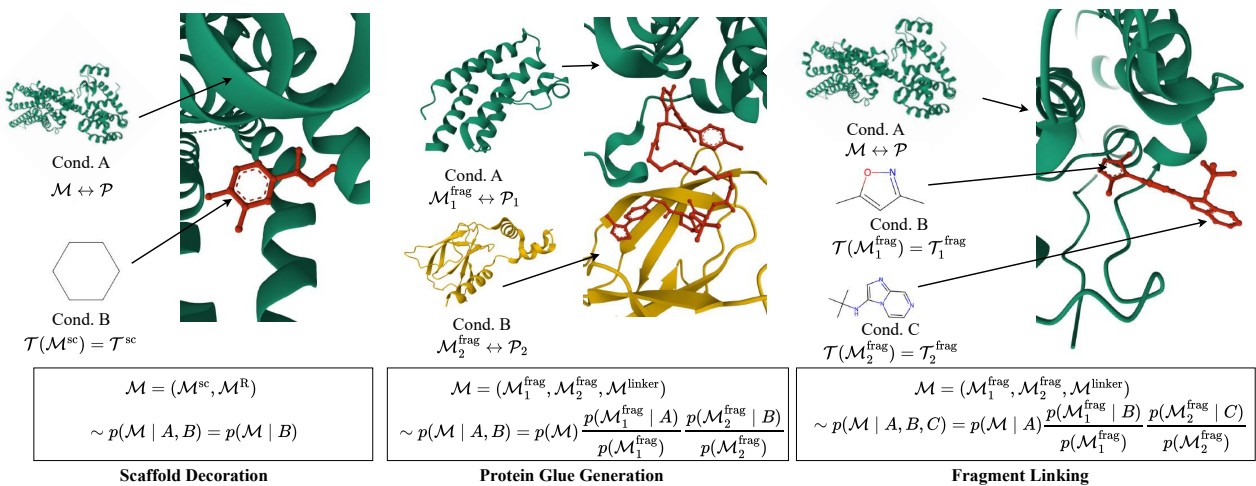

*(b)* Examples in molecular generation: Scaffold Decoration, Protein Glue Generation, Fragment Linking

*Figure E.8.* **Formulation of heterogeneous condition generation and its dominance in molecular generative tasks.** (a) Flexible-pose scaffold decoration mirrors the same structure: topology acts only on the scaffold coordinates, pocket binding acts on the full ligand, and the de-novo model provides the marginal over scaffolds; together they form a DN/CONF/SBDD ratio-of-densities target. (b) Illustration of three molecular generative tasks formulated under the heterogeneous-conditioning framework: scaffold decoration, protein-glue generation, and fragment linking. Each panel specifies the corresponding conditions, target distribution, and the decomposition into expert models for DN, CONF, and SBDD.

*Table E.9.* (**SBDD**) Pocket-conditioned (structure-based drug design) diffusion and flow-based models, including sigmoid, cosine, and quadratic noise schedules across 3D and mixed 3D–categorical domains. $N$ denotes the number of atoms and $A$ the number of atom types. $\mathcal{A} = [A]$ is the categorical space of $A$ atom types. For D3FG, $N_{\text{fg}}$ and $N_{\text{at}}$ denote the numbers of functional-group atoms and remaining atoms, respectively, and $\mathcal{A}_{\text{fg}}$ and $\mathcal{A}_{\text{at}}$ denote the categorical spaces for functional-group types and atom types. Here, we put the data distribution at $t = 1$, $\beta_t$ controls the data signal while $\alpha_t$ controls the noise signal.

| Model / Paper | Scheduler | Main Datasets | Domain |
|---|---|---|---|
| **TargetDiff** (Guan et al., 2023) | $\alpha_t = \exp\left(-\frac{5}{6}[\log(1 + e^{12(t-\frac{1}{2})}) - \log(1 + e^{-6})]\right)$ 
 $\beta_t = \sqrt{1 - \exp\left(-\frac{5}{3}[\log(1 + e^{12(t-\frac{1}{2})}) - \log(1 + e^{-6})]\right)}$ | CrossDocked2020 | $\mathbb{R}^{3N} \times \mathcal{A}^N$ |
| **D3FG** (Lin et al., 2023) | $\alpha_t = \cos\left(\frac{\pi}{2}t\right)$ 
 $\beta_t = \sin\left(\frac{\pi}{2}t\right)$ | CrossDocked2020 | $\mathbb{R}^{3(N_{\text{fg}}+N_{\text{at}})} \times \mathcal{A}_{\text{fg}}^N \times \mathcal{A}_{\text{at}}^N$ |
| **DiffSBDD** (Schneuing et al., 2024) | $\alpha_t = \sqrt{1 - (1 - (t-1)^2)^2}$ 
 $\beta_t = 1 - (t-1)^2$ | CrossDocked2020 | $\mathbb{R}^{3N+AN}$ |
| **BindDM** (Huang et al., 2024b) | $\alpha_t = \exp\left(-\frac{5}{6}[\log(1 + e^{12(t-\frac{1}{2})}) - \log(1 + e^{-6})]\right)$ 
 $\beta_t = \sqrt{1 - \exp\left(-\frac{5}{3}[\log(1 + e^{12(t-\frac{1}{2})}) - \log(1 + e^{-6})]\right)}$ | CrossDocked2020 | $\mathbb{R}^{3N} \times \mathcal{A}^N$ |
| **DualDiff** (Huang et al., 2024a) | $\alpha_t = \exp\left(-\frac{5}{6}[\log(1 + e^{12(t-\frac{1}{2})}) - \log(1 + e^{-6})]\right)$ 
 $\beta_t = \sqrt{1 - \exp\left(-\frac{5}{3}[\log(1 + e^{12(t-\frac{1}{2})}) - \log(1 + e^{-6})]\right)}$ | CrossDocked2020 | $\mathbb{R}^{3N+AN}$ |
| **MolSnapper** (Ziv et al., 2025) | $\alpha_t = \cos\left(\frac{\pi}{2}t\right)$ 
 $\beta_t = \sin\left(\frac{\pi}{2}t\right)$ | CrossDocked2020 | $\mathbb{R}^{3N} \times \mathcal{A}^N$ |
| **PAFlow** (Zhou et al., 2025b) | $\alpha_t = \exp\left(-\frac{5}{6}[\log(1 + e^{12(t-\frac{1}{2})}) - \log(1 + e^{-6})]\right)$ 
 $\beta_t = \sqrt{1 - \exp\left(-\frac{5}{3}[\log(1 + e^{12(t-\frac{1}{2})}) - \log(1 + e^{-6})]\right)}$ | CrossDocked2020 | $\mathbb{R}^{3N} \times \mathcal{A}^N$ |
| **MolFORM** (Huang & Zhang, 2025) | $\alpha_t = 1 - t$ 
 $\beta_t = t$ | CrossDocked2020 | $\mathbb{R}^{3N} \times \mathcal{A}^N$ |

## E.4. Prevalence and Consequences of Marginal Path Collapse

To confirm that Marginal Path Collapse is a systematic vulnerability rather than an edge case, we present a quantitative analysis of its frequency across standard model combinations and examine its downstream impact on generation quality.

**Systematic Prevalence of Collapse.** We evaluated all $5^3 = 125$ annealed compositions of three experts $h_t = q_t^{(1)}(q_t^{(2)}/q_t^{(3)})^w$ formed from five standard noise schedules: DDPM (Ho et al., 2020), cosine (Nichol & Dhariwal, 2021), sigmoid (Xu et al., 2022), linear (Lipman et al., 2023), and polynomial (Hoogeboom et al., 2022). Excluding trivial homogeneous combinations, the results confirm that path collapse is a widespread failure mode. As shown in Table E.10, the collapse rate for heterogeneous compositions starts at **41%** even at a unit guidance scale ($w = 1.0$). Crucially, as the guidance scale increases, as in standard practice to improve sample quality, the collapse rate rises sharply, reaching **66%** at moderate guidance ($w = 2.0$) and **80%** at high guidance ($w = 15$). This indicates that in heterogeneous compositions, operating on invalid paths is the statistical norm, not the exception.

**Empirical Consequences of Collapse.** The violation of the path existence criterion has tangible negative impacts on generation performance, as illustrated in Figures E.7 and E.11–E.16. We observe distinct failure modes across domains:

- **Functional Failure (Molecular Domain):** In scaffold decoration, path collapse correlates directly with the generation of chemically invalid structures. As quantified in Tables 3–4, baselines (NR, FKC) suffer from significantly higher rates of invalid samples when the criterion is violated. Figure E.9 visualizes this stark contrast: ACE produces chemically valid molecules that respect valency constraints, whereas FKC generates fragmented, disconnected structures. While inherent network approximation errors can occasionally yield invalid samples in any method, ACE's failure rate is negligible (maintaining near-perfect validity) compared to the systemic collapse observed in baselines.

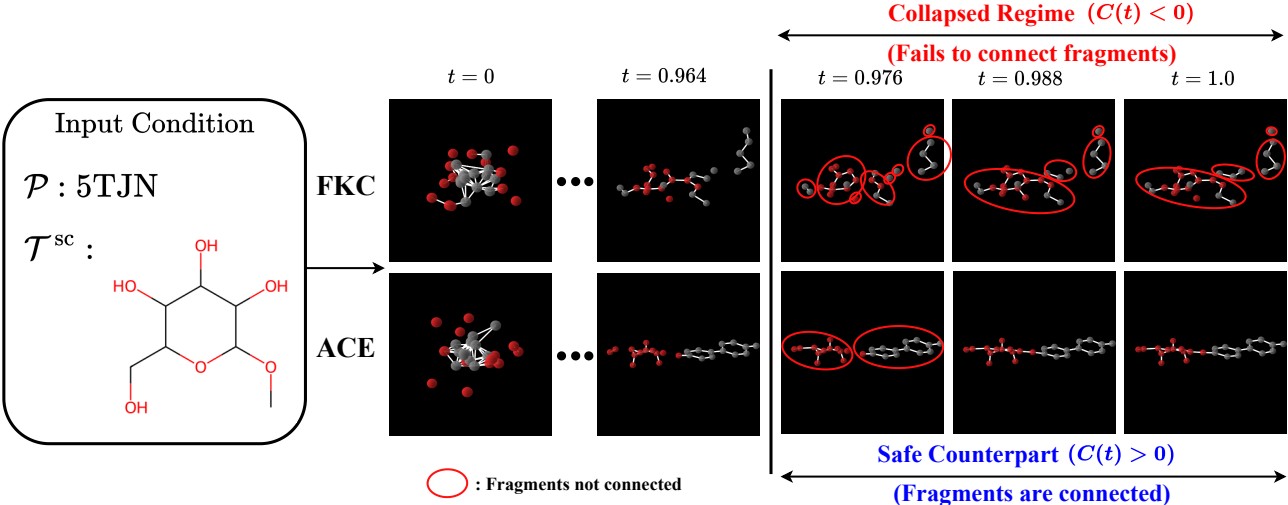

*Figure E.9.* **Impact of Marginal Path Collapse on molecular generation.** We compare sampling trajectories for FKC and ACE on the scaffold-decoration task ($\omega = 1.3$, 500 steps). Both methods target the same distribution $p_\omega(\mathcal{M}) \propto p(\mathcal{M})(p(\mathcal{M} \mid \mathcal{T}^{\text{sc}}, \mathcal{P})/p(\mathcal{M}))^\omega$. Crucially, FKC enters a collapsed regime for $t \in (0.974, 1)$ where the path existence criterion is violated ($C(t) < 0$). This theoretical singularity manifests empirically as a failure to assemble, resulting in disjoint fragments (top row). In contrast, ACE employs the bump correction $b(t) = 30t(1 - t)$ to guarantee $C(t) > 0$ throughout the trajectory, reliably ensuring the successful formation of a connected, chemically valid molecule (bottom row).

*Table E.10.* Frequency of Marginal Path Collapse across 100 heterogeneous schedule compositions. Higher guidance scales $w$ drastically increase the likelihood of encountering invalid paths.

| Guidance Scale ($w$) | # Collapses | % Collapses |
|:---:|:---:|:---:|
| 1.0 | 41 | 41% |
| 1.1 | 47 | 47% |
| 1.5 | 52 | 52% |
| 2.0 | 66 | 66% |
| 7.5 | 77 | 77% |
| 15 | 80 | 80% |

- **Distributional Failure (Synthetic Domain):** On synthetic benchmarks, the consequences manifest as a loss of distributional fidelity (Figure E.11). NR fails to effectively remove out-of-distribution samples due to its heuristic nature, regardless of path collapse (Table E.11 shows non-collapse results). More critically, FKC performance degrades because its importance weights rely on the assumption that the drift term corresponds to the score of a valid probability distribution. When path collapse occurs, the simple mixture of scores, while computationally possible, does not correspond to a score of normalizable distribution. This theoretical violation destabilizes the importance weights, leading to unpredictable behavior such as severe *mode collapse* (see trajectory plots in Figures E.12–E.16).

Marginal Path Collapse is a pervasive practical problem in heterogeneous model composition. Ignoring it leads to measurable degradation in both sample validity and distributional diversity. ACE provides the necessary theoretical and practical fix to guarantee stable, high-quality generation in these heterogeneous regimes.

*Table E.11.* **Metrics under Homogeneous Composition (collapse duration 0%).** Across 5 seeds, We evaluate the scenario where all expert schedules are identical ($\alpha_t^{(i)}$ = DDPM for all $i$), resulting in a collapse duration of 0.0%. ACE criterion is naturally satisfied without correction ($B = 0$), making ACE mathematically equivalent to FKC. Both methods significantly outperform the heuristic NR (CFG) baseline, which suffers from approximation errors even when the path is valid.

| Method | $W_1(\downarrow)$ | $W_2(\downarrow)$ | MMD (RBF) $(\downarrow)$ |
|---|---|---|---|
| NR (CFG) | $0.859 \pm 0.023$ | $1.113 \pm 0.023$ | $0.077 \pm 0.003$ |
| FKC | $\mathbf{0.162 \pm 0.019}$ | $\mathbf{0.282 \pm 0.017}$ | $\mathbf{0.012 \pm 0.001}$ |
| ACE | $\mathbf{0.162 \pm 0.019}$ | $\mathbf{0.282 \pm 0.017}$ | $\mathbf{0.012 \pm 0.001}$ |

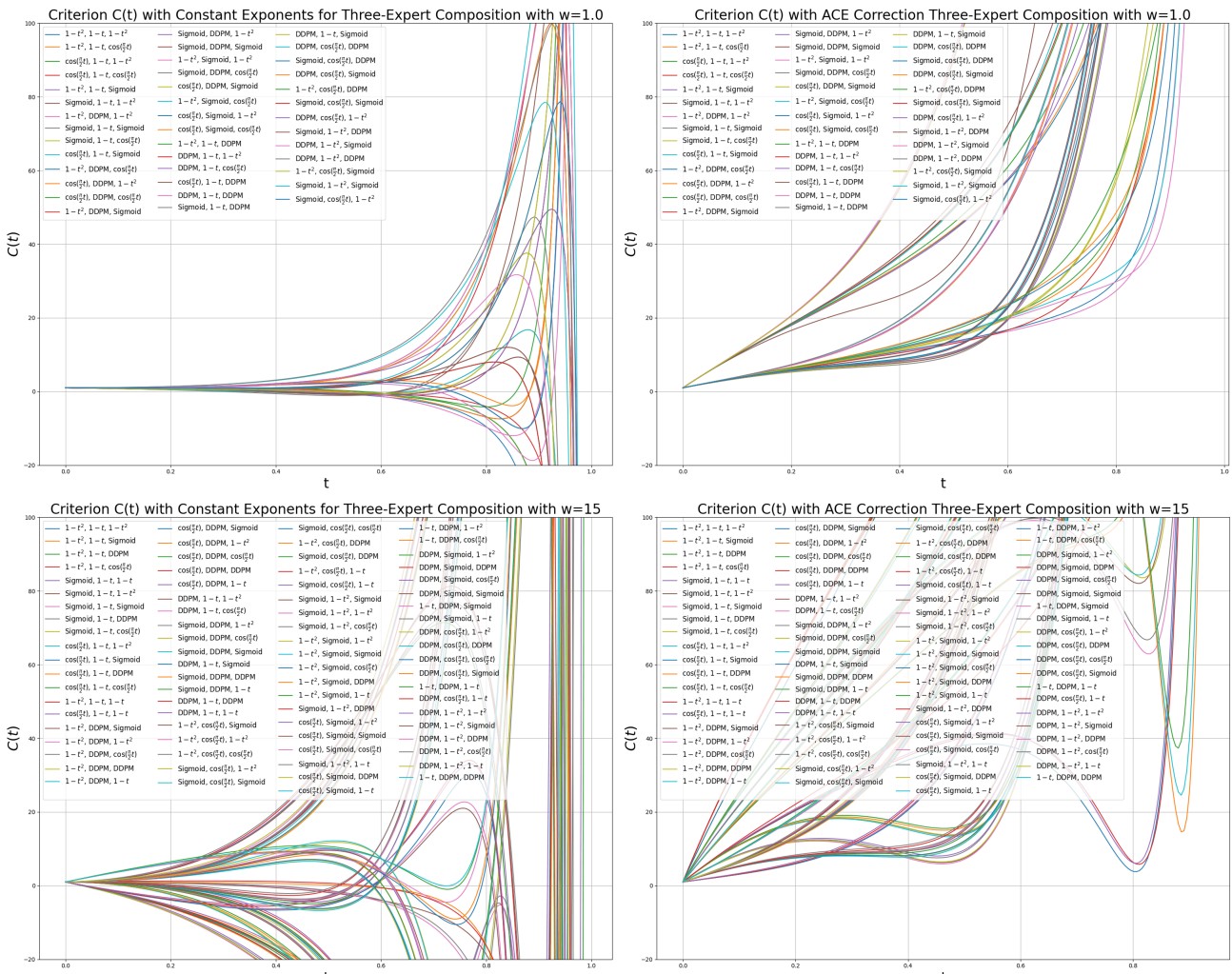

*Figure E.10.* **Path existence criterion $C(t)$ across heterogeneous schedule combinations.** We visualize the evolution of $C(t)$ for the 100 heterogeneous triplets from Table E.10 under low ($w = 1.0$, top) and high ($w = 15.0$, bottom) guidance scales. **Left (Constant Exponents):** A significant fraction of combinations violate the existence condition ($C(t) < 0$), with the frequency and magnitude of collapse increasing sharply as $w$ increases. **Right (ACE):** By applying the adaptive bump correction, ACE guarantees $C(t) > 0$ for all trajectories, restoring valid probability paths even in high-guidance regimes where baselines fail. Note that the y-axis is clipped to $[-20, 100]$ for readability; values are cut off at $t_{\text{end}}$ as practical sampling terminates at $t_{\text{end}} < 1$ (see Theorem B.2).

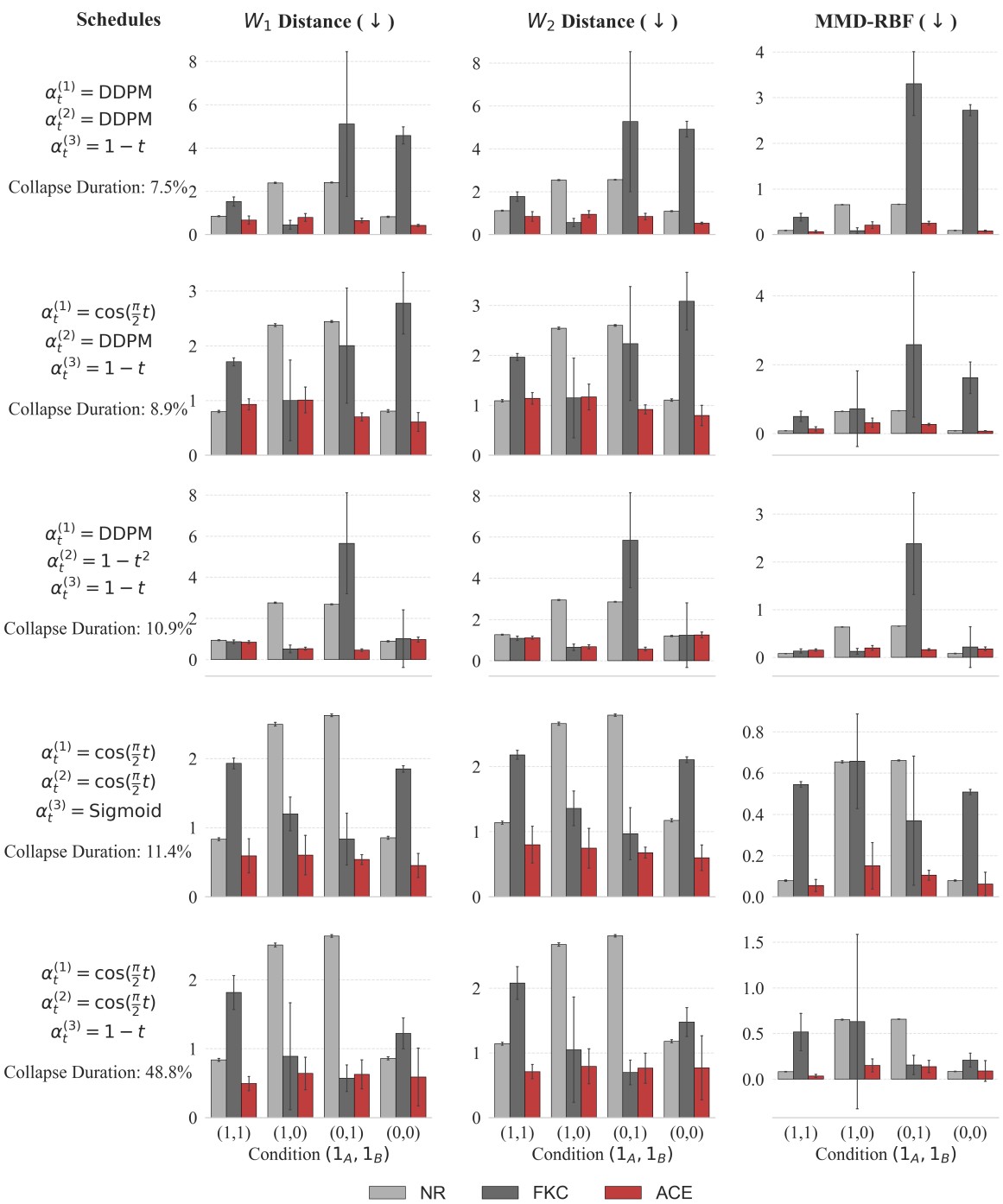

*Figure E.11.* **Quantitative evaluation of Heterogeneous Ratio-of-Densities Sampling.** We evaluate on a checkerboard distribution $(x, y) \sim p_{\texttt{Checker}}[-4, 4]$ subject to constraints $A = \{x \geq 0\}, B = \{x + y \geq 0\}$. The $x$-axis denotes the conditioning configuration $(\mathbf{1}_A, \mathbf{1}_B)$. We consider the case of combining three expert models $q_t^{(1)}, q_t^{(2)}, q_t^{(3)}$ to form the heterogeneous ratio-of-densities $h_t = q_t^{(1)} q_t^{(2)} / q_t^{(3)}$. The rows correspond to configurations of common noise schedules which induce path collapse, where the labels denote the noise schedule $\alpha_t^{(i)}$ assigned to expert $i$. Across varying conditions and metrics $(W_1, W_2, \text{MMD})$, **ACE (red)** consistently outperforms the baselines **NR** and **FKC** (gray). This performance gap reflects the practical benefits of ACE in generating samples across a guaranteed generative path, whereas baselines suffer from path collapse in heterogeneous schedule scenarios. All results are evaluated across 5 seeds with 10k samples, 1k diffusion steps, ESS threshold 0.7, and Bump 30 for ACE. See Figure. E.10 for the corresponding criterion plot.

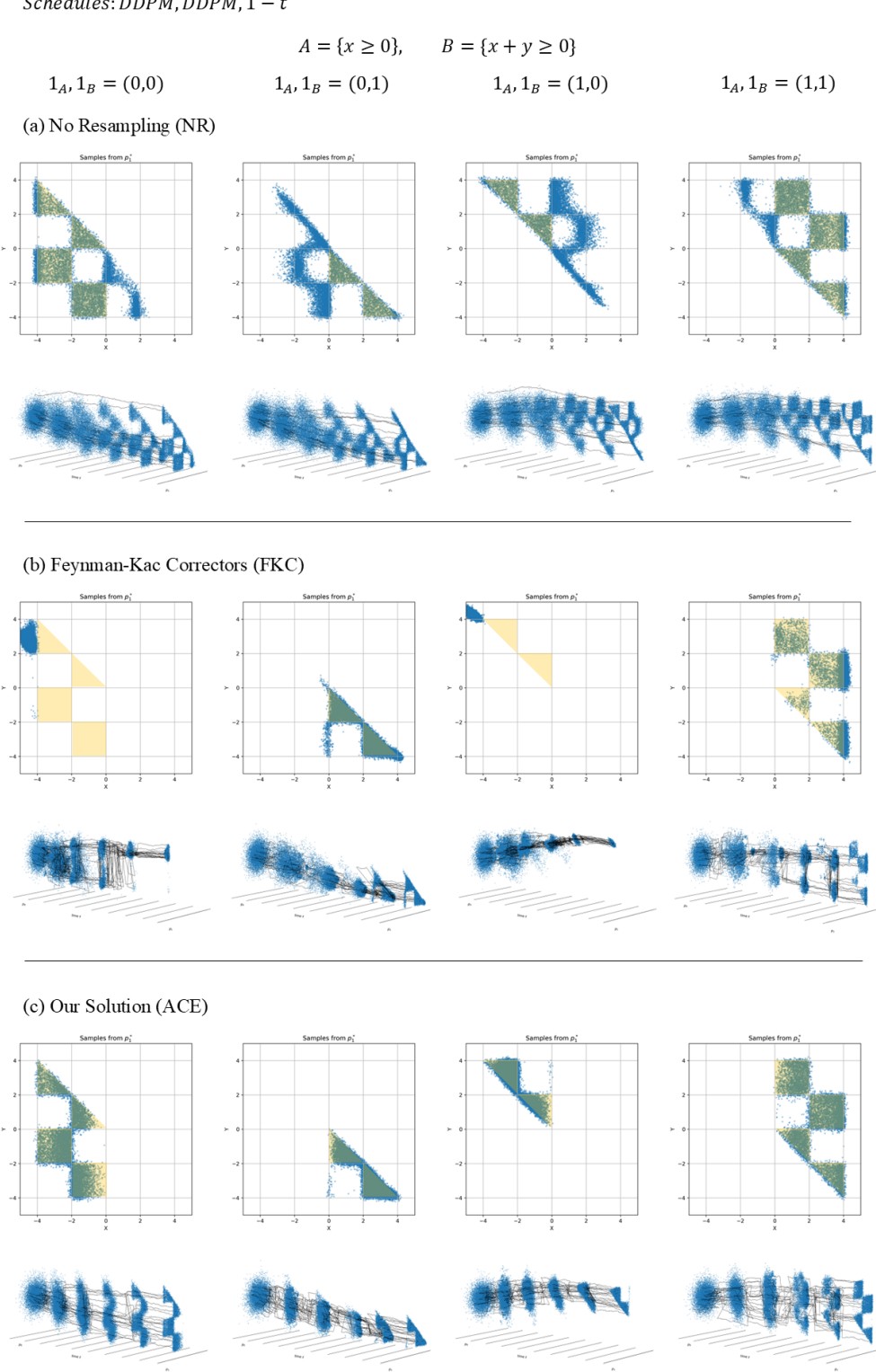

*Figure E.12.* **Visualization of generative trajectories and final samples.** We target the composite density $p^*(x, y) \propto p^{(1)}(x, y \mid B)p^{(2)}(x \mid A)/p^{(3)}(x)$ using the heterogeneous schedule configuration $(\alpha_t^{(1)}, \alpha_t^{(2)}, \alpha_t^{(3)}) = (\texttt{DDPM}, \texttt{DDPM}, 1-t)$ and sampling with NR, FKC, and ACE.

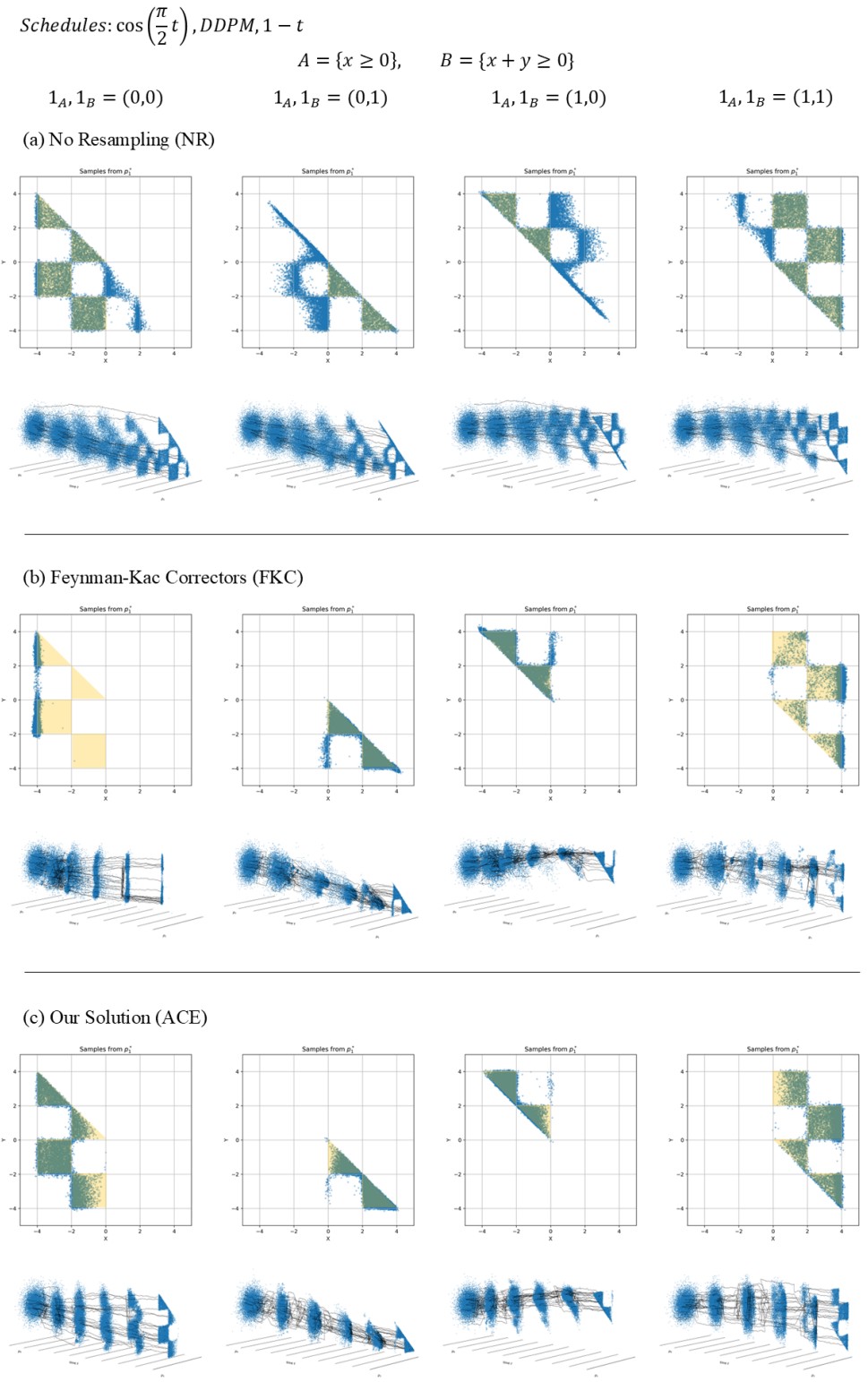

*Figure E.13.* **Visualization of generative trajectories and final samples.** We target the composite density $p^*(x, y) \propto p^{(1)}(x, y \mid B)p^{(2)}(x \mid A)/p^{(3)}(x)$ using the heterogeneous schedule configuration $(\alpha_t^{(1)}, \alpha_t^{(2)}, \alpha_t^{(3)}) = (\cos(\frac{\pi}{2}t), \text{DDPM}, 1 - t)$ and sampling with NR, FKC, and ACE.

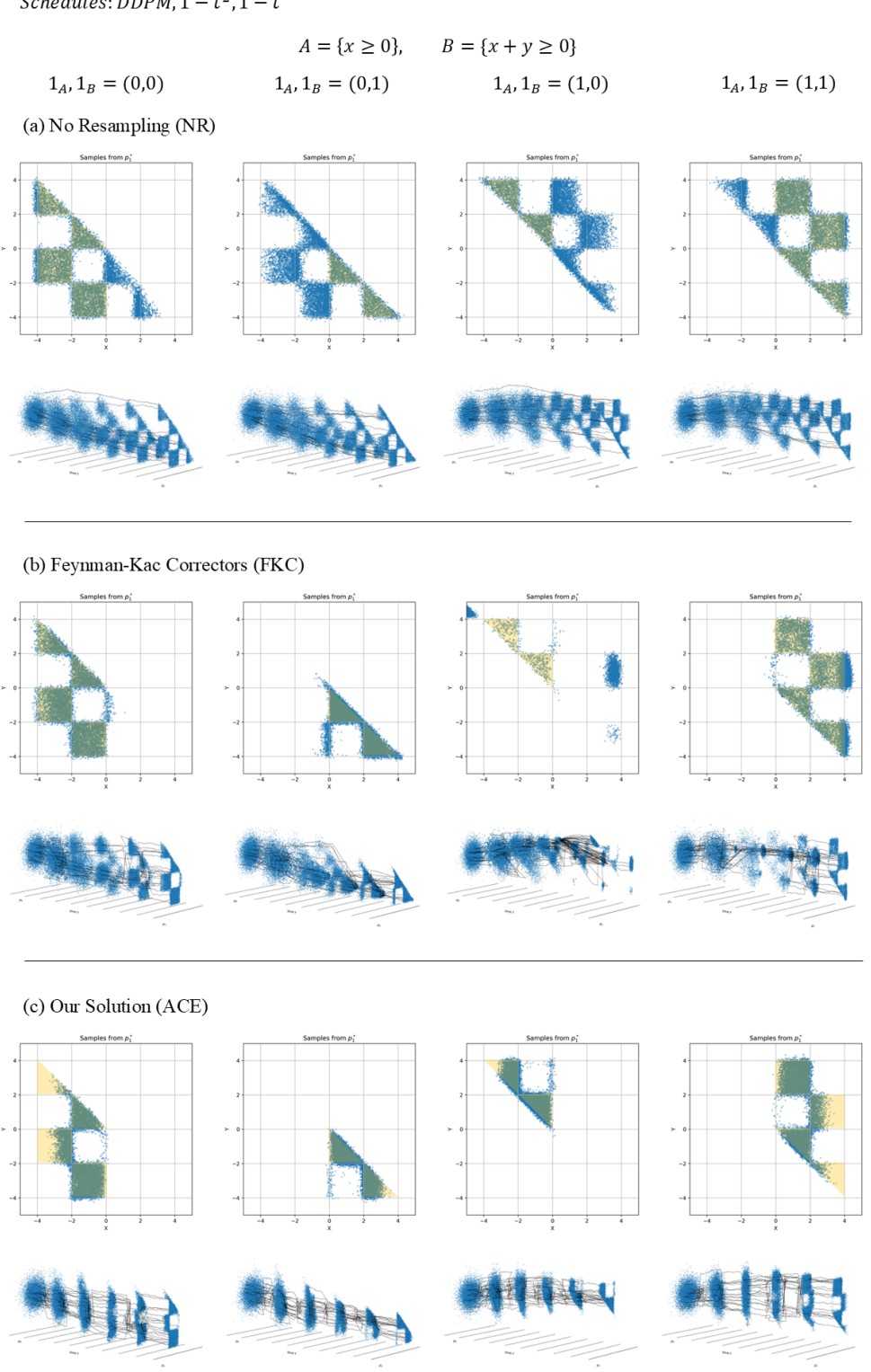

*Figure E.14.* **Visualization of generative trajectories and final samples.** We target the composite density $p^*(x, y) \propto p^{(1)}(x, y \mid B)p^{(2)}(x \mid A)/p^{(3)}(x)$ using the heterogeneous schedule configuration $(\alpha_t^{(1)}, \alpha_t^{(2)}, \alpha_t^{(3)}) = (\text{DDPM}, 1 - t^2, 1 - t)$ and sampling with NR, FKC, and ACE.

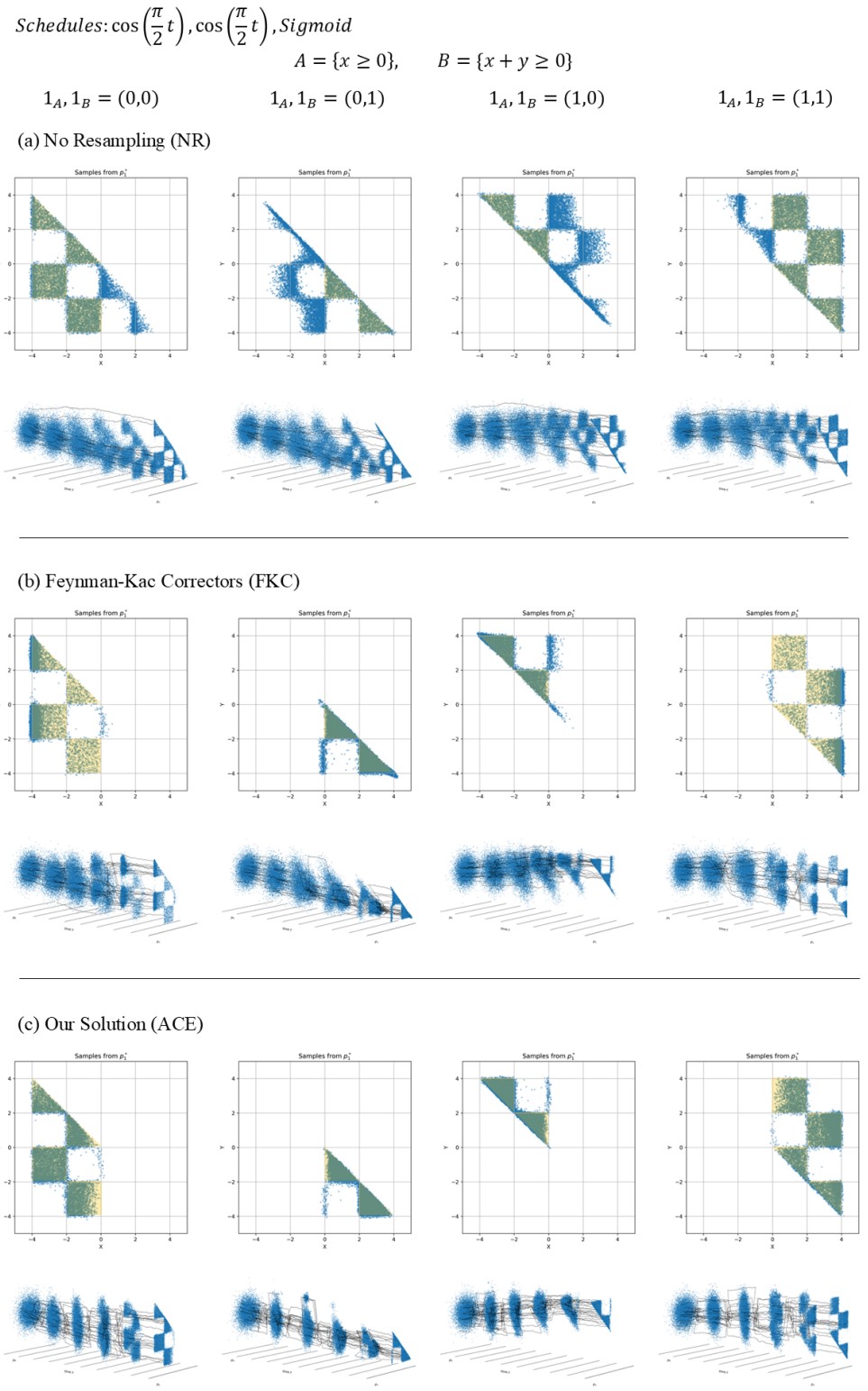

*Figure E.15.* **Visualization of generative trajectories and final samples.** We target the composite density $p^*(x, y) \propto p^{(1)}(x, y \mid B)p^{(2)}(x \mid A)/p^{(3)}(x)$ using the heterogeneous schedule configuration $(\alpha_t^{(1)}, \alpha_t^{(2)}, \alpha_t^{(3)}) = (\cos(\frac{\pi}{2}t), \cos(\frac{\pi}{2}t), \text{Sigmoid})$ and sampling with NR, FKC, and ACE.

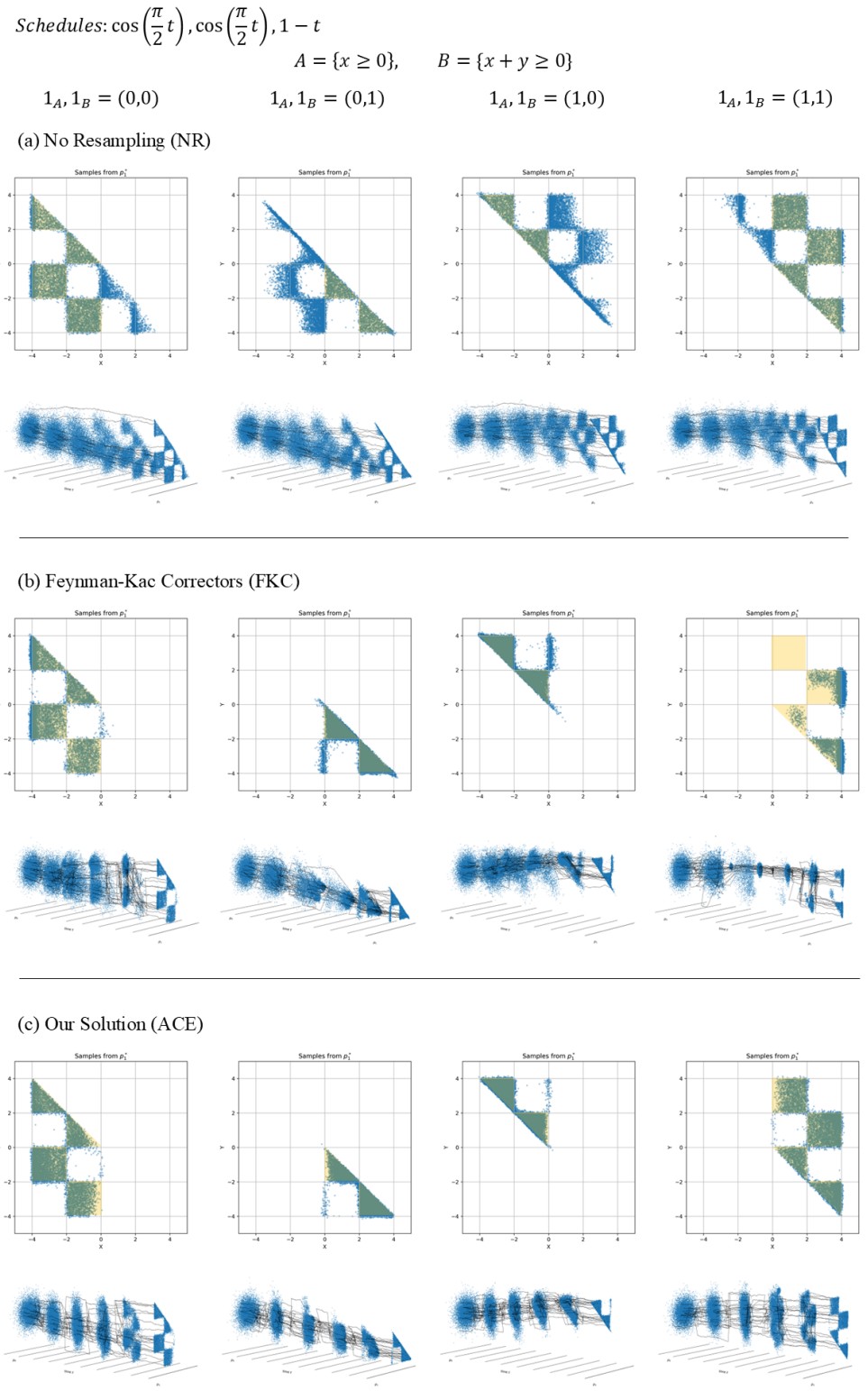

*Figure E.16.* **Visualization of generative trajectories and final samples.** We target the composite density $p^*(x,y) \propto p^{(1)}(x,y \mid B)p^{(2)}(x \mid A)/p^{(3)}(x)$ using the heterogeneous schedule configuration $(\alpha_t^{(1)}, \alpha_t^{(2)}, \alpha_t^{(3)}) = (\cos(\frac{\pi}{2}t), \cos(\frac{\pi}{2}t), 1 - t)$ and sampling with NR, FKC, and ACE.

### E.5. Necessity of Task-Specific Noise Schedulers

A growing body of work on task-specific noise scheduling shows that different generative tasks require different allocations of noise across the denoising trajectory. This follows from the view that diffusion operates in two regimes, high-noise for global exploration and low-noise for local refinement, whose relative importance varies by task. We further validate this trend in molecular generation, and note that recent advances in learnable or adaptive noise schedulers also reinforce this perspective. Together, these observations motivate the use of adaptive or customized noise schedulers, and consequently, heterogeneous scheduler combinations.

*Table E.12.* Comparison of noise regimes and average regime statistics per tasks in Tables E.7,E.8,E.9. The comparison is based on the literature perspective (Choi et al., 2022; Rothchild et al., 2024; Chen, 2023; Pavlova & Wei, 2025). Asterisk(*) indicates significant difference from DN ($p < 0.001$). $|\mathcal{R}_H|$, $|\mathcal{R}_L|$ are the lengths of two regimes.

| (A) Comparison of High-Noise vs. Low-Noise Regimes (Choi et al., 2022; Rothchild et al., 2024; Chen, 2023; Pavlova & Wei, 2025) | | |
|---|---|---|
| **Regime** | **Dynamics** | **Feature Scale Priority** | **Role in Denoising** |
| $\mathcal{R}_H$ **(High-Noise)** | Large Transitions | Global structure | Exploration |
| $\mathcal{R}_L$ **(Low-Noise)** | Small corrective steps | Local fine-grained details | Refinement |

| (B) Average Noise Regime Statistics per Molecular Tasks in Tables E.7,E.8,E.9 | | | | | |
|---|---|---|---|---|---|
| **Task** | **Global vs Local** | **Cond. vs Uncond.** | **Exploration vs Local Refinement** | $|\mathcal{R}_H|$ | $|\mathcal{R}_L|$ | $|\mathcal{R}_H|/|\mathcal{R}_L|$ |
| **DN** | Global | Uncond. | Exploration | 0.53 | 0.47 | 1.11 |
| **CONF** | Local | Cond. | Local Refinement | 0.47* | 0.53* | 0.89* |
| **SBDD** | Local | Cond. | Local Refinement | 0.48* | 0.52* | 0.91* |

**High-noise vs. low-noise regimes.** Recent analyses (Choi et al., 2022; Rothchild et al., 2024; Chen, 2023; Pavlova & Wei, 2025) in Table E.15 characterize two complementary denoising phases. The **high-noise regime** ($\mathcal{R}_H$) performs large transitions that support **global exploration** and formation of coarse structure. The **low-noise regime** ($\mathcal{R}_L$) reduces to **small corrective steps** that refine **local, fine-grained features**. Effective scheduling therefore requires balancing these phases according to task needs. This distinction is summarized in Table E.12(A).

**Empirically validated trends in conditional vs. unconditional molecular generation.** As shown in (Chen, 2023), the roles of $\mathcal{R}_H$ and $\mathcal{R}_L$ indicate that *larger images benefit from a longer $\mathcal{R}_H$*, reflecting their need for stronger exploratory behavior. The insights naturally transfer to molecules. Unconditional generation (DN) needs to explore many possible molecular shapes, so it naturally relies more on the broader, exploratory behavior of $\mathcal{R}_H$. Conditional tasks (CONF, SBDD), however, must fit specific structural requirements, and therefore gain more from the precise, detail-oriented corrections that occur in $\mathcal{R}_L$.

We empirically confirm that existing molecular schedulers adhere to this pattern (Tables E.7, E.8, E.9). Following (Pavlova & Wei, 2025), we divide noise phases using $\text{SNR} \geq 1$ ($\mathcal{R}_H$) and $\text{SNR} < 1$ ($\mathcal{R}_L$). For each model, we compute the interval lengths $|\mathcal{R}_H|$, $|\mathcal{R}_L|$, and their ratio, then average across models per task. The results in Table E.12(B) show clear trends: DN favors $\mathcal{R}_H$, whereas CONF and SBDD prioritize $\mathcal{R}_L$.

These results reinforce our central claim: *heterogeneous noise schedulers are not only reasonable but necessary* for the molecular tasks considered in Section E.4.

**Recent trends in task-specific scheduler design.** Recent works further emphasize that optimal schedules depend on task characteristics (Table E.13). Image models adopt learned or adaptive schedulers (Sahoo et al., 2024), while molecular models employ component-specific or trajectory-based schedules (Vignac et al., 2023; Seo et al., 2025). These methods directly challenge the assumption of a universal, fixed schedule.

### E.6. Comparison with time reparameterization

Motivated by the importance of task-specific noise scheduling, a recent study (Stancevic & Ambrogioni, 2025) proposed a time reparameterization technique that modifies the temporal evolution of the SDE to induce a new sampling path. At a first glance, such reparameterization appears to be a plausible solution for avoiding marginal path collapse, as it seemingly aligns heterogeneous component paths to a shared noise schedule, although this need not preserve the task specific schedules that are optimal for each expert.

*Table E.13.* References on Task-Specific and Adaptive Scheduling

| Reference | Contribution |
|---|---|
| (Vignac et al., 2023) | Introduces component-specific molecular noise scheduling. |
| (Lee et al., 2024) | Presents a fully data-driven adaptive scheduler for time-series diffusion. |
| (Sahoo et al., 2024) | Develops MuLAN: learned multivariate adaptive noise processes. |
| (Seo et al., 2025) | Proposes per-element optimized forward trajectories for molecules. |
| (Sorokin et al., 2025) | Demonstrates adaptive allocation between $\mathcal{R}_H$ (exploration) and $\mathcal{R}_L$ (refinement). |
| (Choi et al., 2025) | Uses distinct channel-wise schedules to balance the diversity–accuracy trade-off. |

*Table E.14.* Quantitative comparison of TRP(Time Reparameterized Path) and ACE.

| Method | $\omega$ | OSR (↑) | | Vina Score (↓) | | | QED (↑) | | | SA (↑) | | | Lipinski (↑) | | |
|---|---|---|---|---|---|---|---|---|---|---|---|---|---|---|---|
| | | P.Worst | Top25% | Avg | Std | Top25% | Top50% | Avg | Top25% | Top50% | Avg | Top25% | Top50% | Avg | |
| FKC-TRP | 1.1 | 0.37 | -4.68 | -7.20 | -5.76 | 0.76 | 0.52 | 0.43 | 0.39 | **0.69** | 0.56 | 0.54 | 1.00 | 1.00 | 0.96 |
| | 1.2 | 0.33 | -4.46 | -7.00 | -5.85 | 0.81 | 0.56 | 0.41 | 0.39 | 0.64 | 0.53 | 0.52 | 1.00 | 1.00 | 0.96 |
| | 1.3 | 0.32 | -4.74 | -7.00 | -5.79 | 0.66 | 0.46 | 0.36 | 0.36 | 0.67 | 0.55 | 0.56 | 1.00 | 1.00 | 0.96 |
| | 1.4 | 0.37 | -5.50 | -7.30 | -6.12 | 0.39 | 0.47 | 0.39 | 0.36 | 0.60 | 0.52 | 0.52 | 1.00 | 1.00 | 0.96 |
| ACE | 1.1 | 0.71 | -6.74 | -8.30 | -7.02 | 0.19 | **0.65** | 0.50 | 0.51 | **0.69** | 0.56 | **0.57** | 1.00 | 1.00 | **0.99** |
| | 1.2 | 0.65 | -6.78 | -8.40 | -7.08 | 0.20 | 0.63 | 0.49 | 0.51 | 0.65 | 0.55 | 0.55 | 1.00 | 1.00 | 0.98 |
| | 1.3 | 0.68 | -6.64 | -8.20 | -6.91 | 0.19 | 0.62 | 0.49 | 0.50 | 0.67 | **0.58** | **0.57** | 1.00 | 1.00 | 0.97 |
| | 1.4 | **0.75** | **-6.84** | **-8.70** | **-7.10** | 0.19 | 0.64 | **0.54** | **0.53** | 0.65 | 0.57 | 0.56 | 1.00 | 1.00 | 0.98 |

However, we empirically demonstrate that composition via time reparameterization (denoted as **FKC-TRP**) leads to degraded performance compared to ACE in terms of accurately approximating the target density. Specifically, in the flexible-pose scaffold decoration task (Section 2.5), Table E.14 shows that FKC-TRP suffers from a substantial drop in both docking affinity (Vina score) and drug-likeness metrics, indicating that it follows a suboptimal generative path. Moreover, increasing the guidance scale to $\omega = 1.4$ rapidly corrupts the output distribution under FKC-TRP, in stark contrast to the consistent performance improvements observed with ACE.

We attribute this behavior to a speed mismatch across heterogeneous noise schedules. In particular, the resulting time reparameterization $\tau(t)$ exhibits non-uniform growth, which can induce accelerated evolution in certain regions of the trajectory. This may exacerbate the degradation of individual paths by skipping task-critical regions and, in turn, destabilize the composed generative path. This behavior is illustrated by the plot of $t' = \tau(t)$ in Figure E.17.

**Implementation details.** We align the component paths $q^{(i)}$ ($i = 1, 2, 3, 4$) in Section 2.5 to a sigmoid noise schedule (Table C.1) via time reparameterization. Concretely, we apply

$$\tau(t) = \sqrt{1 - \sqrt{1 - e^{-\eta(1-t)}}} \, .$$

where

$$\eta(x) = \frac{20}{12} \, \mathrm{softplus}\big(12(x - 0.5)\big) \; + \; 0.001 \, x, \qquad \mathrm{softplus}(z) = \log\big(1 + e^z\big),$$

and define $q_t'^{(i)} = q_{\tau(t)}^{(i)}$ for $i \in \{1, 2, 4\}$, whose original paths follow polynomial scheduling. The remaining component $q^{(3)}$ already employs sigmoid scheduling and is left unchanged, i.e., $q_t'^{(3)} = q_t^{(3)}$. We then compose the reparameterized paths as

$$q_t' = q_t'^{(2)} \cdot \left( \frac{q_t'^{(3)} \, q_t'^{(4)}}{q_t'^{(1)} \, q_t'^{(2)}} \right)^{\omega} \, .$$

For this composed path, we perform Feynman–Kac Corrector (FKC) simulation using a batch size of 5 and 500 integration steps (denoted as FKC-TRP). All other experimental settings are identical to those described in Section C.

**Foundational regime analysis.** Table E.15 compiles key works formalizing these roles. The low-noise regime ($\mathcal{R}_L$) is consistently identified as the phase responsible for *precise, fine-scale refinement*, while the high-noise regime ($\mathcal{R}_H$) drives *exploration and diversity*. These findings directly support task-specific allocations of denoising effort.

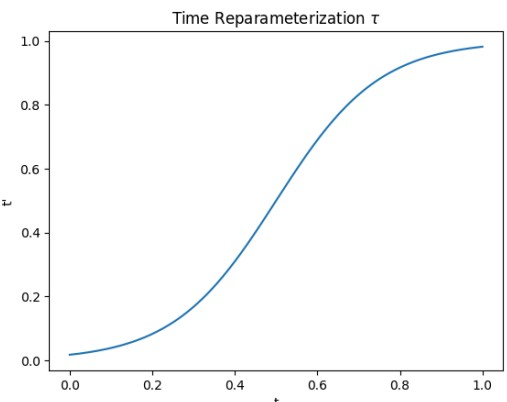

*Figure E.17.* Time reparameterization showing non-uniform time evolution.

*Table E.15.* Foundational References for Noise Regime Mechanism

| Reference | Contribution |
|---|---|
| (Choi et al., 2022) | Identifies that certain noise levels offer a proper pretext task for the model to learn rich visual concepts. |
| (Rothchild et al., 2024) | Anaylze the diffusion dynamics in molecular generative models. |
| (Chen, 2023) | Shows that optimal schedules vary with task and resolution. |
| (Pavlova & Wei, 2025) | Identifies $\mathcal{R}_L$ as the phase in which precision and fine-scale structure dominate. |

### E.7. ACE Beyond Collapse: Compositional Image Generation in the Homogeneous Regime

In the main text, ACE is used primarily to *repair* heterogeneous compositions where the path-existence criterion fails and Marginal Path Collapse occurs. In this section, we show that the same theory also yields *gains beyond collapse avoidance* on a compositional image generation benchmark, even in a strictly homogeneous setting where $C(t) > 0$ everywhere.

**Tail concentration in the homogeneous $\alpha_t$ regime.** Even when the path-existence criterion $C(t) > 0$ holds everywhere (i.e., no collapse), the coefficient $C(t)$ still controls how tightly the composed distribution $p_t^*$ is concentrated. The following result summarizes this dependence and motivates why time-varying exponents can improve sampling quality even in the homogeneous regime.

---

**Proposition E.1** (Tail and quantile control from a quadratic envelope)**.** *Let $\{h_t\}_{t\in[0,1]}$ be a family of nonnegative functions on $\mathbb{R}^d$ and let $p_t^*(x) := h_t(x)/Z_t$ be well-defined with $Z_t := \int_{\mathbb{R}^d} h_t(x)\,dx < \infty$ for all $t \in [0,1]$. Assume there exist constants $m_* > 0$, $K > 0$, $B > 0$ and a function $C : [0,1] \to (0,\infty)$ such that $Z_t \geq m_*$ for all $t$ and*

$$h_t(x) \;\leq\; K \exp\!\Big(-\frac{1}{2}C(t)\|x\|^2 + B\|x\|\Big) \qquad \forall x \in \mathbb{R}^d,\ t \in [0,1].$$

*Then there exist constants $K_0 > 0$ and $c_0 > 0$, depending only on $(d, K, m_*)$, such that for all $t \in [0,1]$ and all $R \geq 0$,*

$$\mathbb{P}_{X\sim p_t^*}\big(\|X\| > R\big) \;\leq\; K_0 \exp\!\Big(\frac{B^2}{C(t)}\Big) C(t)^{-\frac{d}{2}} \exp\!\big(-c_0\, C(t)\, R^2\big). \tag{E.15}$$

*Consequently, for any $\varepsilon \in (0,1)$, the $(1-\varepsilon)$–quantile radius*

$$R_t(\varepsilon) := \inf\{R > 0 : \mathbb{P}(\|X\| \leq R) \geq 1 - \varepsilon\}$$

*satisfies*

$$R_t(\varepsilon) \;\leq\; \sqrt{\frac{1}{c_0\, C(t)}\left[\log\frac{K_0}{\varepsilon} + \frac{B^2}{C(t)} - \frac{d}{2}\log C(t)\right]_+}, \qquad [u]_+ := \max\{u,0\}. \tag{E.16}$$

---

> *In particular, for fixed $(d, \varepsilon)$, larger $C(t)$ yields tighter tails and smaller effective radii (up to logarithmic factors and the $B^2/C(t)$ correction).*

*Proof.* Fix $t$ and abbreviate $C := C(t) > 0$. By the AM-GM inequality, for all $r \geq 0$,

$$Br \ \leq \ \frac{C}{4}r^2 + \frac{B^2}{C} \quad \Rightarrow \quad -\frac{1}{2}Cr^2 + Br \ \leq \ -\frac{C}{4}r^2 + \frac{B^2}{C}.$$

Therefore,

$$h_t(x) \ \leq \ K \exp\left(\frac{B^2}{C}\right) \exp\left(-\frac{C}{4}\|x\|^2\right).$$

Integrating outside a ball and using $Z_t \geq m_*$ gives

$$\mathbb{P}_{X \sim p_t^*}(\|X\| > R) = \frac{1}{Z_t} \int_{\|x\| > R} h_t(x)\, dx \ \leq \ \frac{K}{m_*} \exp\left(\frac{B^2}{C}\right) \int_{\|x\| > R} \exp\left(-\frac{C}{4}\|x\|^2\right) dx.$$

It remains to bound the Gaussian tail integral uniformly. Write $\alpha := C/4$ and use polar coordinates and a constant $S_d$:

$$\int_{\|x\| > R} e^{-\alpha\|x\|^2}\, dx = S_d \int_R^\infty e^{-\alpha r^2} r^{d-1}\, dr = S_d\, \alpha^{-d/2} \int_{\sqrt{\alpha}R}^\infty e^{-y^2} y^{d-1}\, dy.$$

Define $g(u) := e^{u^2/2} \int_u^\infty e^{-y^2} y^{d-1}\, dy$ for $u \geq 0$. Then $g$ is continuous, $g(0) = \int_0^\infty e^{-y^2} y^{d-1} dy < \infty$, and $g(u) \to 0$ as $u \to \infty$ (since the integral decays like $u^{d-2}e^{-u^2}$). Hence $M_d := \sup_{u \geq 0} g(u) < \infty$. Therefore, for all $u \geq 0$,

$$\int_u^\infty e^{-y^2} y^{d-1}\, dy \ \leq \ M_d\, e^{-u^2/2}.$$

Plugging $u = \sqrt{\alpha}R$ yields

$$\int_{\|x\| > R} e^{-\alpha\|x\|^2}\, dx \ \leq \ S_d\, M_d\, \alpha^{-d/2} \exp\left(-\frac{\alpha R^2}{2}\right).$$

With $\alpha = C/4$, this becomes

$$\int_{\|x\| > R} \exp\left(-\frac{C}{4}\|x\|^2\right) dx \ \leq \ S_d M_d\, 4^{d/2}\, C^{-d/2} \exp\left(-\frac{CR^2}{8}\right).$$

Combining all factors, we obtain equation E.15 with $c_0 := 1/8$ and $K_0 := (K/m_*)\, S_d M_d\, 4^{d/2}$. Finally, enforce the RHS of equation E.15 to be at most $\varepsilon$ and solve for $R$; taking $[\cdot]_+$ yields equation E.16. $\qquad\square$

**Application to heterogeneous ratio-of-densities.** In the setting of Theorem B.1, each lifted expert $\tilde{q}_t^{(i)}$ admits upper bounds of the form

$$\tilde{q}_t^{(i)}(x) \ \leq \ C_{+,i} \exp\left(-\tfrac{1}{2} \sum_{k \in I_i} a_{i,k}(t)\, x_k^2 + B_i\|x\|\right),$$

with $a_{i,k}(t) \geq 0$ and constants $C_{+,i}, B_i > 0$ independent of $t$. Multiplying these bounds and raising to the exponents $\gamma_i(t)$ yields

$$h_t(x) \ \leq \ K \exp\left(-\tfrac{1}{2} \sum_{k=1}^d C_k(t)\, x_k^2 + B\|x\|\right),$$

where $K, B > 0$ are uniform constants and $C_k(t)$ are precisely the coordinate-wise coefficients from Equation 4. Defining $C(t) := \min_k C_k(t)$, we have $\sum_k C_k(t) x_k^2 \geq C(t)\|x\|^2$, so the envelope matches the form in Proposition E.1. Theorem B.1 also implies that $h_t \in L^1(\mathbb{R}^d)$ and that the normalizing constants $Z_t$ are uniformly bounded below on any compact set where $C_k(t) > 0$. Therefore, all assumptions of Proposition E.1 hold for our heterogeneous ratio-of-densities path $p_t^* = h_t/Z_t$, and the tail and quantile-radius bounds apply directly with this choice of $C(t)$.

This proposition shows that even when $C(t) > 0$ everywhere (no collapse), the *magnitude* of $C(t)$ still governs how concentrated $p_t^*$ is: for fixed $\varepsilon$, the radius $R_t(\varepsilon)$ shrinks as $C(t)$ grows (up to logarithmic factors). Thus, increasing $C(t)$ at intermediate times (for example, via the time-varying exponents of ACE) tightens the tails of $p_t^*$ and reduces its effective spatial extent. In practice, this suppresses large intermediate spreads, stabilizes importance weights, and improves sample quality, as we observe both in the 1D ratio-of-Gaussians trajectory and image experiment described below.

**Example E.4** (Compositional Image Generation). We consider compositional text-to-image generation with region-wise prompts and bounding boxes. The task is generating an image $X$ conditioned on $n$ region-specific object prompts $c_{1:n}$ (with foreground masks $F_{1:n}$) and a global context prompt $C$. The target distribution factorizes via Bayes' rule into a product of expert likelihoods:

$$p(X \mid c_{1:n}, C) = \frac{1}{Z} \underbrace{p(X \mid C)^{\gamma_0(t)}}_{\substack{\text{Global} \\ \text{Consistency}}} \prod_{i=1}^{n} \underbrace{\left( \frac{p(F_i \mid c_i)}{p(F_i)} \right)^{\gamma_i(t)}}_{\substack{\text{Local} \\ \text{Specifics}}} \tag{E.17}$$

**ACE Implementation with a Single Backbone:** Crucially, we approximate all terms using a *single* pretrained text-to-image model (Stable Diffusion (Rombach et al., 2022) v1.5 / v2.1).

- $p_\theta(X \mid C)$: the base model conditioned on the global prompt.

- $p_\theta(F_i \mid c_i)$: the same model, applied to the cropped region $F_i$ (via masking) with prompt $c_i$.

- $p_\theta(F_i)$: the model on $F_i$ with a null prompt.

All experts share architecture and noise schedule, placing us in the homogeneous regime of Theorem 2.1, where the coefficient $C(t)$ remains strictly positive and no Marginal Path Collapse occurs. In this limit, path existence is guaranteed even for constant exponents.

**ACE vs. FKC and NR in the homogeneous regime.** We apply our generic ACE sampler (Algorithm 1) to this setup by treating each region-specific term in E.17 as an expert and composing their scores and drifts as in Section 2. Since the schedules are homogeneous, ACE does not need to "rescue" a collapsing path. Instead, we use a small bump $B = 5$ in the region exponents $\gamma_i(t)$, which increases $C(t)$ locally at intermediate times while preserving the endpoint distributions. Proposition E.1 implies that this yields a tighter intermediate concentration for $p_t^*$, reducing mass in off-manifold regions and stabilizing importance weights.

We compare three steering schemes using the same backbone, schedules, and hyperparameters[6]:

- **NR (CFG-like):** score-difference heuristic without importance weights.

- **FKC:** constant exponents with Feynman–Kac weighting and resampling (Algorithm 2).

- **ACE:** time-varying exponents with only a quadratic bump $B_1 = 5$ (Algorithm 1).

Because $C(t) > 0$ everywhere, FKC and ACE both operate on valid probability paths; the only difference is the exponent schedule (i.e., constant vs. time-varying).

**Comparison with Existing Paradigms.** Existing approaches to this task generally fall into three categories: (1) Methods that train adapters like GLIGEN (Li et al., 2023b), Make-It-Count (Binyamin et al., 2025); (2) Auxiliary-guided methods like 3DIS (Zhou et al., 2025a) that rely on external signals (depth maps, LLMs); and (3) Architectural interventions such as box-layout guidance (Wang et al., 2024a) or attention-editing techniques (Chefer et al., 2023; Qiu et al., 2025) that modify model's internal signals. In contrast, ACE offers a sampling strategy that achieves unbiased modular composition using a single fundamental expert model without requiring retraining, external data, or architectural changes.

**Empirical results on COCO-MIG.** We evaluate on the COCO-MIG benchmark (Zhou et al., 2024) and report the instance attribute success ratio (%) and mIoU (%) where the instance attribute success ratio measures the fraction of region-wise instructions correctly matched in the generated image. We compare ACE against NR and FKC, multiple diffusion baselines (Rombach et al., 2022; Podell et al., 2023; Esser et al., 2024; Labs, 2024), and task-specific adapters (GLIGEN (Li et al., 2023b), InstanceDiff (Wang et al., 2024a), MIGC (Zhou et al., 2024), 3DIS (Zhou et al., 2025a)). As shown in Table E.16, we observe a consistent hierarchy *NR < FKC < ACE* across backbones: FKC improves substantially over the heuristic NR baseline, and ACE with $B = 5$ further improves both attribute alignment and spatial metrics, despite

---

[6]For the image experiments, it sufficed to resample once at $t_s = 0.3$ due to a small batchsize of $N = 3$.

being applied in a regime with no path collapse. Remarkably, ACE matches or exceeds a task-specific adapter GLIGEN (Li et al., 2023b) on several metrics, while remaining completely training-free.

Qualitatively, Fig. E.18 shows that ACE produces sharply localized objects with correct attributes in their designated boxes, whereas the Stable Diffusion baseline struggles to localize objects or separate attributes. These results provide an empirical complement to Proposition E.1: even when the path-existence criterion is satisfied, carefully shaping the exponent schedule can yield *stronger* conditioning and improved sample quality.

*Table E.16.* Instance attribute success ratio (%) with the corresponding mIoU (%) shown in parentheses. L2–L6 denote the success ratios for tasks with 2 to 6 region-wise guidance conditions. 'CLIP' and 'Local CLIP' refer to CLIP scores computed on the entire image versus the full prompt, and on local crops versus the corresponding local prompts, respectively.

| Method | Backbone | L2 | L3 | L4 | L5 | L6 | Avg | CLIP↑ | Local CLIP↑ |
|---|---|---|---|---|---|---|---|---|---|
| | | | | *Base Diffusion Model with global prompt* | | | | | |
| SD1.5 | SD1.5 | 5.59 (18.83) | 4.79 (17.43) | 2.83 (14.95) | 2.41 (13.93) | 2.21 (15.94) | 3.11 (15.75) | 24.64 | 18.36 |
| SDXL | SDXL | 5.59 (19.78) | 4.48 (18.54) | 2.81 (16.67) | 2.14 (15.72) | 2.80 (18.42) | 3.17 (17.55) | 25.71 | 18.63 |
| SD3.5-M | SD3.5-M | 8.05 (21.57) | 8.46 (21.37) | 6.07 (18.98) | 5.02 (17.39) | 4.40 (17.80) | 5.86 (18.85) | 26.41 | 18.77 |
| Flux.1-dev | Flux.1-dev | 8.83 (22.00) | 7.50 (20.93) | 4.86 (17.75) | 4.53 (16.77) | 3.22 (16.49) | 5.08 (18.03) | 26.17 | 18.56 |
| | | | | *Inference-time Control using **only** the base Diffusion Model* | | | | | |
| NR | SD1.5 | 24.61 (30.50) | 24.19 (29.81) | 19.36 (25.64) | 17.67 (24.42) | 19.54 (25.91) | 20.24 (26.53) | 24.96 | 20.21 |
| FKC | SD1.5 | 33.02 (34.43) | 31.18 (33.34) | 28.80 (31.53) | 25.46 (29.63) | 22.88 (28.61) | 28.27 (31.51) | 25.44 | 20.66 |
| **ACE** ($B_1 = 5, B_2 = 0$) | SD1.5 | 45.31 (41.48) | 42.50 (40.60) | 36.25 (35.20) | 32.75 (32.66) | 32.40 (33.40) | 37.84 (36.67) | 25.59 | 21.18 |
| NR | SD2.1 | 26.52 (31.18) | 25.76 (30.50) | 20.94 (27.24) | 18.83 (24.77) | 18.78 (25.31) | 21.04 (26.93) | 24.89 | 19.96 |
| FKC | SD2.1 | 39.69 (37.78) | 34.93 (35.91) | 31.98 (33.95) | 28.42 (32.07) | 24.93 (30.02) | 31.99 (33.95) | 25.27 | 20.64 |
| **ACE** ($B_1 = 5, B_2 = 0$) | SD2.1 | 46.25 (42.24) | 41.04 (39.45) | 41.72 (37.96) | 38.38 (37.17) | 34.90 (34.82) | 40.46 (38.33) | 24.94 | 21.09 |
| | | | | *Adapter rendering methods (Requires Additional Training on new data or External Models)* | | | | | |
| GLIGEN | SD1.4 | 38.36 (33.96) | 32.79 (29.58) | 28.67 (25.95) | 25.02 (23.88) | 26.98 (24.93) | 28.84 (26.47) | 24.91 | 20.78 |
| InstanceDiffusion | SD1.5 | 68.24 (62.67) | 60.47 (55.75) | 59.88 (54.15) | 53.92 (49.02) | 57.14 (51.34) | 58.49 (53.12) | 25.97 | 21.90 |
| MIGC | SD1.4 | 66.37 (57.02) | 63.10 (54.47) | 61.27 (52.48) | 57.25 (49.49) | 59.13 (51.38) | 60.41 (52.16) | 25.39 | 21.42 |
| 3DIS | SD1.5 | 58.09 (52.76) | 51.48 (46.92) | 46.15 (42.46) | 40.39 (38.16) | 41.22 (38.47) | 45.23 (41.89) | 24.02 | 21.24 |

*Table E.17.* Runtime per image and peak GPU memory usage for generating a single image. Lower is better. Each method was evaluated on an NVIDIA A100 GPU, except 3DIS which was evaluated on an NVIDIA A6000 GPU.

| Method | Backbone | Time (s) ↓ | | | | | | VRAM (GB) ↓ | | | | | |
|---|---|---|---|---|---|---|---|---|---|---|---|---|---|
| | | L2 | L3 | L4 | L5 | L6 | Avg | L2 | L3 | L4 | L5 | L6 | Avg |
| SD1.5 | SD1.5 | 1.48 | 1.48 | 1.54 | 1.52 | 1.52 | 1.51 | 2.64 | 2.64 | 2.64 | 2.64 | 2.64 | 2.64 |
| NR | SD1.5 | 12.02 | 14.03 | 17.58 | 21.95 | 25.10 | 18.14 | 4.64 | 4.65 | 4.65 | 4.65 | 4.86 | 4.69 |
| FKC | SD1.5 | 12.13 | 14.07 | 17.47 | 21.07 | 24.05 | 17.76 | 4.64 | 4.65 | 4.65 | 4.65 | 4.86 | 4.69 |
| **ACE** | SD1.5 | 15.47 | 18.75 | 23.53 | 28.12 | 32.01 | 23.58 | 4.64 | 4.65 | 4.65 | 4.65 | 4.86 | 4.69 |
| SDXL | SDXL | 3.45 | 3.36 | 3.29 | 3.36 | 3.38 | 3.37 | 8.98 | 8.98 | 8.98 | 8.98 | 8.98 | 8.98 |
| SD3.5-M | SD3.5-M | 4.56 | 4.55 | 4.55 | 4.55 | 4.55 | 4.55 | 17.61 | 17.61 | 17.61 | 17.61 | 17.61 | 17.61 |
| Flux.1-dev | Flux.1-dev | 47.73 | 47.69 | 47.71 | 47.71 | 47.71 | 47.71 | 67.64 | 67.64 | 67.64 | 67.64 | 67.64 | 67.64 |
| GLIGEN | SD1.4 | 2.61 | 2.62 | 2.62 | 2.62 | 2.61 | 2.62 | 6.01 | 6.01 | 6.01 | 6.01 | 6.01 | 6.01 |
| InstanceDiffusion | SD1.5 | 6.77 | 8.22 | 9.70 | 11.14 | 12.67 | 9.70 | 6.60 | 6.60 | 6.60 | 6.60 | 6.60 | 6.60 |
| MIGC | SD1.4 | 2.58 | 2.58 | 2.56 | 2.57 | 2.58 | 2.57 | 5.43 | 5.43 | 5.43 | 5.43 | 5.43 | 5.43 |
| 3DIS *(A6000)* | SD1.5 | 10.56 | 10.84 | 11.03 | 11.14 | 11.38 | 10.99 | 7.55 | 7.55 | 7.55 | 7.55 | 7.55 | 7.55 |

# F. Future Directions for ACE

We conclude by outlining limitations and several promising directions for further development of the ACE framework and heterogeneous model composition, both in theory and in practice.

**Error propagation in model composition.** A natural open question concerns the propagation of errors when composing multiple expert models using ACE. In practice, does the modular composition of experts lead to error accumulation that degrades performance compared to training a single, larger network? A systematic error analysis could clarify whether modularity introduces compounding approximation error or whether the benefits of specialization dominate in practical scenarios.

**Extension to arbitrary transport.** We assumed that the stochastic interpolation is between a Gaussian and compactly supported distribution, which is a common formulation in generative modeling. However, tasks such as molecule or image

A photo of a red chair and a yellow chair and a white teddy bear and a brown dining table

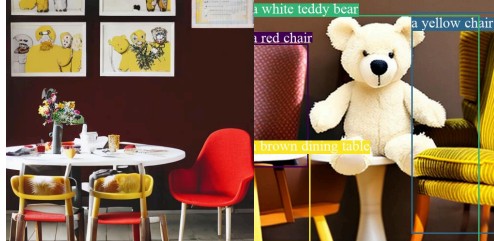

A photo of a black cup and a black donut and a green dining table and a blue donut

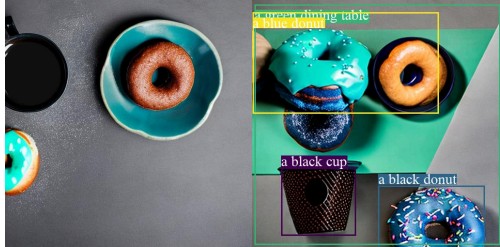

A photo of a blue boat and a red boat and a white boat and a green boat

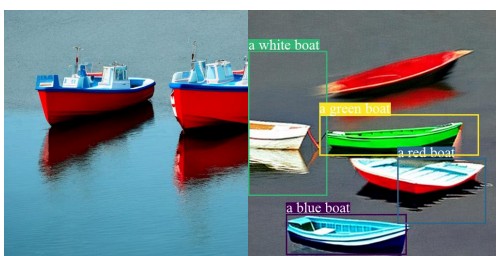

A photo of a white couch and a yellow dining table and a white oven and a brown chair

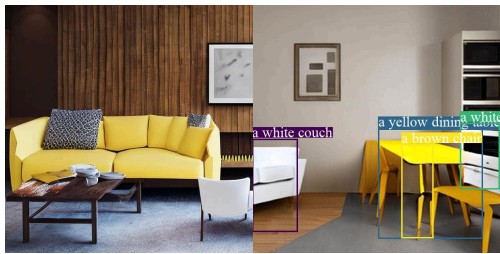

A photo of a brown airplane and a red airplane and a blue airplane

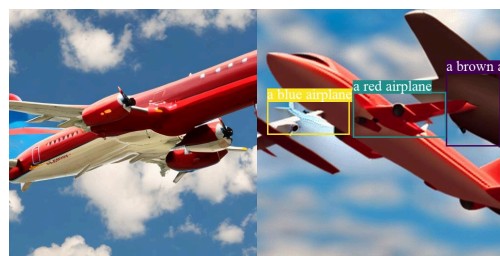

A photo of a white bird and a red boat and a blue boat

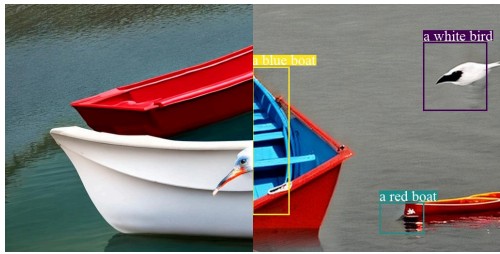

A photo of a blue dog and a white bed and a green tie

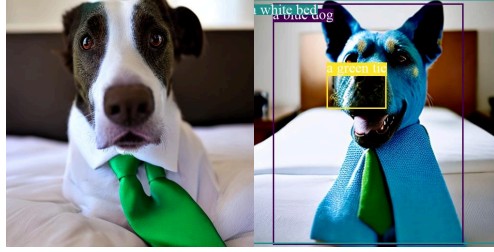

A photo of a black dog and a blue dog and a brown dog

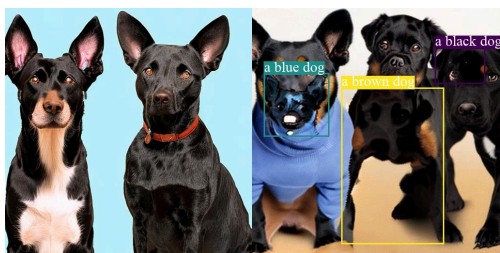

A photo of a white umbrella and a red umbrella

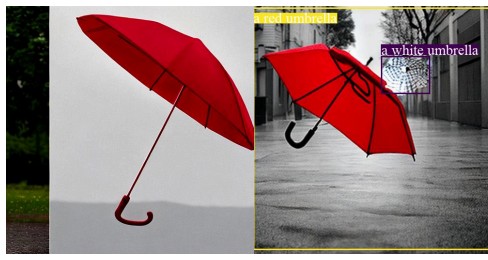

A photo of a brown boat and a brown boat and a yellow boat and a black boat and a white boat and a green boat

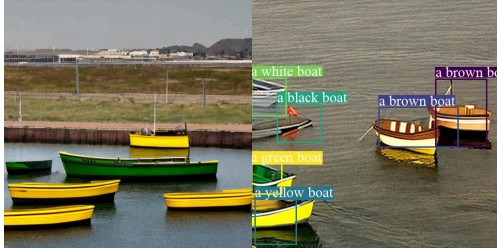

*Figure E.18.* Qualitative results of compositional image generation with ACE. Compared to the base Stable Diffusion 2.1 model (left), simulating the same model with ACE (right) yields better prompt alignment and layout guidance – completely without additional training or external models.

editing, where a sample from the source distribution is given and the task is to transport that sample to a target distribution, require a different formulation since these models interpolate between two compactly supported distributions. Future work could explore when Marginal Path Collapse occurs in these general transport problems.

**Extension to infinite-dimensional spaces.** Our reasoning was kept as general as possible so that it may be naturally extended to infinite-dimensional settings. Viewing each data sample as a function, rather than a point in finite-dimensional space, is both theoretically appealing and practically relevant in certain generative modeling applications. For instance, by replacing $\mu$ in Definition A.1 with a Gaussian measure on a Hilbert space, one can develop a function-space analogue of ACE with direct implications for models defined over functional data.

**Practical efficiency via distillation.** A key limitation of ACE in practice is that the inference cost scales linearly with the number of experts. However, the weighted SDE/ODE formulation of Theorem A.1 suggests that existing techniques for model distillation—such as consistency models, flow map matching, or efficient SDE/ODE solvers—could be adapted to mitigate this cost.

*Remark* (Distillation of Weighted SDE). The probability path $\{p_t^*(X_t)\}_{t \in [0,1]}$, $X_t \in \mathbb{R}^d$ can equivalently be simulated by solving the ODE $Y_0 \sim p_0^* \otimes \delta(\mathbf{1})$, $\quad dY_t = \left( \begin{bmatrix} v_t^* \\ g_t \end{bmatrix} \circ \pi \right)(Y_t)dt$. where $Y_t = \begin{bmatrix} X_t \\ \log w_t \end{bmatrix} \in \mathbb{R}^{d+1}$ and $\pi$ projects $Y_t$ to $X_t$. Applying distillation techniques to this formulation will enable efficient sampling while maintaining fidelity to the target distribution.

**Expanding modular generation to scientific frontiers.** We demonstrated ACE's capacity to resolve path collapse in high-dimensional, multi-modal settings through the scaffold decoration task, where it successfully coordinated distinct modalities (bond topology, protein pocket, and 3D conformation). This success suggests that ACE can extend to even more complex scientific tasks, such as protein-glue generation and fragment linking, by decomposing them into families of existing models (de novo, conformer, and SBDD). Appendix E.2 provides an initial fragment-linking validation, while broader empirical validation–especially for protein-glue generation–remains future work.

**Optimizing exponent schedules for scientific and creative tasks.** Our current ACE schedules use simple bump functions chosen to satisfy the path-existence criterion and, in the homogeneous regime, to modestly increase $C(t)$ at informative times. The COCO-MIG experiments in Appendix E.7 show that even such simple time-varying exponents can significantly improve sample quality over both NR and FKC, despite the absence of path collapse. This suggests a promising direction: treating the exponent schedules $\gamma_i(t)$ themselves as objects to be optimized for downstream objectives such as alignment or diversity. A complementary line of work is to extend ACE to truly heterogeneous creative pipelines that combine distinct backbones and modalities (e.g., global image-editing models with local text-conditioned generators), where incompatible noise schedules would otherwise induce collapse. In both cases, ACE provides the theoretical guarantees needed to explore richer steering schemes without sacrificing validity.

**Extension to hybrid processes in mixed continuous–categorical domains.** As discussed in Appendix E.1, many scientific and molecular generative tasks involve data living in a product space of continuous coordinates (e.g., 3D positions) and categorical variables (e.g., atom and bond types). Current diffusion-based steering methods, including ACE, operate on continuous domains where intermediate densities are well defined under Gaussian interpolation. However, categorical variables do not admit Gaussian smoothing, and thus lack a natural notion of intermediate densities or scores along the diffusion path. This mismatch makes ratio-of-densities steering, and the study of Marginal Path Collapse, substantially more challenging in hybrid domains. Future work could develop principled stochastic interpolants for mixed continuous–discrete representations, identify conditions under which well-defined joint paths exist, and characterize when collapse arises in such hybrid settings. Such an extension would enable theoretically sound steering for full molecular structures, jointly evolving coordinates and atom/bond types, thereby broadening ACE's applicability to richer molecular generation pipelines.

**A long-term vision: Towards a modular generative ecosystem.** ACE demonstrates that complex conditional generation can be achieved through the modular composition of pretrained expert models, offering a scalable alternative to monolithic, task-specific architectures. In the long run, this suggests a paradigm shift toward a standardized practice for generative modeling on continuous domains. Instead of training a bespoke model for every novel application, the community could curate a library of highly efficient expert models for fundamental marginal densities, composing them ad hoc using ACE. This modular paradigm opens the door to a principled framework for the collective intelligence of AI, where distinct models collaborate to solve complex tasks beyond the scope of their individual training.

