# OpenReview forum: "On the Collapse of Generative Paths: A Criterion and Correction for Diffusion Steering"
_ICML.cc/2026/Conference — ICML 2026 regular_

### Official Review · Reviewer_JJVP · 2026-03-07

**Soundness:** 3
**Presentation:** 4
**Significance:** 4
**Originality:** 4
**Overall Recommendation:** 5
**Confidence:** 4

**Summary:**

This paper presents an important failure case, namely Marginal Path Collapse (MPC), in using existing inference-time steering methodologies such as classifier guidance (CFG) and  Feynman-Kac correctors (FKC). The insight is that when considering the heterogeneous raiot-of-densities mixture to construct a path, the path of generation will become invalid when the mixed path $h_t(x)$ is not integrable. In such a case, the sampler would use a different density path rather than the expected path. The paper also introduces a Path Existence Criterion to evaluate whether the intermediate densities are well-defined. Finally, the paper proposes a correction protocol, namely Adaptive path Correction with Exponents (ACE) that solves this issue by generalising FKC with an auxiliary SDE such that the resultant path correctly follows the intended path. The empirical results and toy examples validate the theoretical results and demonstrate the practical usage of the algorithm. Also, the paper provides a detailed discussion on sensitivity analysis for the algorithm, Superiority and efficiency over the baselines.

**Compliance With Llm Reviewing Policy:**

Affirmed.

**Final Justification:**

After the rebuttal, I would like to increase my confidence in my positive score, as my concerns are fully resolved, and I believe this is a good paper with solid, theoretically grounded results, and motivates an interesting research direction in solving the pattern of Marginal Path Collapse (MPC).

**Key Questions For Authors:**

- Please refer to the weakness part and answer the last question there.

- I am curious to see why even the intermediate path for generation is invalid; the sampling could still converge to meaningful samples in CFG and FKC? I see in Appendix A.2 a discussion in computability and validity is made, but I am still slightly confused on which path $p_t^\prime$ the model would lean toward? Can you kindly provide some intuition on this?

- Also, in Figure 2 there exists a non-integrable region. I am wondering why only a partial of the path is non-integrable, but for the other parts it is well-defined?

**Limitations:**

yes

**Strengths And Weaknesses:**

Strengths:

- Overall, the paper provide a solid theoretical results on spotting an issue of MPC and attempting to introduce a novel sampler that correctly follows the intended path. I checked the mathematical results (though more generally), and the theoretical results are solid and sound. The paper presentation is clear and easy to follow, with toy examples that appear where it is needed. The theoretical results are also well-visualized with figures. The empirical results are comprehensive, and the discussion is carefully done.

Weaknessnes:

- One major concern (which may also result in my misunderstanding of the algorithm) is that when introducing ACE Sampling Dynamics, the whole design is based on the assumption that the path existence criterion is satisfied. In such a scenario, the author provides an unbiased sampler for the intended path. From my understanding, this does not solve the problem of the marginal path collapse problem, and previous samplers such as CFG and FKC can still provide valid sampling results. The authors mentioned that using CFG and FKC will still result in an alternative path and give samples. Thus, I am uncertain if the MPC problem is already solvable in this case, or it is just a problem spotted by the authors without a clear solution.

---

> ### Author Rebuttal · Authors · 2026-03-31
>
> We sincerely thank the reviewer for the highly positive recommendation and for recognizing the soundness, clarity, and significance of our theoretical results. Your questions highlight crucial conceptual nuances regarding SDE dynamics and path integrability. We are grateful for the opportunity to clarify them, and we will include these expanded discussions in our camera-ready appendix.
>
> **[W1, Q1] Does ACE solve MPC, or does it merely assume the path is already valid?**
> The reviewer correctly notes that the ACE Sampling Dynamics (Thm 2.3) assumes the Path Existence Criterion (PEC) is satisfied. However, we clarify that our framework does not assume the PEC is *naturally* satisfied; it actively forces it to be satisfied. We provide a complete mathematical solution to Marginal Path Collapse (MPC) via a three-step pipeline:
>
> 1. **Detection (Theorem 2.1):** a diagnostic criterion that certifies when the intended ratio-of-densities path is valid and when it has collapsed.
> 2. **Correction (Theorem 2.2):** The structural fix. It adaptively modifies exponents via a bump function, mathematically guaranteeing the intermediate path remains integrable while perfectly anchoring to the exact target endpoints.
> 3. **Sampling (Theorem 2.3):** The unbiased generalized sampling dynamics for the *corrected* path.
>
> Regarding whether previous heuristic samplers (CFG/NR) can solve MPC: applying them to an *uncorrected* path fails. However, our ACE-lite variant demonstrates that applying standard No Resampling (NR) sampling to the *corrected* path can potentially restore performance. ACE-lite isolates the contribution of Thms 2.1 and 2.2; while slightly less optimal than the full unbiased ACE sampler, it still drastically outperforms uncorrected NR (compare Table 4 ACE-lite to Table 3 NR). ACE-lite shows that a heuristic sampler can improve substantially once the path is repaired, but this is **not** the same as proving the heuristic is exact. Thus, Thms 2.1 and 2.2 fundamentally solve MPC by repairing the path, whether one samples that repaired path using our unbiased sampler (Thm 2.3) or a heuristic one.
>
> **[Q2] Why do CFG/FKC still converge to meaningful samples, and what is $p'_t$?**
> First, producing a "meaningful sample" is distinct from sampling the "intended target distribution." For example, in 1D, the value '0' is a valid sample from a Dirac delta, a standard Gaussian, or a uniform distribution on $[-1, 1]$. We cannot determine the distribution just by looking at a single sample.
>
> When MPC occurs, the intended path $p^*_t$ ceases to exist as a normalizable density, but the mixed score field may remain finite. The numerical SDE solver, therefore, continues to run, but it is no longer tracking the intended ratio path. Instead, it follows another well-posed density evolution, which we denote $p'_t$. In the Gaussian example discussed in our response to **Reviewer HnNF**, this off-path evolution ends at $p'=\mathcal{N}(0,2.415)$ instead of the intended $\mathcal{N}(0,7)$. Since both are supported on the real line $\mathbb{R}$, samples from $p'$ may locally appear "valid" (e.g., forming a chemically stable molecule), but they are drawn from an entirely different distribution that can fail to satisfy the intended joint constraints (e.g., failing to bind to the specific protein pocket).
>
> **[Q3] Why is the non-integrable region in Figure 2 only partial?**
> This partial collapse is exactly what makes MPC deceptive. In Figure 2, the ratio path is well-defined at the endpoints, but the time-dependent coefficient $C(t)$ becomes non-positive only over an intermediate window. Thus the path can be valid near the beginning and end, yet non-integrable in the middle.
>
> Specifically, the ratio path is well-defined at the endpoints, ensuring $C(t) > 0$. However, because we compose heterogeneous experts with varying noise schedules and negative exponents, the individual variances $(\alpha_t^{(i)})^2$ evolve at entirely different rates.
>
> In the intermediate region, negative terms from the denominator can temporarily overpower the positive terms. This drives $C(t) < 0$ for a brief, hazardous window, causing the combined density to physically flip from a decaying Gaussian ($e^{-cx^2}$) to an exploding exponential ($e^{+c'x^2}$, for constants $c, c' > 0$). This tears a non-integrable hole in the middle of the path despite the endpoints being perfectly valid. Theorem 2.2 (the bump function) acts as a mathematical bridge, temporarily boosting the positive exponents to safely cross this gap without altering the valid endpoints.

---

> > ### Author Rebuttal · Reviewer_JJVP · 2026-04-02
> >
> > I thank the authors for their detailed rebuttal, especially the discussion of my Q2 and Q3. I have no further concerns and would like to maintain my positive score.

---

> > > ### Author Response · Authors · 2026-04-06
> > >
> > > We sincerely thank the reviewer for maintaining their positive score and for their encouraging feedback. We are glad that our explanations adequately addressed your questions. We are committed to incorporating these discussions and intuitive examples into the revision to further strengthen the paper.

---

### Official Review · Reviewer_3MV9 · 2026-03-09

**Soundness:** 3
**Presentation:** 2
**Significance:** 3
**Originality:** 4
**Overall Recommendation:** 4
**Confidence:** 2

**Summary:**

This paper studies inference-time composition of pretrained diffusion or flow models by multiplying and dividing their time-marginal densities, i.e. a ratio-of-densities path. The core claim is that the intermediate composed density can become non-normalizable even when the start and end distributions are perfectly valid in heterogeneous composition (using several generative models trained with different noise schedules, different dimensions, and/or possibly negative exponents). They call this failure Marginal Path Collapse (MPC). Their solution is a theory-driven correction called ACE that makes the exponents time-dependent so the path remains valid throughout sampling.

**Compliance With Llm Reviewing Policy:**

Affirmed.

**Final Justification:**

Originally, I misunderstood certain aspects of the work, especially its motivation. I raised this concern to the authors using a simple 2D example. They then built on that example to clarify their motivation, which made the work much clearer to me.

Overall, the work appears logically sound and mathematically rigorous. However, I am not sufficiently familiar with some of the relevant background literature, so I cannot fully assess certain aspects of the paper. I also believe the presentation could be improved by including more intuitive explanations and simple examples, which the authors indicated they will add in the revision.

I initially assigned a score of 3 because of my misunderstanding. After the authors’ clarification, I increased my score to 4. However, I kept my confidence very low, as I am not familiar with this specific area.

**Key Questions For Authors:**

1. Specifically, I do not understand the practical use of this idea in the application described in Section 2.5. As I understand it, the authors multiply two densities because they want to enforce two conditions simultaneously:
(1) the scaffold should match the required scaffold, and
(2) the full molecule should bind to the pocket.
However, I do not find this intuition convincing. Simply multiplying two densities does not, in general, guarantee that both conditions are satisfied. The paper formulates scaffold decoration as
$$p\left(M \mid T^{s c}, P\right) \propto \frac{p\left(M^{s c} \mid T\left(M^{s c}\right)=T^{s c}\right) p(M \mid M \leftrightarrow P)}{p\left(M^{s c}\right)},$$
where the scaffold-topology condition acts only on $M^{sc}$, while the pocket-binding condition acts on the full molecule $M=(M^{sc}, M^R)$.

   For example, consider a 2D variable $x = (x_1, x_2) \in \mathbb{R}^2$ and suppose I want to impose two constraints:
   (1) points should lie on the $x$-axis, and
   (2) points should lie on the $y$-axis.
   Define two distributions in 2D, $x=\left(x_1, x_2\right) \in \mathbb{R}^2$: $q_A(x) \propto \exp \left(-\frac{x_1^2}{0.2^2}\right)$ and $q_B(x) \propto \exp \left(-\frac{x_2^2}{0.2^2}\right)$. Individually, these correspond to a horizontal stripe and a vertical stripe in $\mathbb{R}^2$. If both constraints are to hold simultaneously, then the only valid point should be the origin. However, multiplying the two gives the standard Gaussian. In diffusion terms, one model pushes toward $x_1 = 0$ and the other pushes toward $x_2 = 0$. But this does not seem equivalent to enforcing the intersection of the two constraints in a strict sense. Therefore, I do not yet understand why multiplying densities should be interpreted as satisfying both conditions.

2. As a follow-up to Question 1, could you clarify the practical usage of this composition of diffusion/flow models, or provide more use cases?

3. How much of the benefit is from “validity restoration” versus simply changing the guidance schedule?

**Limitations:**

Yes

**Strengths And Weaknesses:**

Disclaimer: I do have a background in generative models, especially diffusion and flow models, but I am not familiar with steering methods or the heterogeneous composition of diffusion/flow models. I am unable to assess the practical use, novelty, and relevance of this work.

# Strengths:

1. This work seems to be theoretically well-founded.

# Weaknesses:

1. This paper is very confusing, even to an audience with sufficient background in diffusion/flow models. Additional background knowledge, simple examples, and analogies would be appreciated. For example, what does it mean to combine multiple diffusion/flow models? Does combining diffusion models simply correspond to a weighted sum of scores? Another example would be to explain more intuitively what it means when the density does not exist in practical terms. In other words, the normalization constant is infinite. What does this imply for actual sampling?

2. The application described in this paper seems like it could also be achieved by a single generative model specifically trained for the task, yet in the experiments, only steering methods (NR, FKC) are compared.

3. ACE uses a bump function with hand-tuned parameters $B_1, B_2$. Although sensitivity analysis and recommended conservative values are provided, but this still means the method introduces new tuning parameters.

4. See question #1.

---

> ### Author Rebuttal · Authors · 2026-03-31
>
> We sincerely thank the reviewer for their feedback. We address their questions below and will include the discussions in the appendix.
>
> **[W1, Q1a] Density Products vs. Exact Bayesian Ratios.**
> We do **not** simply multiply two densities to enforce two conditions simultaneously. In general, multiplying conditional densities does *not* guarantee joint satisfaction. Our framework instead relies on exact Bayesian decompositions such as Eq. (10), detailed in **[Q1b]**.
>
> That said, the specific 2D axis example reflects a different setting. Let constraint $A,B$ restrict points to the $x-,y-$ axis, respectively, such that $p(x_1,x_2|A)=p_1(x_1)\delta(x_2),p(x_1,x_2|B)=\delta(x_1)p_2(x_2)$. Their product is$$p(x_1,x_2|A) p(x_1,x_2|B)=\delta(x_1)\delta(x_2)$$ which is a point mass at the intersection $(0,0)$. Without the Dirac $\delta$ terms, the extension of a 1D Gaussian to 2D is not normalizable ($\int_\mathbb{R}\int_\mathbb{R}\exp(-\frac{x_1^2}{\sigma^2})dx_2dx_1=\infty$) and renders the density undefined.
>
> **[Q1b] When multiplying/dividing densities exactly represents joint satisfaction.**
> In scaffold decoration, ($x_1$: 3D scaffold, $x_2$: R-group, $A$: 2D topology, $B$: pocket), the scaffold fully determines its topology, meaning $A\perp(x_2,B)|x_1$. By applying Bayes' rule and this **conditional independence**, the true target decomposes exactly into a ratio (Eq (10)):
> $$p(x_1,x_2|A,B)\propto p(x_1,x_2,B|A)\propto p(x_1|A)p(x_2,B|x_1)\propto p(x_1|A)\frac{p(x_1,x_2|B)}{p(x_1)}$$
> Thus, in this setting, the ratio on the RHS **exactly** represents the desired joint target up to normalization.
>
> **[W1b] Combining Models & Practical Implications of Non-Existence** At the score level, the composed path induces the weighted score sum. The key point is that this score is meaningful only when it comes from a valid normalizable ratio path; otherwise the solver can still run, but it follows an unintended path $p'_t$ rather than the desired $p^*_t$.
>
> So "combining models" is not blindly adding scores; it means multiplicatively joining existing probability paths (e.g., individually enforcing single conditions) to construct a new target path (e.g., enforcing multiple conditions; Sec. 2.1), a definition we share with FKC (Skreta et al., 2025). Intuitively, the probability path is a *track* from noise to data, and the SDE sampler is a *vehicle* following local steering instructions (the score). Combining experts merges blueprints to build a new track. Under Marginal Path Collapse, the combined blueprint demands a mathematical impossibility (e.g., negative variance), so a section of the track ceases to exist.
>
> Practically, because the local steering instructions (score) remain finite, the SDE solver does not crash. Instead, with the underlying track gone, the vehicle silently drives off-route to the wrong destination (see exact SDE detachment derivation in **HnNF [Q3]**). ACE solves this: Thm 2.1 detects collapse, Thm 2.2 repairs the blueprint, and Thm 2.3 safely samples from the repaired path.
>
> **[W2, Q2] Why Steer? (Single Model vs. Composition with ACE)**
> **Table 4** already compares against specialized fixed-pose baselines (Delete, AutoFragDiff); ACE outperforms them on OSR and docking while remaining competitive on drug-likeness. Importantly, composition offers **flexibility**. Training specialized models demands custom architectures, heavy computing, and curated multi-constraint datasets. ACE instead reuses simpler pretrained experts as modular Lego blocks and recombines for novel tasks without retraining, even extending to compositional image generation (App. E.5).
>
> **[Q2] More use cases: Fragment linking.** We also demonstrate this on a harder task: fragment linking (Example E.3). Generating a molecule $M$ containing two fragment topologies $T_1,T_2$ binding to pocket $P$ is given by:$$p_w(M)\propto p(M)\left(\frac{p(M|T_1,T_2,P)}{p(M)}\right)^w$$ACE combines two conformer/de novo experts with an SBDD model (see **Eq (E.11)** for formal derivations) and applies the bump function to avoid path collapse. We evaluated on non-overlapping fragment pairs sampled from CrossDocked test set: ACE achieved strong performance (OSR $\ge$ 0.45), remained highly competitive with FFLOM (Jin et al., 2023, a specialized fragment linking model), and outperformed FKC, NR, which degraded due to path collapse (see https://anonymous.4open.science/r/anonymous-63B1/tab1.png).
>
> **[W3] Hyperparameter Tuning** Our Path Existence Criterion gives a sufficient lower-bound on the magnitude needed to avoid collapse, greatly reducing blind tuning of $B_1,B_2$.
>
> **[Q3] Validity Restoration vs. Changing Guidance Schedules** We additionally evaluated multiple time-varying schedules from prior work (Wang et al., 2024, see **Araw [W3]**). Despite tuning, these schedules underperformed ACE in our settings because time variation alone does not guarantee a valid path. This indicates that **validity restoration, not schedule design, drives the improvement**.

---

> > ### Author Rebuttal · Reviewer_3MV9 · 2026-04-02
> >
> > I will increase my rating to 4, as the paper appears logically sound given the authors' clarification, and the mathematical arguments seem to be well supported.
> >
> > I will keep my confidence at 2, however, because I am not fully confident in my assessment of this work due to my limited background in diffusion and flow steering.

---

> > > ### Author Response · Authors · 2026-04-06
> > >
> > > We sincerely thank the reviewer for raising their score and for dedicating the time to evaluate the logical soundness and mathematical arguments of our work. We deeply appreciate your constructive engagement. We will ensure that the expanded clarifications provided in our rebuttal are integrated into the final revision to make these theoretical concepts as clear and accessible as possible.

---

### Official Review · Reviewer_HnNF · 2026-03-10

**Soundness:** 2
**Presentation:** 4
**Significance:** 2
**Originality:** 3
**Overall Recommendation:** 4
**Confidence:** 4

**Summary:**

The authors identify a failure mode, Marginal Path Collapse, which occurs when compositions of multiple samplers are applied to ratio-of-densities inference steering. Intuitively, the proposed Path Existence Criterion detects collapsing points along the probability path. The proposed method, ACE, adjusts the steering weights over time to ensure the composed path remains valid, rather than relying on numerator shrinking or other heuristic fixes.

**Compliance With Llm Reviewing Policy:**

Affirmed.

**Final Justification:**

An issue about the main experimental setting is addressed.
After seeing the second round response: My concern has been addressed. I am impressed with this work, and the authors' observation and the proposed fix are simple but interesting. I will give a overall score of 4, and the only reason that it is not 5 is the domain issue, not the authors': finding a complex task of Bayesian components, whose score models can also produce the same structured output for 'steering' is hard.

**Key Questions For Authors:**

1. **My key concern** is about the assumption that the Bayesian composition of densities holds along the entire diffusion path. The paper assumes that if the endpoint distribution satisfies a relation such as $p(x) ∝ q¹(x)q²(x)/q³(x)q⁴(x)$, then the noisy marginals should satisfy $h_t(x) = q^1_t(x)q^2_t(x)/q^3_t(x)q^4_t(x)$. However, in many applications (e.g., molecular scaffold decoration), the noisy samples do not correspond to valid objects, so the probabilistic interpretation of these intermediate densities becomes unclear. In practice, inference steering is implemented via score combination rather than enforcing a true probabilistic relation between the noisy densities. Therefore, it is unclear whether the collapse of the constructed density path necessarily indicates a practical failure of the sampler, or whether it only reflects a limitation of the assumed probabilistic formulation. **If it is addressed, I will increase the overall score at least to 4.**

2. Since inference steering is the main task studied in this paper, it would be helpful to clearly explain the exact density functions involved in the scaffold decoration example and the Bayesian relationships between them. In particular, how the different conditional densities combine to form the composed target distribution should be explicitly written.

3. Rather than the high-level discussion in Appendix A.2, it would be useful to clarify the effect of Marginal Path Collapse from the score-function perspective. Even if a density p becomes non-integrable, the score field may still remain finite. For example, if p' = p + 1, then the gradients are identical. In such a case, the score used in the sampler may still be well defined locally. As sampling proceeds toward smaller timesteps, could the dynamics implicitly recover a valid distribution? Clarifying whether collapse truly changes the sampling dynamics would strengthen the practical significance of the theory.

If they are addressed, I will increase my overall score.

**Limitations:**

yes

**Strengths And Weaknesses:**

## Strengths
- The paper is well motivated. When performing compositional inference steering using multiple models, it is important to understand whether the resulting probability path remains valid.
- The presentation is clear and well organized. The theoretical development, diagnostic analysis, and algorithmic solution are presented in a coherent way.
- The scaffold decoration experiment is well designed and represents a realistic application scenario where ACE could be useful.

## Weaknesses
The following questions mainly concern clarification of the practical implications of Marginal Path Collapse and the assumptions behind the proposed theory.

---

> ### Author Rebuttal · Authors · 2026-03-31
>
> We sincerely thank the reviewer for their thoughtful feedback and for recognizing our scaffold decoration experiments as a realistic application. We address your questions below and will include these expanded discussions in our camera-ready appendix.
>
> ### Part 1 (Q1)
> **[Q1] Does the paper assume the Bayesian composition holds along the entire path?**
> We do **not** assume Bayesian composition of densities holds along the entire diffusion path. In fact, demonstrating that the widely used naive ratio $h_t(x) = \frac{q^1_t(x)q^2_t(x)}{q^3_t(x)q^4_t(x)}$ frequently breaks sampling and fixing this via adaptive exponents (ACE) is our central contribution.
>
> The misunderstanding may arise from interpreting Eq. (8) and (10) as decomposing an **intermediate** Bayesian relation $p_t(M_t|T^{sc},P),p_t(X_t,Y_t|A,B)$ which is not the case. To clarify, Eq. (8) and (10) decompose the *target* distribution $p$, not the *intermediate* distribution $p_t$. Thus, we only require Bayesian decomposition at the final timestep where samples correspond to valid objects and interpretation is clear.
>
> Thus the lack of physical interpretation for noisy intermediate samples is not the issue. Diffusion models are transport mechanisms; the issue is whether the constructed intermediate path (composing marginal experts $q^{(i)}_t$) remains a valid normalizable probability path.
>
> ### Part 2 (Q2)
> The ratio formulation rigorously emerges from Bayesian conditional independence. Let $M=(M^{sc},M^{R})$ be the 3D molecule ($M^{sc}$: 3D scaffold, $M^{R}$: R-group) with 2D topology $T^{sc}$ and pocket $P$.
>
> Since the 3D scaffold perfectly contains its 2D topology, $T^{sc}$ yields no extra information about $M^R$ or $P$ given $M^{sc}$. Thus: $T^{sc} \perp (M^{R}, P)|M^{sc}$.
>
> By Bayes' rule and this exact conditional independence, our target distribution decomposes (as in Eq. 10):
> $$\begin{aligned} p(M^{sc},M^{R}|T^{sc}, P) &\propto p(M^{sc},M^{R}, P|T^{sc}) \\\\ &= p(M^{sc}|T^{sc}) p(M^{R}, P|T^{sc}, M^{sc}) \\\\ &= p(M^{sc}|T^{sc}) p(M^{R}, P|M^{sc}) \\\\ &\propto p(M^{sc}|T^{sc}) \frac{p(M^{sc},M^{R}|P)}{p(M^{sc})} \end{aligned}$$
>
> Thus, the Conformer Generator $p(M^{sc}|T^{sc})$ and SBDD Model $p(M|P)$ (numerators) with the Unconditional 3D Prior $p(M^{sc})$ (denominator) combine to form the composed target.
>
> ### Part 3
> **[Q3, Q1] Finite score fields and implicit recovery under path collapse.** The reviewer's $p' = p + 1$ intuition is correct: a non-integrable function's score ($\frac{\nabla p}{p+1}$) can remain finite and locally well-defined. We agree the mixed score of a collapsed path can remain finite, allowing the sampler to run and generate some distribution $p'$. However, this simulated path $p'_t$ (which is a valid SDE path on its own) structurally detaches from the intended $p_t$ and **cannot implicitly recover**.
>
> To explicitly demonstrate this detachment, consider a 4-expert Gaussian path (from Fig. 2). Let $h_t(x) = \frac{q^1_t(x)q^2_t(x)}{q^3_t(x)q^4_t(x)}$ with $q^{(1)}_t \sim {N}(0,\sigma_1^2(t))$, $q^{(2)}_t \sim {N}(0,\sigma_2^2(t))$, and $q^{(3)}_t=q^{(4)}_t \sim {N}(0,\sigma_3^2(t))$.
>
> Using schedules $\sigma_1^2(t)=(1-t)^2+0.5t^2$, $\sigma_2^2(t)=(1-t)^2+7t^2$, and $\sigma_3^2(t)=1.5(1-t)^2+t^2$, the intended path is $h_t(x) \propto \exp(-\frac{1}{2} C(t) x^2)$ where $C(t) = \frac{1}{\sigma_1^2(t)}+\frac{1}{\sigma_2^2(t)}-\frac{2}{\sigma_3^2(t)}$. The intended endpoints are $p^* _0 = {N}(0,1.5)$ and $p^* _1={N}(0,7)$.
>
> * **The Collapse:** At $t=0.5$, $C(0.5) = -1/30$ drops below zero, making $h_{0.5}(x) \propto \exp(x^2/60)$ non-integrable.
> * **The Paradox:** The mixed score $s_\text{mix}(x,t) = -C(t)x$ remains perfectly finite ($x/30$ at $t=0.5$). The SDE is well-posed and does not crash.
> * **The Detachment:** By standard Fokker-Planck arguments, any Gaussian path $p_t={N}(0,\sigma_t^2I)$ can be simulated by the SDE:
>
> $$dX_t = \left(\frac{g_t^2}{2}-\sigma_t\dot \sigma_t\right)s(X_t,t)dt + g_tdW_t$$
>
> We can run the well-posed SDE with the **mixed score** by choosing a reference schedule $\sigma_1(t)$ and constant noise $g_t=\sqrt{2}$ (though the following structural failure applies universally to any choice of reference schedule and $g_t$):
>
> $$dX_t = (1 - \sigma_1(t)\dot\sigma_1(t)) s_\text{mix}(X_t,t)dt + \sqrt{2} dW_t$$
>
> By Ito's Lemma, the variance $V(t) := \mathbb{E}[X_t^2]\ge0$ rigorously follows:
> $$\dot{V}(t) = (3t - 4) C(t) V(t) + 2$$
> To succeed, $V(t)$ must track $1/C(t)$ to reach $V(1)=7$. However, since $V(t) \ge 0$, $V(t)$ cannot become negative. Therefore, once $C(t) < 0$ the sampler cannot remain on the intended path: the target variance $1/C(t)$ is negative and hence not realizable by any probability law. In our concrete well-posed mixed-score SDE above, numerical integration from $V(0)=1.5$ yields $V(1) \approx 2.415 \neq 7$, so the terminal marginal is also wrong. More generally, any later return to the correct endpoint would require specially tuned off-path dynamics rather than automatic recovery.

---

> > ### Author Rebuttal · Reviewer_HnNF · 2026-04-02
> >
> > Thank you for your detailed response. I will raise my score to 3, and I still have the following concern:
> >
> > - Q2: Please include this clarification in the revision.
> >
> > - Q3 & Q1: In your derivation for the example, the assumption that
> >   $V(t) := \mathbb{E}[X_t^2] \ge 0$ corresponds to the variance is not entirely trivial. In particular, when combining multiple score functions, it is not guaranteed that $\mathbb{E}[X_t] = 0$ still holds, even if each individual score-based process satisfies this property under its own setting. Therefore, identifying $V(t)$ directly with the variance may require additional justification.
> >
> >   As a result, my main concern remains: whether the breakpoint (where the intended path becomes invalid) necessarily leads to failure of the generative process is still not fully clear.
> >
> > I would appreciate a more detailed explanation on this point.

---

> > > ### Author Response · Authors · 2026-04-06
> > >
> > > We sincerely thank the reviewer for raising their score and for engaging deeply with the mathematical nuances of our framework. We will gladly include these expanded discussions, including Q2, in the revision.
> > >
> > > Below, we derive the variance $\text{Var}(X_t):=\mathbb{E}[(X_t-\mathbb{E}[X_t])^2]$ by explicitly showing $\mathbb{E}[X_t]=0$, yielding $\text{Var}(X_t)=\mathbb{E}[X_t^2]=:V(t)$, and clarify that collapse does lead to failure.
> > >
> > > ## 1. Detailed explanations for the Gaussian example
> > >
> > > The reviewer is correct that composing arbitrary score functions of zero-mean processes generally does not guarantee $\mathbb{E}[X_t]=0$. We do not assume zero mean for arbitrary composed processes.
> > >
> > > However, $V(t)$ is exactly the variance here because for **this specific setup**, the mixed score $s_\text{mix}(x,t) = -C(t)x$ is linear in $x$, inducing a linear drift. Moment dynamics follow directly from standard Itô calculus for linear SDEs (e.g., Øksendal, *Stochastic Differential Equations*).
> > >
> > > **[Itô Derivation]**
> > > We may write the SDE induced by the *mixed* score as:
> > > $$dX_t = a(t)s_\text{mix}(X_t,t)dt + g_tdW_t, \quad s_\text{mix}(x,t)=-C(t)x$$
> > > where $a(t)=1-\sigma_1(t)\dot\sigma_1(t)$. Since the initial distribution is $X_0\sim p^* _0=\mathcal{N}(0,1.5)$, the initial mean is $m(0):=\mathbb{E}[X_0]=0$.
> > >
> > > Taking the expectation of the SDE eliminates the stochastic part, yielding the ODE for $m(t):=\mathbb{E}[X_t]$:
> > > $$\dot m(t) = -a(t)C(t)m(t), \quad m(0)=0$$
> > > Solving this homogeneous linear ODE yields $m(t)=0$ for all $t$. Thus, the second moment is the variance: $V(t):=\mathbb{E}[X_t^2]=\text{Var}(X_t)$.
> > >
> > > By Itô's formula, $d(X_t^2) = 2X_tdX_t + (dX_t)^2$. Using standard Itô multiplication rules, $(dX_t)^2$ simplifies to $g_t^2 dt$. Grouping the terms yields the SDE for the squared process:$$d(X_t^2) = ( -2a(t)C(t)X_t^2 + g_t^2 ) dt + 2g_t X_t dW_t$$Taking the expectation and using linearity yields the ODE for $V(t)$ (since the Itô integral term $2g_t X_t dW_t$ has zero expectation):
> > > $$\dot V(t) = -2a(t)C(t)V(t) + g_t^2$$
> > > For the concrete sampler used in the rebuttal ($g_t=\sqrt{2}$ and $2a(t)=4-3t$), all intermediate distributions generated by the sampler have zero mean and variance $V(t)$. However, numerically integrating the ODE from $V(0)=1.5$ yields an actual variance of $V(1)\approx 2.415$, completely missing the intended target (zero mean and variance 7).
> > >
> > > ## 2. Does the breakpoint necessarily lead to failure of the generative process?
> > >
> > > Yes, the breakpoint necessarily leads to a failure of the generative process.
> > >
> > > **First, regarding intermediate correctness, the process strictly fails.** At the breakpoint where $C(t)<0$, the intended unnormalized density $h_t$ becomes non-integrable, and the probability path $p^* _t=h_t/Z_t$ ceases to exist. Standard samplers (e.g., CFG, NR) heuristically attempt to bypass this by simply plugging the invalid mixed score into an SDE with predefined coefficients. While this runs numerically, it simulates an unintended path $p'_t$ lacking a ground truth, rendering intermediate correctness undefined.
> > >
> > > **Second, regarding terminal alignment, divergence is a mathematical consequence of integrating this unintended path.** As shown analytically in our Gaussian example above, the mixed score SDE yields a terminal variance of $V(1) \approx 2.415$. Since the intended target variance is 7, terminal divergence from the intended target is provably inevitable in the Gaussian case.
> > >
> > > **Third, empirically, this strict divergence severely degrades practical performance across all tested scenarios.** In real-world applications where closed-form analysis is infeasible, we demonstrate the severity of this failure mode via *empirical* evidence. Our 2D, scaffold decoration, and fragment linking experiments (see response to Reviewer 3MV9 [Q2]) consistently demonstrate that samplers operating over collapsed paths degrade significantly in terminal alignment and final sample quality.
> > >
> > > Furthermore, simply tuning time-varying exponents used in existing literature is insufficient to restore performance (see response to Reviewer Araw [W3]). Comparing Tables 2 and E.10 confirms unbiased samplers like FKC fail *specifically* due to the breakpoint: FKC shows low distributional errors on non-collapsed paths (Table E.10) but breaks under path collapse (Table 2). Correcting a collapsed path via ACE alone (Tables 2, 3), without changing other components, completely restores distributional fidelity and boosts docking metrics. These results provide compelling empirical evidence that the breakpoint leads to the failure of the generative process.
> > >
> > > Therefore, hoping for a "lucky recovery" using standard samplers on collapsed paths is mathematically unjustified and empirically fragile. Our contribution is precisely to identify and replace this vulnerability with a method that is provably well-defined by construction (Thms 2.1 & 2.2) and enables robust, unbiased sampling (Thm 2.3).

---

### Official Review · Reviewer_Araw · 2026-03-16

**Soundness:** 2
**Presentation:** 4
**Significance:** 4
**Originality:** 3
**Overall Recommendation:** 5
**Confidence:** 3

**Summary:**

In the context of inference-time steering of score-based generative models, the authors identify Marginal Path Collapse, a failure mode in which the intermediate density associated with the time/dimension-varying guidance schedule is non-normalizable. To address this, they claim to provide a necessary and sufficient Path Existence Criterion (PEC) for well-defined intermediate densities as a diagnostic, assuming compactly supported densities. They then propose Adaptive Path Correction with Exponents (ACE), which extends Feynman-Kac weighted SDE to time-varying exponents. Empirically, ACE outperforms constant-exponent baselines wrt steered property metrics on a drug design task with three experts.

**Compliance With Llm Reviewing Policy:**

Affirmed.

**Final Justification:**

I recommend acceptance, as the work is theoretically well motivated while proposing practical algorithms for diagnostics and steering.

The rebuttal addressed my main theoretical concerns regarding the boundary condition and the gap between Thm 2.2 and B.2. For the former, the authors modified their claims. For the latter, they updated the text and added a corollary. Empirically, I believe the additional time variation baseline helps isolate the importance of validity preservation, as claimed.

**Key Questions For Authors:**

Addressing the following questions regarding the Theorems 2.1 and 2.2 would resolve my current concerns on soundness, additional context given above. Please provide revisions, or clarifications if my understanding is incorrect.

1. Regarding Thm 2.1, what can we say about the $C_k(t) = 0$ boundary case? It seems important to establish the "necessary and sufficient" claim.
2. Could Thm 2.2 be revised to make the bump function protocol agree with the PEC of Thm 2.1? Perhaps additional assumptions on which dimensions the experts act on, or the parameterization of the bump function (e.g., forcing it to $d$-dimensional)?

**Limitations:**

Yes

**Strengths And Weaknesses:**

### Soundness
I focused on verifying the theoretical claims, as I'm not too familiar with the experimental setup of flexible-pose scaffold decoration. They seem sound overall, and I outline some minor concerns below and in "Questions."
- The proof of Theorem 2.1: While the arguments for sufficiency for path existence ($C_k(t) > 0$) and the sufficiency for MPC at $k*, t*$ ($C_{k*}(t*)<0$) look correct to me, the theorem doesn't seem to address the $C_k(t) = 0$ case.
- The statement of Thm 2.2 in the main text claims there's a $\{\tilde \gamma_i(t)}_{i=1}^n$ such that exponent boundary values are satisfied for all $i$ and $C(t) = \min_k C_k(t) > 0$ for all $t$. But the appendix version Thm B.2 replaces the latter condition with a scalar sum over experts, $S(t, ...) > 0$ for all $t$. But it's possible for $S(t, ...)> 0$ for all $t$ while $C_k(t) < 0$ for some $k$. This discrepancy seems problematic, since, if $S < 0$, then making it positive with a bump function may not make all $C_k$ positive, so the path still collapses as per Thm 2.1.
- The proof of Thm 2.3 seems correct to me, largely following from the expectation identity (Prop A.1) given in Skreta et al. 2025.
- I believe the experiments should include at least one time-varying exponent baseline, as constant exponents are known to perform poorly. Any of the time-varying heuristics discussed in Related Work can be used. Including this would help disentangle which aspect of ACE helps more: the bump function parameterization or the time-varying exponent in general.

### Presentation
Overall, the presentation is very clear, and the development from PEC, to ACE existence, to ACE sampling easy to follow. While the
- Could the figures be made higher resolution, with legible axis labels (esp. Figure 4)?
- Typo: Theorem B.2 -> B.3 in Sec 2.4? Please double check all references to appendix sections.

### Significance & novelty

The proposed PEC diagnostic and ACE are of practical relevance, particularly for multi-modal molecular modeling where each expert model may act on some task- or modality-specific dimensions. The PEC is lightweight to compute, and so is the ACE per expert. ACE-lite is a useful ablation and allows compute gains. The generalization of Feynman-Kac correctors to $\dot \gamma_i(t) \neq 0$ is important and interesting. The experimental setup emphasizes the key advantages of PEC and ACE, and the results are compelling.

---

> ### Author Rebuttal · Authors · 2026-03-31
>
> We sincerely thank the reviewer for their rigorous feedback. We address the soundness concerns below and provide revisions. We will also increase Figure 4 resolution and fix the appendix theorem cross-reference typo.
>
> ## [W1, Q1] Regarding Theorem 2.1 and the $C_k(t)=0$ boundary case
> The reviewer correctly identified that claiming "necessary and sufficient" for $C_k(t) = 0$ is an overstatement. At $C_k(t) = 0$, the dominant quadratic term vanishes (Eq B.7), so sub-quadratic terms determine existence or collapse.
>
> As we show below, $C(t)=0$ can yield either outcome. Hence, $C(t)>0$ and $C(t)<0$ represent the **sharpest possible universal conditions**. We will revise the text to claim *"sharp sufficient criterion"* instead of "necessary and sufficient" and detail this boundary behavior in the appendix.
>
> **Example.** Consider a 3-expert 1D unnormalized path $h_t(x) = \frac{q_t^{(1)}(x) q_t^{(2)}(x)}{q_t^{(3)}(x)}$ between noise ${N}(0, 1)$ and data. Experts 1,2 target ${U}[-a,a]$ (uniform distribution) via $(\alpha_t, \beta_t)$; Expert 3 targets ${U}[-b,b]$ via $(\tilde{\alpha}_t, \tilde{\beta}_t)$.
> * **Valid Boundaries:** $h_0 = {N}(0,1)$, and at $t=1$, $h_1(x)=\frac{b}{2a^2}1_{[-a,a]}(x)$ (for $a \le b$). Both are normalizable.
> * **Intermediate State:** Let $\alpha_t=1-t^2$, $\tilde \alpha_t = 1-t$, $\beta_t=\tilde\beta_t=t$. At $t^* =\sqrt{2}-1$, $\alpha_{t^* }^2=2\tilde \alpha_{t^* }^2$, causing $C(t^* )=0$.
> * **Tail Expansion:** By exact convolution (Lemma B.2) and the Mills ratio ($\bar\Phi(z)/(\frac{e^{-z^2/2}}{\sqrt{2\pi}}) \sim 1/z$), the experts' tails decay as $p_t(x) \sim \frac{C_t}{|x|-a\beta_t} \exp(-\frac{(|x|-a\beta_t)^2}{2\alpha_t^2})$ where $\sim$ means that the quotient of the two functions converges to 1 as $|x|\to\infty$ (Small, Christopher G. (2010)). For $q^{(3)}$ we may simply replace $a$ with $b$. Canceling quadratic terms at $t^* $, the ratio's tail behaves as (up to a constant factor):
> $$h_{t^* }(x)\sim \frac{1}{|x|}\exp \left( \frac{2(a\beta_{t^* }-b\tilde\beta_{t^* })}{\alpha_{t^* }^2}|x| \right) \text{ as } |x|\to\infty$$
>
> Near $x=0$, $h_{t^* }$ is finite and positive. Integrability depends on the tails:
> 1. **Collapse** ($a\beta_{t^* }=b\tilde\beta_{t^* }$): Exponent is zero. Tails decay as $1/|x|$, which diverges logarithmically.
> 2. **Existence** ($a\beta_{t^* } < b\tilde\beta_{t^* }$): Exponent is strictly negative ($-c|x|$). Tails decay exponentially ($\sim \frac{1}{|x|} e^{-c|x|}$), making it integrable.
>
> ## [W2, Q2] Resolving discrepancy between Thm 2.2 and Thm B.2
> The discrepancy between the coordinate-wise $C_k(t) > 0$ and the scalar sum $S(t, \\{\gamma_i\\}_{i=1}^n) > 0$ in Thm B.2's proof is seamlessly resolved by applying Thm B.2's exact logic repeatedly across coordinate subsets. Thus, the statement of Thm 2.2 remains valid. We will update the text to say "changing $\tilde{\gamma}_j(t)$ for a covering subset $J$" instead of "one index $j$" and include this generalized step after Thm B.2 as a corollary:
>
> **Corollary.** For partial-support, apply the following steps to guarantee $C_k(t) > 0$ for all $k$:
> 1. **Subset Application:** Let $I_{(k)} = \\{i : k \in I_i\\}$ be the experts acting on coordinate $k$. $C_k(t)$ is exactly the scalar sum $S(t, \\{\gamma_i\\} _{i\in I _{(k)}})$.
> 2. **Base Case:** Since Thm 2.2 assumes $C(0)=\min_kC_k(0) > 0$ and $\alpha^{(i)} _0=1$, we have $S(0,\\{\gamma_i\\} _{i\in I _{(k)}})=C _k(0)>0\forall k$.
> 3. **Coordinate-wise Bumps:** Applying Thm B.2 to $I_{(k)}$ yields nonnegative bumps $b_{i,k}(t)$ for $i\in I_{(k)}$ ensuring $C_k(t) > 0 \forall t\in[0,t_\text{end}]$.
> 4. **Global Fix:** Adding a positive bump strictly increases $S$. Thus, for each expert $i$, summing the required bumps $\tilde{\gamma} _i(t) = \gamma _i(t) + \sum _k b _{i,k}(t)$ guarantees $C _k(t) > 0$ for all $k,t$.
>
> Remark. In our experiments, modifying a single index $j$ sufficed (one expert covered all coordinates). By adding this corollary, we do not need to force $d$-dimensionality nor change the bump.
>
> ## [W3] Time-varying exponent baseline
> To disentangle our bump from general time-varying exponents, we tested empirical dynamic schedules (Wang et al., 2024) on scaffold decoration (3 seeds) and 2D checker (5 seeds): linear increasing ($wrt$), decreasing ($wr(1-t)$), lambda ($wr(1-|2t-1|)$), and V-shaped ($wr|2t-1|$) for $r\in\\{0.5,1,2,4,8\\}$ with a task-specific base weight $w$. For the scaffold, we also tested quadratic bumps $b(t)=B_1 t(1-t)$ on $\gamma_4$ ($B_1\in\\{10,20,30\\}$), scaled to deliberately fail the Path Existence Criterion. Across all samplers (ACE/FKC/NR), these failed to prevent path collapse, degrading performance versus ours. This proves **time variation alone is insufficient in our settings; validity-preserving design is critical.** We will include full results and details in the revision (see (scaffold) https://anonymous.4open.science/r/anonymous-63B1/fig1.png, (2D) https://anonymous.4open.science/r/anonymous-63B1/fig2.png).

---

> > ### Author Rebuttal · Reviewer_Araw · 2026-04-02
> >
> > Thank you for modifying your "necessary and sufficient" claims to account for the boundary case. The resolution of the discrepancy between Thm 2.2 and Thm B.2 makes sense to me as well (and understood that the experiments involved one expert covering all coordinates). These were my main theoretical hangups. On the empirical side, it is very clarifying to see that dynamic schedules alone is not sufficient -- thank you for adding this experiment.
> >
> > I'll raise my score to recommend acceptance.

---

> > > ### Author Response · Authors · 2026-04-06
> > >
> > > We sincerely thank the reviewer for raising their score to recommend acceptance and for their highly constructive feedback. Your keen eye regarding the boundary conditions and the theorem discrepancies tightened our theoretical claims, and the suggestion to test dynamic schedules strengthened our empirical evaluation. We will fully incorporate these theoretical clarifications and the new baseline results into our final revision.

---

### Decision · Program_Chairs · 2026-04-30

**Decision:**

Accept (regular)

**Comment:**

This paper studies an interesting problem - how to stitch several pretrained diffusion or flow models together in inference. The authors identified a failure mode namely Marginal Path Collapse (MPC). The solution ACE adjusts weights and makes the exponents time-dependent so the final overall path remains valid. Reviewers feel the motivation is strong, the technique is theoretically sound, and the method is overall novel. Some concerns regarding the theoretical clarity and presentation were raised, and the rebuttal seemed to resolve the main issues. To me, the viewpoint of this work is interesting but its applications may be limited as the settings are quite restrictive. Nevertheless, I recommend acceptance given that the authors committed to incorporating the discussion in the revision.